# FedQS: Optimizing Gradient and Model Aggregation for Semi-Asynchronous Federated Learning

**Yunbo Li**[*], **Jiaping Gui**[*†], **Zhihang Deng**, **Fanchao Meng**, **Yue Wu**[†],
School of Computer Science, Shanghai Jiao Tong University, Shanghai, China
{li-yun-bo, jgui, dzh1227, mactavishmeng, wuyue}@sjtu.edu.cn

## Abstract

Federated learning (FL) enables collaborative model training across multiple parties without sharing raw data, with semi-asynchronous FL (SAFL) emerging as a balanced approach between synchronous and asynchronous FL. However, SAFL faces significant challenges in optimizing both gradient-based (e.g., FedSGD) and model-based (e.g., FedAvg) aggregation strategies, which exhibit distinct trade-offs in accuracy, convergence speed, and stability. While gradient aggregation achieves faster convergence and higher accuracy, it suffers from pronounced fluctuations, whereas model aggregation offers greater stability but slower convergence and suboptimal accuracy. This paper presents FedQS, the first framework to theoretically analyze and address these disparities in SAFL. FedQS introduces a *divide-and-conquer strategy* to handle client heterogeneity by classifying clients into four distinct types and adaptively optimizing their local training based on data distribution characteristics and available computational resources. Extensive experiments on computer vision, natural language processing, and real-world tasks demonstrate that FedQS achieves the highest accuracy, attains the lowest loss, and ranks among the fastest in convergence speed, outperforming state-of-the-art baselines. Our work bridges the gap between aggregation strategies in SAFL, offering a unified solution for stable, accurate, and efficient federated learning. The code and datasets are available at `https://github.com/bkjod/FedQS_`.

## 1 Introduction

Federated learning (FL) has emerged as a promising paradigm for enabling multiple parties to collaboratively train a shared model without sharing their raw local data [1, 2, 3, 4, 5, 6, 7]. FL has found widespread applications in domains such as healthcare [8, 9, 10] and finance [11, 12, 13]. Among various FL communication modes, semi-asynchronous FL (SAFL) strikes a balance between synchronous FL and fully asynchronous FL, offering a flexible trade-off between model consistency, training latency, and resource utilization [14, 15, 16, 17]. In SAFL, devices operate independently with partial coordination, for instance, through buffered updates [16] or clustered synchronization [18], making it adaptable to heterogeneous network conditions and device capabilities in real-world deployments [19, 20].

Despite its advantages, designing an effective SAFL system presents significant challenges, particularly in selecting an appropriate aggregation strategy. Recent empirical studies [21] reveal that gradient-based aggregation (e.g., FedSGD [1]) in SAFL achieves higher accuracy and faster convergence but suffers from severe fluctuations, whereas

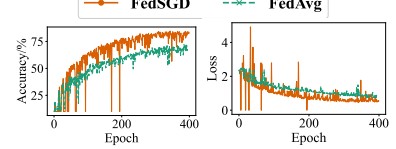

Figure 1: FedSGD vs. FedAvg in SAFL.

---

[*]Equal contribution.
[†]Corresponding authors.

39th Conference on Neural Information Processing Systems (NeurIPS 2025).

model-based aggregation (e.g., FedAvg [1]) offers stability at the cost of slower convergence and reduced accuracy. Figure 1 shows the distinct performance of these strategies when training ResNet-18 on CIFAR-10 in SAFL. While existing SAFL research has focused on straggler mitigation [22], client drift [23], resource heterogeneity [19], and client selection [24], the critical differences between aggregation strategies remain understudied.

Optimizing both gradient and model aggregation in SAFL faces three key challenges: 1) **Lack of Theoretical Understanding:** Current analyses of aggregation discrepancies [21] are empirical, lacking theoretical foundations to guide solution design. 2) **Inherent Aggregation Disparity:** In neural network training, the loss function defines a mapping [25, 26] from the parameter space to the loss space. Gradient aggregation computes first-order derivatives of this mapping, capturing both the direction and magnitude of the local updates in the loss space. In contrast, model aggregation operates directly on the parameter space, which hardly maintains a clear correspondence with the loss space due to the lack of linearity or convexity conditions in loss functions—assumptions rarely satisfied in deep neural networks. 3) **Server- or Client-Centric Limitations:** Existing approaches are predominantly server-centric, relying on a single aggregation method, while client-centric methods struggle with insufficient global information.

To address these challenges, we propose FedQS, the first framework that optimizes both gradient and model aggregation in SAFL. Our key insight is that stale updates and data heterogeneity empirically induce distinct continuity in the optimization trajectories of different aggregation strategies. Building on this observation, we introduce a divide-and-conquer strategy that classifies clients into four types (Fast-but-Strongly-Biased, Fast-and-Weakly-Biased, Straggling-but-Weakly-Biased, and Straggling-and-Strongly-Biased) and adapts their training strategies dynamically. We provide a formal convergence analysis of FedQS, proving that it achieves exponential convergence rates under both aggregation strategies. Our theoretical results demonstrate that FedQS addresses two key limitations in SAFL: the convergence instability of gradient aggregation and the suboptimal convergence capability of model aggregation.

We evaluate FedQS on computer vision (CIFAR-10), natural language processing (Shakespeare), and real-world data (UCI Adult) tasks. Results show that FedQS consistently outperforms state-of-the-art baselines. Compared to the fastest-converging model aggregation and gradient aggregation baselines, FedQS improves average accuracy by 38.98% and 5.65%, respectively, while reducing training time by 58.85% and 3.68%. Against the highest-precision baselines, FedQS achieves 15.74% and 12.93% faster convergence (in rounds) and reduces training time by 72.63% and 48.04%, respectively. Ablation studies validate the impact of each module, while hyperparameter and system setting analyses demonstrate FedQS's robustness. Our work bridges the gap between theory and practice in SAFL, offering a principled approach to harness the strengths of both aggregation strategies.

## 2 Background & Motivation

**Limitations in Existing SAFL Aggregation Methods.** Existing studies in Semi-Asynchronous Federated Learning (SAFL) predominantly focus on optimizing either gradient aggregation or model aggregation, but not both. For gradient aggregation, prior work has addressed challenges such as model convergence [27, 15], optimal aggregation frequency [28, 29], and advanced optimization techniques (e.g., momentum [30]). For model aggregation, solutions target the straggler problem [31, 18, 19]. While AAFL [32] incorporates both gradients and models, it ultimately adopts model aggregation, using gradients only as auxiliary validation signals. To our knowledge, only [21] empirically compares gradient and model aggregations but offers no mitigation for their performance gap. In contrast, we propose a unified optimization framework with theoretical guarantees.

**Federated Learning Basics.** FL involves a server $S$ and clients $\mathcal{C} = \{C_1, C_2, ..., C_N\}$. The server $S$ maintains a global model $w_g$, while each client $C_i$ trains a local model $w_i$ on its dataset $\mathcal{D}_i$ (with $n_i$ samples). The goal is to minimize the global loss: $\min F(w_g) \triangleq \frac{1}{N} \sum_{i=1}^{N} F_i(w_i)$, where $F_i(\cdot)$ is the local objective. Clients access stochastic gradients $\nabla F_i(w_i; \xi_{i,j})$ for each data sample $\xi_{i,j} \in \mathcal{D}_i$.

**Synchronous vs. Semi-Asynchronous FL.** Synchronous FL is based on server-coordinated training where only activated clients upload their local updates in one global epoch with others idling. In contrast, clients in SAFL train autonomously and push updates asynchronously, with aggregation triggers upon conditions (e.g., sufficient updates [15]) at the server.

**Aggregation Strategies.** In Synchronous FL, for gradient aggregation, during the $(t+1)$-th global epoch, the server updates $w_g^t$ by gradient descent via aggregated gradients from activated client set $\mathcal{S}$: $w_g^{t+1} = w_g^t - \eta_g \sum_{i \in \mathcal{S}} \frac{n_i}{n} \nabla F_i(w_i^t)$, where $\nabla F_i(w_i^t) \triangleq \sum_{e=1}^{E} \nabla F_i(w_{i,e-1}^t; \mathcal{D}_i)$ and $e$ represents local training epochs. For model aggregation, the server averages local parameters directly: $w_g^{t+1} = \sum_{i \in \mathcal{S}} \frac{n_i}{n} w_i^t$. However, in SAFL, staleness arises in the local updates as client $C_i$ may use an outdated global model $w_g^{\tau_i^t}(\tau_i^t \leq t)$ for local training, leading to divergent optimization trajectories.

**Key Observations.** Inspired by empirical results [21], we identify two factors that cause performance gaps in SAFL: *Stale Updates* (Factor 1) and *Data Heterogeneity* (Factor 2). Staleness affects the continuity of global optimization trajectories differently in gradient and model aggregation. Gradient aggregation preserves continuity by performing gradient descent on the latest global model $w_g^{t-1}$, where stale gradients only influence the current update direction and magnitude. In contrast, model aggregation averages stale parameters directly, resetting the trajectory and disrupting the optimization continuity on

Table 1: The average best accuracy and corresponding differences between two aggregation strategies under varying influencing factors.

| Activated Factors | | Average Best Acc. (%) | | Gap (%) |
|---|---|---|---|---|
| Factor 1 | Factor 2 | Gradient Aggregation | Model Aggregation | |
| ○ | ○ | 90.93 | 91.05 | 0.12 |
| ● | ○ | 90.73 | 90.51 | 0.22 |
| ○ | ● | 86.79 | 87.29 | 0.50 |
| ● | ● | 82.63 | 71.11 | **11.52** |

* All experiments involve training ResNet-18 on CIFAR-10 across three independent runs. ○ indicates the absence of a factor, while ● indicates its presence.

the loss landscape. However, under Independent and Identically Distributed (IID) settings, stale updates exhibit limited divergence as all clients optimize identical local objectives derived from the identical data distributions. Conversely, non-IID data distributions exacerbate this issue, since increased local training rounds amplify local-global deviation [33] and semi-asynchronous updates bias the server toward frequent updaters [15] in such distributions. In gradient aggregation, this bias leads to over-optimization on dominant clients' data distribution; in model aggregation, the global model retains more information from frequent clients, restarting optimization from a skewed initial point. These dynamics exacerbate the performance gap between the two strategies in SAFL.

**Empirical Validation.** We conducted experiments training ResNet-18 on CIFAR-10 with 100 clients. The results show that when both factors are active (i.e., SAFL + non-IID), the accuracy gap between gradient aggregation and model aggregation surges to 11.52%. Table 1 shows detailed results.

This paper proposes FedQS, a novel framework that enables clients to select optimal local training modes autonomously by quantifying staleness (by update speed) and heterogeneity (by update similarity), compatible with two aggregation strategies.

## 3 Design of FedQS

### 3.1 System overview

As shown in Figure 2, FedQS[1] consists of three modules: the global aggregation estimation module (Mod①), the local training adaptation module (Mod②), and the global model aggregation module (Mod③). The first two modules are deployed on clients, while Mod③ is deployed on a centralized server. The goal of Mod① is to empower each participant to ascertain the approximate gradient update of the global model, which is utilized to compute the gradient update similarity between the server and the client. Since Mod① operates independently of the transmitted data required for global model updates, it facilitates the integration of various aggregation strategies. The goal of Mod② is to adapt each client to different training mechanisms based on its data distribution and available training resources. This module alleviates the impact of local heterogeneity on the global model, addressing issues such as unstable convergence and sluggish convergence speed. Meanwhile, the goal of Mod③ is to selectively weigh local updates for global aggregation through a feedback mechanism, thereby tackling the challenge of low accuracy under the model aggregation strategy.

---

[1]QS denotes Quadrant Selection. We also use the meaning of QS rankings to refer to the self-evaluation mechanism in FedQS.

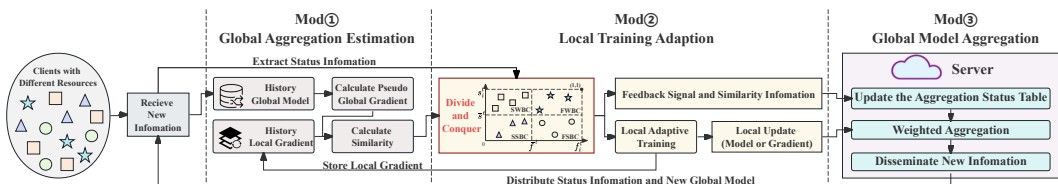

Figure 2: Workflow of FedQS, featuring clients with diverse resource capabilities. In FedQS, during a global training round, Mod① first utilizes the global model distributed by Mod③ to compute pseudo-global gradients and sends them to Mod②. Then, Mod② employs the information disseminated by Mod③ as input when determining a local training strategy and leverages the global model from Mod③ as the starting point for local training. Finally, Mod③ uses the local update data from Mod② for global aggregation and leverages the similarity information from Mod② to update the global state table.

## 3.2  Global aggregation estimation (Mod①)

Existing SAFL solutions typically leverage information (e.g., from historical local gradients [15]) uploaded by clients to facilitate global model aggregation. A key problem within these solutions is that they are proposed from the server's perspective, resulting in a design tightly coupled to a specific aggregation strategy. To address this problem, we introduce Mod① within FedQS to ensure compatibility with both aggregation strategies. Specifically, Mod① enables each participant to acquire global aggregation information during the local training phase from the perspective of local clients. To do it, Mod① first stores the latest two global models locally and then adopts the existing approach [16, 20] to derive a pseudo-global gradient $L_g(w_g^t)$ by comparing two consecutive global models, i.e., $L_g(w_g^t) = w_g^t - w_g^{t-1}$.

This pseudo-global gradient contains information about the global update within the current round. We then compute the local-global gradient update similarity $s_i^t$ on each client by comparing the latest local update gradient and the derived pseudo-global gradient, utilizing a similarity function such as cosine similarity. A larger $s_i^t$ suggests that this client's updates in the current round are more aligned with the global update and vice versa.

## 3.3  Local training adaptation (Mod②)

Mod② aims to orchestrate heterogeneous clients and enable each client to adaptively train its local model, facilitating the optimization of global model aggregation in Mod③. A key strength of Mod② lies in its ability to enable clients to dynamically adjust their local training strategies. There are two key benefits associated with it: (1) During the initialization phase, the server does not require prior knowledge (e.g., performance distribution) of clients. This contrasts with existing algorithms (e.g., FedAT [18] and FedMDS [19]), which rely on such prior knowledge for hierarchical client classification. (2) Given that the local information is processed and shared with the server in real-time, Mod② can handle dynamic FL environments where the performance (e.g., available resources) of individual clients may vary during training.

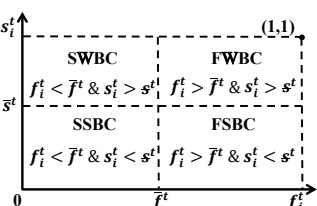

Figure 3: Categorization of clients in Mod②.

Specifically, Mod② adopts a divide-and-conquer strategy to categorize clients into four types, corresponding to the four quadrants in Figure 3. Mod② leverages the local update speed $f_i^t$ and the local-global gradient update similarity $s_i^t$ for categorization. Besides the aforementioned two indicators, the server also calculates the average value of all clients' local speeds ($\bar{f}^t$) and gradient update similarities ($\bar{s}^t$), respectively, which serve as the dividers. Equations 1 and 2 demonstrate the calculation formulas for these indicators.

$$n(i) = \begin{cases} n(i), & \text{if } i \notin \mathcal{S} \\ n(i) + 1, & \text{if } i \in \mathcal{S} \end{cases}, s_g(i) = \begin{cases} s_g(i), & \text{if } i \notin \mathcal{S} \\ s_i^t, & \text{if } i \in \mathcal{S} \end{cases}, \tag{1}$$

$$f_i^t = \frac{n(i)}{\sum_{i=1}^N n(i)}, \quad \bar{f}^t = \frac{\sum_{i=1}^N f_i^t}{N}, \quad \bar{s}^t = \frac{\sum_{i=1}^N s_g(i)}{N}, \tag{2}$$

where $n(i)$ is the total number of times client $C_i$ participates in the global model aggregation, $s_g(i)$ is the latest local-global gradient update similarity that $C_i$ shares with the server, and $N$ is the total number of clients.

Below, we explain the training strategy employed for each of the four types of clients.

**Fast-but-Strongly-Biased Clients (FSBC):** This type of client exhibits a rapid update speed (i.e., $f_i^t > \bar{f}^t$) but produces biased local gradient updates that deviate from the global model (i.e., $s_i^t < \bar{s}^t$). We attribute this phenomenon to the heterogeneous local data distribution, whose features the server may not fully extract. Due to these biased updates, reducing the local learning rates could hinder the global model's ability to learn sparse features from local data. Therefore, Mod② maintains the local learning rates of these clients unchanged. To facilitate the learning of the missing information from local data, Mod② instructs the server to assign appropriate (typically higher) aggregation weights to these clients through a feedback mechanism (see Mod③ in Section 3.4).

**Fast-and-Weakly-Biased Clients (FWBC):** For such clients, the server has effectively extracted their local information for global model aggregation. However, due to their fast local update speed (i.e., $f_i^t > \bar{f}^t$), the global aggregation may become biased towards their local models. To mitigate this effect, Mod② reduces the learning rate of these clients (i.e., $\eta_i^t = \eta_i^{t-1} - a\mathcal{F}$, where $a$ is the change rate and $\mathcal{F} = \frac{\bar{f}^t}{f_i^t}$) to slow down their update speed. A potential side effect of this operation is that it may decrease the convergence speed of local model training. To address this issue, we introduce a momentum term during the local training phase, as shown in Equation 3.

$$w_{i,e}^t = w_{i,e-1}^t - \eta_i^t [\overbrace{\sum_{r=1}^{e}(m_i^t)^r \nabla F_{i,e-r}(w_{i,e-r-1}^t)}^{\text{Momentum Term}} + \nabla F_{i,e}(w_{i,e-1}^t)], \tag{3}$$

where $m_i^t$ is the momentum rate and has the relationship: $m_i^t = m_0 + k(\frac{1}{\mathcal{G}} - 1)$, where $m_0, k$ are hyperparameters, and $\mathcal{G} = \frac{\bar{s}^t}{s_i^t}$.

**Straggling-but-Weakly-Biased Clients (SWBC):** Despite effectively utilizing local information for global model aggregation (i.e., $s_i^t > \bar{s}^t$), this type of client experiences a slow update speed (i.e., $f_i^t < \bar{f}^t$), suggesting limited local computational resources. Therefore, Mod② increases its learning rate (i.e., $\eta_i^t = \eta_i^{t-1} + a\mathcal{F}$) to compensate for these resource constraints. However, an excessive increase in the learning rate may hinder the optimization of the local model. To address this issue, Mod② employs the same momentum term used for FWBC to expedite the extraction of local data information and facilitate model convergence.

**Straggling-and-Strongly-Biased Clients (SSBC):** To address the problem posed by these clients' limited computational resources (i.e., $f_i^t < \bar{f}^t$), Mod② increases the local learning rate (i.e., $\eta_i^t = \eta_i^{t-1} + a\mathcal{F}$). To tackle the issue of local update bias diverging from the global model update direction (i.e., $s_i^t < \bar{s}^t$), we leverage a local validation set on each of these clients to assess the running environment. If the global model performs similarly on each label of the local validation dataset, we consider the issue to be a straggling problem (*Situation 1*) and adopt a strategy akin to that used for SWBC, enabling momentum optimization algorithms to mitigate the impact of outdated models. Conversely, if the global model exhibits significant performance differences across labels in the validation dataset, we consider the issue to be a dispersed distribution problem (*Situation 2*) and utilize a feedback mechanism similar to that employed for FSBC to alleviate the effects of heterogeneous data distribution.

Notably, the momentum term in FedQS is not primarily a speed accelerator but a trajectory stabilizer for clients whose updates align well with the global model (high $s_i^t$). For these clients, momentum mitigates oscillations while accelerating convergence speed (Equation 3). In contrast, for high-bias clients (FSBC and SSBC in *Situation 2*), premature momentum application could amplify the divergence between their local updates and the global one, as their updates are not yet globally beneficial.

### 3.4 Global model aggregation (Mod③)

This module aims to facilitate the central server in weighing the local models for the aggregation of the global model. Specifically, upon receiving new local update data from client $C_i$, the server

first calculates the average speed $\bar{f}^t$, average similarity $\bar{s}^t$, and the local update speed $f_i^t$, and then updates the aggregation status table accordingly using Equations 1 and 2. Then, the server will persistently wait and start aggregation once it receives $K$ available local updates. For each client that has uploaded its local update data, the server assigns an initial weight parameter $p_i = \frac{n_i}{n}$ and then iterates through these clients. If a client has triggered the feedback mechanism (SSBC with *Situation 2* or FSBC), the server updates its weight parameter via $p_i = \frac{\exp(\phi - \mathcal{F})}{2^{\phi - \mathcal{F}}} \cdot \frac{(1 + \mathcal{G})^2}{K}$, where $\exp(\cdot)$ is the natural exponential function and $\phi = \frac{K}{N}$; otherwise, the weights remain unchanged. Next, the server normalizes these weight parameters (i.e., $p_i = \frac{p_i}{\sum_{i \in \mathcal{S}} p_i}$, $\forall i \in \mathcal{S}$) before proceeding with the global aggregation process. The term $\frac{\exp(\phi - \mathcal{F})}{2^{\phi - \mathcal{F}}}$ addresses the effect of outdated weights (inspired by [34, 15]), while $\frac{(1 + \mathcal{G})^2}{K}$ accounts for the quadratic relationship between the convergence bound and the model weight difference, as outlined in Theorem 4.2 and Theorem 4.3.

Given that FedQS optimizes both aggregation strategies, we denote the gradient aggregation strategy-based implementation as FedQS-SGD and the model aggregation-based as FedQS-Avg in this paper. For FedQS-SGD, Mod③ incorporates the momentum term as part of the update data and calculates the pseudo-gradient [16, 20] as $\Delta F_{i,e}^{\tau_i^t} = \sum_{r=1}^{e} (m_i^{\tau_i^t})^r \nabla F_{i,e-r}(w_{i,e-r-1}^{\tau_i^t}) + \nabla F_{i,e}(w_{i,e-1}^{\tau_i^t})$. Then, Mod③ aggregates the new global model using $w_g^t = w_g^{t-1} - \sum_{i \in \mathcal{S}} p_i \eta_i^{\tau_i^t} \sum_{e=1}^{E} \Delta F_{i,e}^{\tau_i^t}$. For FedQS-Avg, Mod③ performs the weighted aggregation by $w_g^t = \sum_{i \in \mathcal{S}} p_i w_i^{\tau_i^t}$.

# 4 Convergence analysis of FedQS

In this section, we present the theoretical convergence guarantees for FedQS, demonstrating its ability to effectively optimize both gradient and model aggregation strategies within the SAFL framework. Complete theoretical details are provided in Appendix A due to space constraints.

## 4.1 Assumptions

We begin by stating our key assumptions, which are standard in the federated learning literature. The following conditions hold:

(1) For $\forall i$, the loss function $F_i$ is $L$-smooth [35, 36].
(2) The expected squared norm of local stochastic gradients $\nabla F_i(w_i^t)$ is uniformly bounded by $G_c$ [35, 16].
(3) The degree of heterogeneity in the training task is bounded by $\delta$ [3, 15].

*Remark* 4.1. Assumption 4.1(2) is introduced *solely* to simplify the interpretation of convergence bounds by providing a deterministic upper bound for the gradient variation term $\mathcal{W}$ (see Theorems A.4 and A.5 in the appendix). Crucially, it is not used in proving our core convergence theorems (Theorems 4.2 and 4.3), which rely only on Assumptions 4.1(1) and (3). The convergence guarantees and heterogeneity analysis in Section 4.2 remain fully valid without this condition.

## 4.2 Convergence guarantees

Our convergence analysis employs the expected function value gap, $\mathbb{E}[F(w_g^t)] - F^*$, as the primary performance metric, following established SAFL literature [18, 28, 37]. This choice is motivated by its direct quantification of solution quality relative to the ideal optimum $F^*$, its unique ability to reveal stability dynamics (e.g., oscillations and transient regressions under semi-asynchronous updates that gradient norms often obscure), and its alignment with theoretical and practical SAFL conventions for assessing convergence behavior and aggregation-strategy efficacy, thereby providing a more comprehensive evaluation of FedQS's performance under dual aggregation strategies. We present our main convergence results below.

**Theorem 4.2** (Gradient Aggregation Convergence). *Under Assumptions 4.1, let $\beta = \max_{i,t}\{\eta_i^t, \eta_g\}$ with $\sqrt{\frac{1}{RK-1}} < \beta < \sqrt{\frac{3}{2RK-3}}$, where $R = \frac{E\theta - E\theta^2 - \theta^2 + \theta^{E+2}}{(1-\theta)^2}$. In R's formula, E is the maximum local epoch, and $\theta = \max_{i,t}\{m_i^t\}$. Let K be defined in Section 3.4, then FedQS-SGD satisfies:*

$$\mathbb{E}[F(w_g^t)] - F^* \leq L\mathcal{V}^t \mathbb{E}[||w_g^0 - w^*||^2] + \mathcal{U} + \mathcal{W}, \tag{4}$$

where $\mathcal{V} = (3 - \frac{2\beta^2 KR}{\beta^2+1}) \in (0,1)$ *controls the convergence rate,* $\mathcal{U} = \mathcal{O}(\delta^2)$ *captures the data heterogeneity, and* $\mathcal{W} = \mathcal{O}(G_c^2)$ *bounds gradient variations. Notice that* $\mathcal{W} \leq [4LE^2 + 4LRQ(t) + \frac{(\beta^2 L+L)(2RQ(t)+3E^2)}{2\beta^2 R - 2\beta^2 - 2}]\beta^2 G_c^2$, *where* $Q(t)$ *denotes the maximum number of clients that execute the momentum update at global round* $t$.

**Theorem 4.3** (Model Aggregation Convergence). *Under Assumptions 4.1, let* $0 \leq q < p_i < p \leq 1$, *for* $\sqrt{\frac{1}{KR+E^2-1}} < \beta < \sqrt{\frac{3}{2RK+2E^2-3}}$. *Let K be defined in Section 3.4, then FedQS-Avg satisfies:*

$$\mathbb{E}[F(w_g^t)] - F^* \leq (3LpK^2 + L)\mathcal{V}^t \mathbb{E}[||w_g^0 - w^*||^2] + \mathcal{U} + \mathcal{W}, \tag{5}$$

*with* $\mathcal{V}, \mathcal{U}, \mathcal{W}$ *having similar interpretations as in Theorem 4.2.*

*Remark* 4.4. The exponentially decaying $\mathcal{V}^t$ term accelerates convergence within the sublinear regime characteristic of non-convex FL, yielding a rate strictly faster than the standard $O(1/t)$ while maintaining the overall $O(1/t + \mathcal{U} + \mathcal{W})$ bound. Unlike the linear convergence under strong convexity/Polyak-Lojasiewicz conditions, this confirms that $\mathcal{V}^t$ enhances practical performance without altering the fundamental sublinear convergence landscape.

*Remark* 4.5. The $\mathcal{U}$ term highlights the impact of data heterogeneity ($\delta^2$), while $\mathcal{W}$ captures gradient norm effects ($G_c^2$). The non-vanishing terms $\mathcal{U} + \mathcal{W}$ reflect fundamental limitations inherent to semi-asynchronous FL systems [28, 37], capturing unavoidable convergence errors from non-simultaneous aggregation and irreducible data heterogeneity.

*Remark* 4.6. Theorem 4.2 demonstrates that FedQS-SGD limits the gradient variation amplification in $\mathcal{W}$ through controlled momentum updates since the momentum term is only applied to a subset of clients (i.e., FWBC, SWBC, and SSBC with *Situation 1*), thereby improving convergence stability.

*Remark* 4.7. Theorem 4.3 shows that FedQS-Avg achieves comparable convergence with FedQS-SGD, with the $\mathcal{V}^t$ term guaranteeing asymptotic convergence, alleviating the slow convergence speed and suboptimal convergence utility inherent to model aggregation strategies in SAFL.

## 5 Evaluation

### 5.1 Experimental setup

We evaluate FedQS on three task types in SAFL: Computer Vision (CV) using ResNet-18 [38] on CIFAR-10 [39], Natural Language Processing (NLP) with LSTM [40] on Shakespeare [1], and Real-World Data (RWD) using FCN on UCI Adult [41]. Resource constraints limited our experiments to moderate-scale models, though our approach remains model-agnostic as FL is an infrastructure independent of models or datasets. We simulate a heterogeneous federated system with clients having uniformly distributed computing resources (default: 100 clients, resource ratio 1:50, i.e., the fastest client exhibiting a training speed 50 times that of the slowest). The default similarity function in Mod① is cosine similarity.

All experiments were conducted on a Linux system (Ubuntu 22.04 LTS) using an Intel Xeon Platinum 8468 Processor and an NVIDIA H100 80GB HBM3 GPU, with system memory capped at 20 GB per run. The software stack included Python 3.8.0, PyTorch 2.1.0, and Torchvision 0.16.0, all within a Conda environment. Due to space constraints, comprehensive details on datasets, model architectures, client resource distributions, hyperparameters, baseline algorithms, and additional results are provided in Appendix D.

### 5.2 Performance evaluation

As the first solution optimizing both gradient and model aggregation strategies in SAFL, we compare FedQS against four model aggregation (FedAvg [1], SAFA [31], FedAT [18], M-step [37]) and four gradient aggregation (FedSGD [1], FedBuff [16], WKAFL [15], FedAC [20], DeFedAvg [42], FADAS [43], $CA^2$FL [44]) baselines. To our best, there is no related work that supports both strategies. We employ the same metrics as those in [21] to evaluate the performance of FedQS and baselines: 1) Accuracy and loss performance, which reflect the prediction capabilities of the trained global model on the test dataset. 2) Convergence speed, determined by the number of epochs (denoted as $T_f$) required to achieve the target accuracy for the first time [21]. 3) Runtime, recorded as the duration from the initiation to the completion of the $T$-th ($T$ being the maximum global training epoch) rounds of global aggregation.

Table 2: Accuracy and convergence speed of FedQS and the baselines.

| Metrics | Tasks | Algorithms | | | | | | | | | | | | |
|---|---|---|---|---|---|---|---|---|---|---|---|---|---|---|
| | | FedAvg | SAFA | FedAT | M-step | FedQS (Avg) | FedSGD | FedBuff | WKAFL | FedAC | DeFedAvg | FADAS | $CA^2$FL | FedQS (SGD) |
| Accuracy (%) | $x = 0.1$ | 56.05 | 56.15 | 28.15 | 62.17 | **63.91** | 65.71 | 64.43 | 64.66 | 56.52 | 52.33 | 65.34 | 42.29 | **68.88** |
| | $x = 0.5$ | 73.71 | 58.31 | 45.65 | **80.49** | 80.26 | 83.87 | 80.73 | 85.14 | 82.65 | 83.51 | 82.2 | 63.79 | **86.11** |
| | $x = 1$ | 77.86 | 62.16 | 47.58 | 82.46 | **82.74** | 85.42 | 81.43 | 86.02 | 85.94 | 86.17 | 84.21 | 70.16 | **86.79** |
| | $R = 200$ | 47.04 | 43.65 | 37.13 | 49.38 | **50.43** | 48.04 | 48.37 | 50.49 | 50.43 | 40.19 | 46.83 | 25.36 | **52.22** |
| | $R = 600$ | 45.52 | 40.90 | 36.35 | 48.12 | **50.08** | 49.64 | 47.42 | 50.09 | 51.62 | 48.23 | 48.95 | 28.86 | **52.49** |
| | Gender | 77.10 | 77.05 | 77.01 | 78.20 | **78.94** | 77.15 | 76.37 | **78.96** | 77.69 | 77.82 | 78.04 | 76.09 | 78.74 |
| | Ethnicity | 77.25 | 77.07 | 77.26 | 78.01 | **78.85** | 78.33 | 78.71 | 76.97 | 77.71 | 78.11 | 78.54 | 66.68 | **79.24** |
| Conv. speed (# epochs) | $x = 0.1$ | 304 | 362 | 317 | 329 | **276** | 281 | 334 | 277 | 243 | 307 | 255 | 272 | **239** |
| | $x = 0.5$ | 295 | 344 | 304 | 276 | **234** | 264 | 293 | 255 | 232 | 257 | 220 | 256 | **213** |
| | $x = 1$ | 154 | 272 | 244 | 163 | **119** | 144 | 259 | 257 | 136 | 155 | 143 | 200 | **127** |
| | $R = 200$ | 288 | 342 | 357 | 264 | **231** | 234 | 278 | 221 | 243 | 256 | 220 | 259 | **188** |
| | $R = 600$ | 293 | 336 | 375 | 275 | **249** | 251 | 303 | 248 | 277 | 238 | 254 | 272 | **216** |
| | Gender | 43 | 57 | 63 | 46 | **35** | 29 | 66 | 51 | **17** | 59 | 25 | 59 | 18 |
| | Ethnicity | 55 | 67 | 61 | 44 | **33** | 36 | 76 | 54 | 27 | 20 | 24 | 325 | **22** |

* In the "Tasks" column, $x$ represents the parameter of the Dirichlet distribution within CV tasks; $R$ denotes the number of roles within NLP tasks; Gender and Ethnicity are the data types within RWD tasks.
* The convergence accuracy is measured as the average global accuracy over the last 20 rounds. The target accuracy for convergence speed is set to 95% of convergence accuracy in CV and NLP tasks and 98% of the convergence accuracy in RWD tasks.

Table 3: Runtime of FedQS and the baselines.

| Tasks | Algorithms | | | | | | | | | | | | | | |
|---|---|---|---|---|---|---|---|---|---|---|---|---|---|---|---|
| | FedAvg (SFL) | FedAvg | SAFA | FedAT | M-step | FedQS (Avg) | FedSGD (SFL) | FedSGD | FedBuff | WKAFL | FedAC | DeFedAvg | FADAS | $CA^2$FL | FedQS (SGD) |
| $x = 0.1$ | 78,048 | 24,204 | 204,588 | 66,851 | 49,971 | 32,827 | 76,592 | 24,184 | 25,654 | 34,862 | 30,329 | 30,776 | 32,380 | 108,097 | 32,784 |
| $x = 0.5$ | 77,956 | 24,214 | 144,371 | 66,746 | 49,953 | 33,254 | 76,822 | 23,921 | 25,720 | 35,286 | 28,721 | 31,754 | 32,966 | 109,635 | 33,656 |
| $x = 1$ | 79,095 | 24,208 | 142,594 | 66,851 | 46,859 | 33,477 | 79,816 | 24,108 | 25,513 | 32,707 | 29,152 | 30,280 | 31,471 | 108,756 | 33,400 |
| $R = 200$ | 22,417 | 4,925 | 27,642 | 21,089 | 116,906 | 6,023 | 22,118 | 4,669 | 5,096 | 32,849 | 8,340 | 4,989 | 5,118 | 36,441 | 5,248 |
| $R = 600$ | 53,784 | 7,358 | 39,475 | 33,300 | 171,965 | 9,528 | 53,590 | 7,211 | 8,320 | 59,277 | 12,982 | 8,720 | 9,002 | 36,548 | 9,135 |
| Gender | 30,149 | 5,508 | 10,413 | 6,338 | 12,469 | 5,701 | 30,088 | 5,385 | 6,164 | 13,420 | 5,423 | 4,841 | 5,476 | 19,650 | 5,523 |
| Ethnicity | 33,275 | 5,513 | 9,127 | 6,437 | 12,389 | 5,665 | 32,787 | 5,148 | 6,037 | 14,215 | 5,711 | 4,827 | 5,299 | 19,370 | 5,465 |

* Shadowed columns represent evaluations under synchronous FL (SFL).

Table 2 presents the experimental results. FedQS attains the overall highest accuracy and fastest convergence speed across almost all tasks. Specifically, for model aggregation, FedQS's average accuracy is 68.88%, 1.71% higher than that of the best baseline (i.e., M-step), and for gradient aggregation it is 72.06%, 2.51% higher than the best baseline (i.e., WKAFL). SAFA, FedAT, and FedBuff do not perform as well as their corresponding foundational algorithms, FedAvg and FedSGD. This is because these three algorithms sacrifice the accuracy of the global model for a more stable training process (see Table 9 and Figure 10 in the appendix for the analysis of convergence stability). Figure 4 depicts the loss function curves[2] of representative tasks for both model and gradient aggregations. This figure illustrates that FedQS achieves the minimum loss as training progresses, implying FedQS's capability to converge the model to its optimum.

Table 3 shows the runtime performance results. FedQS achieves comparable efficiency to the top-performing baselines, demonstrating its feasibility for real-world deployment. For reference, we include the runtime measurements of FedAvg and FedSGD under synchronous settings. When compared to these synchronous baselines, FedQS-Avg and FedQS-SGD achieve average runtime reductions of 70.34% and 70.91%, respectively, highlighting SAFL's superior resource efficiency over synchronous configurations. Meanwhile, as evidenced by Figures 8 and 9 (in the appendix), FedQS maintains substantially lower time latency between global aggregation rounds than other SAFL optimization algorithms, further validating its operational efficiency.

## 5.3 Effectiveness of FedQS under different system settings

In this section, we evaluate the impact of different numbers of clients and resource distributions on the performance of FedQS. We vary the number of clients (50 or 200) and the resource distribution ratio (1:20 or 1:100), conducting experiments with $x = 0.1, 0.5, 1$ in CV tasks under both aggregation strategies and reporting average values. Table 4 shows representative results of FedQS compared to two foundational algorithms in SAFL (with complete results in Table 10 in the appendix). FedQS

---

[2]WKAFL's overfitting curve in Figure 4f is due to the large learning rate and WKAFL's adaptive learning rate adjustment module.

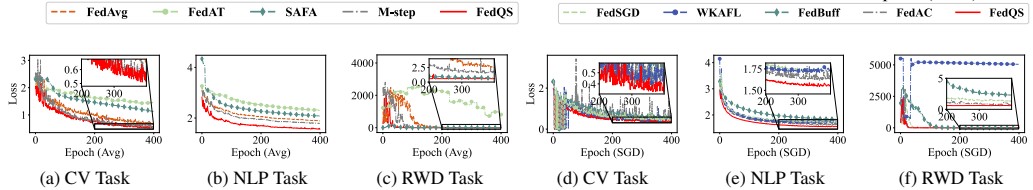

Figure 4: Loss of FedQS and the baselines under a representative CV task ($x = 0.5$), an NLP task ($R = 600$), and an RWD task (gender). The left three subfigures are based on model aggregation, while the right three are based on gradient aggregation.

Table 4: Robustness of FedQS.

| Scenario | Method | Metrics | | |
|---|---|---|---|---|
| | | Accuracy (%) | Conv. speed | # Oscillations |
| N=50 (1:20) | FedAvg | 70.1 | 224 | 0.0 |
| | FedQS-Avg | **79.2** | **179** | 3.0 |
| | FedSGD | 77.4 | 193 | 37.0 |
| | FedQS-SGD | **80.7** | **152** | **26.3** |
| N=200 (1:100) | FedAvg | 49.4 | 277 | 0.0 |
| | FedQS-Avg | **64.7** | **256** | 0.3 |
| | FedSGD | 74.4 | 248 | 7.3 |
| | FedQS-SGD | **80.1** | **203** | **3.0** |

* $N = 50$ means the task has 50 clients, and 1:20 means the fastest client exhibits a training speed 20 times that of the slowest one. The threshold used to calculate the number of oscillations is set to 15. The target accuracy for convergence speed is set to 95% of convergence accuracy

* Each result is an average value of three experiments corresponding to $x = 0.1, 0.5, 1$ in CV tasks.

Table 5: Ablation study results in CV tasks.

| Module | Method | Metrics | | | | | |
|---|---|---|---|---|---|---|---|
| | | Accuracy (%) | | Conv. speed (# epochs) | | # Oscillations | |
| | | Avg | SGD | Avg | SGD | Avg | SGD |
| Mod① | Cosine | 74.14 | **80.59** | 251 | 230 | **4.0** | **7.6** |
| | Euclidean | 75.69 | 79.55 | 244 | 232 | 5.6 | 11.3 |
| | Manhattan | **76.56** | 80.28 | **228** | **221** | 4.6 | 9.6 |
| Mod② | w/o momentum | 73.21 | 78.88 | 269 | 242 | 4.3 | 9.6 |
| | with momentum | **74.14** | **80.59** | 251 | 230 | **4.0** | **7.6** |
| Mod③ | w/o feedback | 68.35 | 78.83 | 284 | 268 | **0.0** | 5.6 |
| | with feedback | **74.14** | **80.59** | 251 | 230 | 4.0 | 7.6 |

* Avg and SGD represent FedQS-Avg and FedQS-SGD, respectively.
* Each result is an average value corresponding to $x = 0.1, 0.5, 1$. The threshold used to calculate the number of oscillations is set to 15. The target accuracy for convergence speed is set to 95% of convergence accuracy

consistently outperforms both FedAvg and FedSGD across all three metrics, aligning with the results in Table 2.

To further validate the robustness of FedQS in dynamic environments, we conduct additional CV experiments under three scenarios where client resources vary during training: (1) Dynamic Resource Scale (Scenario 1): the speed ratio shifts from 1:50 to 1:100 at round 200; (2) Unstable Resource per Client (Scenario 2): each client's resource fluctuates within $[-10, +10]$ unit times per update, bounded between 1 and 50 units; (3) Client Dropout (Scenario 3): 50% of clients randomly churn at round 100. As shown in Table 6, FedQS maintains stable convergence across all dynamic scenarios.

These results demonstrate that FedQS is effective under various system settings and maintains robustness in dynamic environments. We further discuss the scalability of FedQS in Appendix C.

Table 6: Performance comparison of different algorithms under dynamic scenarios

| Dynamic Scenario | Algorithm | x=0.1 | | | x=0.5 | | | x=1 | | |
|---|---|---|---|---|---|---|---|---|---|---|
| | | Accuracy (%) | Conv. Speed | Runtime (s) | Accuracy (%) | Conv. Speed | Runtime (s) | Accuracy (%) | Conv. Speed | Runtime (s) |
| Scenario 1 | FedAvg | 52.77 | 322 | 46488 | 77.29 | 288 | 47732 | 76.79 | 227 | 48003 |
| | FedQS-Avg | **66.47** | **264** | 53986 | **82.90** | **235** | 54575 | **83.22** | **201** | 55529 |
| | FedSGD | 68.29 | 244 | 47443 | 85.66 | 211 | 47728 | 87.86 | 147 | 48280 |
| | FedQS-SGD | **76.17** | **206** | 54095 | **88.12** | **177** | 55268 | **88.80** | **115** | 55257 |
| Scenario 2 | FedAvg | 53.12 | 310 | 47397 | 76.74 | 294 | 47156 | 79.90 | 286 | 47788 |
| | FedQS-Avg | **64.52** | **268** | 55535 | **83.53** | **241** | 55816 | **84.35** | **252** | 56253 |
| | FedSGD | 65.13 | 268 | 47613 | 85.37 | 207 | 47096 | 88.63 | 159 | 47634 |
| | FedQS-SGD | **69.65** | **223** | 55037 | **87.96** | **185** | 55201 | **89.14** | **133** | 55808 |
| Scenario 3 | FedAvg | 44.39 | 343 | 58993 | 68.44 | 332 | 59362 | 72.47 | 296 | 59447 |
| | FedQS-Avg | **53.22** | **297** | 62623 | **79.33** | **272** | 63308 | **81.08** | **253** | 63776 |
| | FedSGD | 58.40 | 292 | 58345 | 81.73 | 288 | 59487 | 84.43 | 262 | 59732 |
| | FedQS-SGD | **60.08** | **263** | 62870 | **83.39** | **245** | 63992 | **86.18** | **241** | 64703 |

## 5.4 Hyperparameter analysis

This section presents a hyperparameter analysis of FedQS. Using a grid search, we fine-tune four key hyperparameters within the adaptive module (Mod②): the initial learning rate $\eta_0$ (where $\eta_0 \triangleq \eta_i^0, \forall i$), the learning rate change rate $a$, the initial momentum $m_0$, and the momentum change speed $k$.

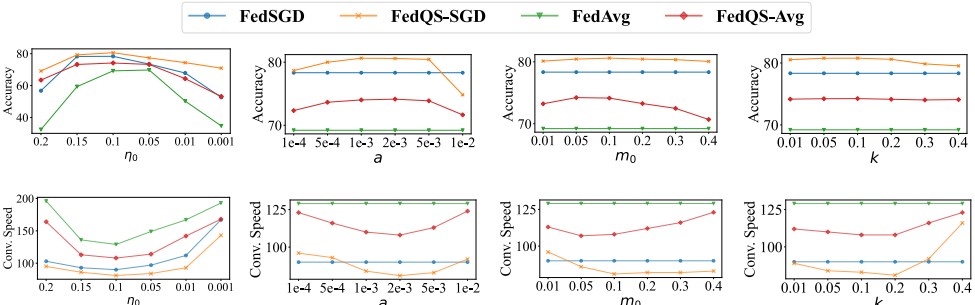

Figure 5: Impact of different hyperparameters on FedSGD, FedAvg, and FedQS's performance in CV tasks. The target accuracy of convergence speed is set to 80% of convergence accuracy.

Figure 5 summarizes the results for CV tasks (with detailed results provided in Appendix Tables 11-14). Our findings indicate that an excessively large or small $\eta_0$ leads to reduced accuracy and slower convergence, a trend also observed in FedSGD and FedAvg. However, FedQS's adaptive learning strategy yields significant improvements in both metrics over these baselines. Furthermore, we observe that overly large values of $a$, $m_0$, and $k$ can adversely affect FedQS's performance in both modes. Among these, $a$ has the most pronounced impact on accuracy, while $k$ has the least.

Crucially, FedQS exhibits robust performance across broad hyperparameter ranges (e.g., $a \in [0.001, 0.005]$ and $k \in [0.001, 0.2]$), achieving strong results with default settings ($a = 0.002$, $m_0 = 0.1$, $k = 0.2$) as shown in our main experiments. The adaptive mechanisms in Mod② inherently mitigate sensitivity to hyperparameter selection, while Mod① and Mod③ remain hyperparameter-free. This design substantially reduces tuning overhead and facilitates practical deployment in real-world federated learning scenarios.

## 5.5 Ablation studies

In this section, we conduct ablation studies on FedQS's modules, with the results presented in Table 5.

**Similarity Function (Mod①).** We evaluate three similarity measures: Cosine Function (CF), Euclidean Distance (ED), and Manhattan Distance (MD). While MD achieves slightly higher accuracy, CF exhibits better stability with fewer oscillations. The comparable performance across measures validates FedQS's robustness to different similarity metrics.

**Momentum (Mod②).** Removing momentum terms degrades performance significantly, with average accuracy dropping by 4.3% and convergence requiring 6% more epochs. This demonstrates momentum's crucial role in both optimization efficiency and final model quality.

**Feedback Mechanism (Mod③).** Disabling the feedback mechanism causes substantial accuracy degradation (7.81% for FedQS-Avg, 2.18% for FedQS-SGD), highlighting its importance for maintaining model performance. The feedback mechanism proves particularly valuable for averaged models, where its absence leads to more pronounced performance drops.

## 6 Conclusions

In this paper, we presented FedQS, a novel framework that jointly optimizes gradient and model aggregation in SAFL. FedQS employs a divide-and-conquer strategy that dynamically categorizes clients into four types and adaptively adjusts their local training schemes. Theoretical analysis establishes that FedQS achieves convergence under both aggregation paradigms, effectively addressing the instability of gradient-based methods and the suboptimality of model-based ones. Experiments across diverse settings demonstrate that FedQS achieves accurate predictions, rapid convergence, and stable training, consistently outperforming state-of-the-art baselines. Future work will integrate differential privacy for stronger security and evaluate under more realistic conditions (e.g., non-IID data and limited client-side data volumes) to further validate practical utility.

## Acknowledgments and Disclosure of Funding

This work was partly supported by the National Key R&D Program of China (No. 2023YFB2704903), Research and Development Program of the Department of Industry and Information Technology in Xinjiang Autonomous District (No. SA0304173), the Natural Science Foundation of Shanghai (No. 23ZR1429600), and a Huawei research grant (No. TC20250717058).

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

# A    Convergence analysis details

We first provide the detailed descriptions of Assumption 4.1, which are also adopted in related works.

**Assumption A.1** (*L*-smooth [35, 36]). We assume that for $\forall i$, the loss function $F_i$ is $L$-smooth, i.e., for all $x, y$ as input, we have:

$$F_i(y) - F_i(x) \leq \langle \nabla F_i(x), (y-x) \rangle + \frac{L}{2} ||y - x||^2, \tag{6}$$

where $L > 0$ is a constant.

**Assumption A.2** (Bounded Gradient [35, 16]). We assume that the expected squared norm of local stochastic gradients $\nabla F_i(w_i^t)$ is uniformly bounded, i.e.,

$$\mathbb{E}[||\nabla F_i(w_i^t)||^2] \leq G_c^2, \tag{7}$$

where $G_c > 0$ is a constant.

**Assumption A.3** (Bounded Heterogeneous Degree [3, 15]). We assume that the degree of heterogeneity in the training task is finite, i.e.,

$$\mathbb{E}[||\nabla F_g(w_g^t) - \nabla F_i(w_i^t)||^2] \leq \delta^2, \tag{8}$$

where $\delta > 0$ is a constant and $\nabla F_g(\cdot)$ is the ideal global gradient, which is defined in detail in Appendix E. This also implies that the discrepancy between the gradients of each individual client and the ideal global gradient is bounded.

We then give the justifications of the reasonability for these assumptions.

**Justification of Assumption A.1.** Assumption A.1 is reasonable because loss functions (e.g., sigmoid) in neural networks are generally smooth. The L-smoothness assumption aligns with differentiable loss function requirements, enabling gradient computation via backpropagation. The convergence analyses of FL typically rely on this assumption to facilitate a simplified mathematical analysis. We avoided the convexity assumption in our analysis for a more general analysis, because convexity relates to the difficulty of optimization problems. While convexity properties hold in simpler models like logistic regression or SVMs, deep neural networks generally operate in non-convex loss landscapes.

**Justification of Assumption A.2.** Assumption A.2 is reasonable because, in SAFL, gradient clipping is employed to alleviate the accuracy degradation issue caused by non-IID data distribution. As long as clients apply gradient clipping, this assumption naturally stands. Violating this assumption implies that there exists a client whose gradient is unbounded, leading to the gradient explosion and potentially causing the training to fail completely. This assumption also has a wide application in FL [35, 45, 18, 16, 23]. Meanwhile, since gradient clipping is commonly adopted in SAFL [15], the local gradient magnitudes during training remain bounded by the clipping threshold. Consequently, the gradient clipping threshold can be directly utilized as the upper bound for gradients in our theoretical assumptions. In our experiments, we set the bound $G_c$ as 20.

**Justification of Assumption A.3.** Assumption A.3 holds since data distribution heterogeneity is constrained due to limited training data. The heterogeneity of data distribution is determined once participating clients are selected for aggregation. Violating this assumption indicates either an infinite amount of data, which is impractical, or the data distribution would be constantly changing, making it impossible to capture its exact values and ultimately leading to training failure.

Based on these assumptions, we rewrite Theorems 4.2 and 4.3 to give the complete theoretical results.

**Theorem A.4** (The convergence of FedQS-SGD). *Let Assumption A.1,A.2,A.3 hold and $L, \delta, G_c$ be defined therein. Let $\tau_i^t$ be defined Section 2 and K be defined in Section 3.4, $\beta = \max_{i,t}\{\eta_i^t, \eta_g\}$ and $\sqrt{\frac{1}{RK-1}} < \beta < \sqrt{\frac{3}{2RK-3}}$, where E is the maximum local epoch, $\theta = \max_{i,t}\{m_i^t\}$ and $R = \frac{E\theta - E\theta^2 - \theta^2 + \theta^{E+2}}{(1-\theta)^2}$. Then, we have the following convergence results:*

$$\mathbb{E}[F(w_g^t)] - F^* \leq L\mathcal{V}^t\mathbb{E}[||w_g^0 - w^*||^2] + \mathcal{U} + \mathcal{W}, \tag{9}$$

*where $\mathcal{U} = [2L\beta^2 + \frac{6\beta^2(\beta^2 L + L)}{2\beta^2 R - 2\beta^2 - 2}]E^2\delta^2, \mathcal{V} = (3 - \frac{2\beta^2 KR}{\beta^2 + 1})$, and*

$$\mathcal{W} = 4LE[||\eta_i^{\tau_i^t} \sum_{e=1}^{E} \nabla F_{i,e}(w_{i,e-1}^{\tau_i^t})||^2] + 4LE[||\eta_i^{\tau_i^t} \sum_{e=1}^{E} \sum_{r=1}^{e} (m_i^t)^r \nabla F_{i,e-r}(w_{i,e-r-1}^{\tau_i^t})||^2]$$

$$+ \sum_{j=0}^{t} \mathcal{V}^j 3LE[||\eta_i^{\tau_i^{t-1}} \sum_{e=1}^{E} \nabla F_{i,e}(w_{i,e-1}^{\tau_i^{t-1}})||^2] \tag{10}$$

$$+ \sum_{j=0}^{t} \mathcal{V}^j 2LE[||\eta_i^{\tau_i^{t-1}} \sum_{e=1}^{E} \sum_{r=1}^{e} (m_i^{t-1})^r \nabla F_{i,e-r}(w_{i,e-r-1}^{\tau_i^{t-1}})||^2]$$

$$\leq [4LE^2 + 4LRQ(t) + \frac{(\beta^2 L + L)(2RQ(t) + 3E^2)}{2\beta^2 R - 2\beta^2 - 2}]\beta^2 G_c^2.$$

*Notice that the value of $\mathcal{V}$ is in $(0,1)$ and $Q(t)$ denotes the maximum number of occasions upon which all participating nodes execute the momentum update module at global round $t$.*

**Proof.** See Appendix E.1. □

**Theorem A.5** (The convergence of FedQS-Avg). *Let Assumption A.1,A.2,A.3 hold and $L, \delta, G_c$ be defined therein. Let $\tau_i^t$ be defined Section 2 and $K$ be defined in Section 3.4. Denoted $E$ as the maximum local epoch and assume the aggregation weight parameter $p_i$ satisfies $0 \leq q < p_i < p \leq 1$. Let $\beta = \max_{i,t}\{\eta_i^t, \eta_g\}$ and $\sqrt{\frac{1}{KR+E^2-1}} < \beta < \sqrt{\frac{3}{2RK+2E^2-3}}$, where $R = \frac{E\theta - E\theta^2 - \theta^2 + \theta^{E+2}}{(1-\theta)^2}$. In R's formula, $\theta = \max_{i,t}\{m_i^t\}$. Then, we have the following convergence results:*

$$\mathbb{E}[F(w_g^t)] - F^* \leq (3LpK^2 + L)\mathcal{V}^t \mathbb{E}[||w_g^0 - w^*||^2] + \mathcal{U} + \mathcal{W}, \tag{11}$$

*where $Q(t)$ is defined in Theorem A.4, $\mathcal{V} = (3 - \frac{2\beta^2(R+E^2)}{\beta^2+1}) \in (0,1)$, $\mathcal{U} = [3p^2KL + \frac{8(3pK^2+1)(\beta^2L+L)}{2\beta^2(R+E^2)-2\beta^2-2}]\beta^2E^2\delta^2$, and*

$$\mathcal{W} = (3Lp^2K + L)\sum_{j=1}^{t} \mathcal{V}^j \mathbb{E}[||\eta_i^{t-1} \sum_{e=1}^{E} [\nabla F_i(w_{i,e}^{t-1})]||^2] + p^2KL(\mathbb{E}[||\eta_i \sum_{e=1}^{E} \nabla F_i(w_{i,e}^{\tau_i^t})||^2]$$

$$+ \mathbb{E}[||\eta_i^t \sum_{e=1}^{E} \nabla F_i(w_{i,e}^t)||^2]) + (3Lp^2K + L)\sum_{j=1}^{t} \mathcal{V}^j \mathbb{E}[||\eta_i^{t-1} \sum_{e=1}^{E} \sum_{r=1}^{e} (m_i^{t-1})^r \nabla F_i(w_{i,e-r}^{t-1})||^2])$$

$$+ 3p^2KLE[||\eta_i^t \sum_{e=1}^{E} \sum_{r=1}^{e} (m_i^t)^r \nabla F_i(w_{i,e-r}^t)||^2]$$

$$\leq [p^2KL(2E^2 + 3RQ(t)) + \frac{(3p^2K+1)(\beta^2L+L)(E^2+RQ(t))}{2\beta^2(R+E^2) - 2\beta^2 - 2}]\beta^2 G_c^2.$$

$$\tag{12}$$

**Proof.** See Appendix E.2. □

# B Different communication and aggregation strategies in FL

**Synchronous vs. Semi-Asynchronous FL.** The primary difference between Synchronous FL and Semi-Asynchronous FL (SAFL) is that the former is based on server-controlled coordinated training, while the latter involves fully autonomous client execution with conditional-triggered global aggregation. Specifically, in Synchronous FL, the server initiates each global round by selecting a subset of clients as activated clients. These activated clients are required to complete local training using the latest global model and submit their updates, while inactive clients remain idle until this round concludes. The server performs aggregation through designated aggregation strategies after

receiving all activated clients' updates, then broadcasts the modified global model. In contrast, SAFL eliminates centralized coordination, where the server passively awaits client updates and triggers global aggregation immediately upon meeting specific triggering conditions (e.g., sufficient accumulated updates [15, 16]). Each client autonomously executes local training at its own pace. Upon completing local training, clients immediately transmit local updates to the server and verify the availability of the new global model. If it is available, they synchronize their local models accordingly; otherwise, they persist with existing parameters while continuing local training.

**Gradient aggregation vs. Model aggregation.** There are two foundational aggregation strategies in FL: gradient aggregation and model aggregation. The key difference between these two strategies is that the former enables the server to train the global model by aggregating gradients and then performing gradient descent on the original one, whereas the latter allows the server to construct the global model based on local model parameters directly.

In Synchronous FL, due to the server-enforced synchronization requirement, each activated client is restricted to uploading a single update per global round, which is derived from training on the latest global model. Consequently, during the $(t + 1)$-th global aggregation round, given the index set (without duplicates) of activated clients in this round as $\mathcal{S}$, each client $\forall i \in \mathcal{S}$ performs local training within $E$ local epochs according to $w_i^t = w_g^t - \eta_i \sum_{e=1}^{E} \nabla F_i(w_{i,e-1}^t; \mathcal{D}_i)$, where $w_{i,0}^t \triangleq w_g^t$.

Denoted $\eta_g$ as the global learning rate and $n \triangleq \sum_{i \in \mathcal{S}} n_i$, the gradient aggregation strategy can be shown as $w_g^{t+1} = w_g^t - \eta_g \sum_{i \in \mathcal{S}} \frac{n_i}{n} \nabla F_i(w_i^t)$, where $\nabla F_i(w_i^t) \triangleq \sum_{e=1}^{E} \nabla F_i(w_{i,e-1}^t; \mathcal{D}_i)$.
*Remark* B.1. In FedSGD [1], the local epoch parameter $E$ is typically set to 1. When $E > 1$, this aggregation paradigm is commonly referred to as *model difference aggregation* [16, 37, 23], where clients transmit parameter differences calculated through multiple local epochs as $\Delta w_i^t = w_i^t - w_g^t = \eta_i \sum_{e=1}^{E} \nabla F_i(w_{i,e-1}^t; \mathcal{D}_i)$. The server then performs aggregation using $w_g^{t+1} = w_g^t - \sum_{i \in \mathcal{S}} \frac{n_i}{n} \Delta w_i^t$. For conciseness and without loss of generality, we collectively refer to scenarios with $E \geq 1$ as the gradient aggregation strategy. This unified terminology is justified since both FedSGD $(E = 1)$ and model difference aggregation $(E > 1)$ fundamentally utilize local gradient information $\nabla F_i(w_{i,e-1}^t; \mathcal{D}_i)$ during global aggregation processes.

In contrast, the model aggregation strategy requires each activated client to transmit its local model parameters to the server instead of the local gradients. The server then uses $w_g^{t+1} = \sum_{i \in \mathcal{S}} \frac{n_i}{n} w_i^t$ to form the new global model.

In SAFL, due to the autonomously local training across all clients, updates from clients with constrained computational resources may exhibit parameter staleness. Specifically, during the $(t + 1)$-th global round, client $C_i$ might complete its local training using the global model from the $\tau_i^t$-th round as $w_i^t = w_g^{\tau_i^t} - \eta_i \sum_{e=1}^{E} \nabla F_i(w_{i,e-1}^t; \mathcal{D}_i)$, where $w_{i,0}^t \triangleq w_g^{\tau_i^t}$.

Therefore, given the index list (possibly with duplicates) of the clients participating in aggregation as $\mathcal{S}$ at the $(t + 1)$-th global round, the gradient aggregation strategy and the model aggregation strategy in SAFL can be denoted as $w_g^{t+1} = w_g^t - \eta_g \sum_{i \in \mathcal{S}} \frac{n_i}{n} \nabla F_i(w_i^{\tau_i^t})$ and $w_g^{t+1} = \sum_{i \in \mathcal{S}} \frac{n_i}{n} w_i^{\tau_i^t}$, respectively, where $\nabla F_i(w_i^{\tau_i^t}) \triangleq \sum_{e=1}^{E} \nabla F_i(w_{i,e-1}^t; \mathcal{D}_i)$ with $w_{i,0}^t = w_g^{\tau_i^t}$.

## C  Discussion

### C.1  Superiority of SAFL.

The SAFL framework introduces an asynchronous federated learning paradigm that eliminates the necessity for synchronization among participants, enabling autonomous execution of local model updates and seamless parameter/gradient transmission to the server. This architecture fundamentally differs from conventional Synchronous FL implementations, where frequent idling occurs for inactivated clients to await completion of all activated clients' local training during synchronized global epoch, as well as from Asynchronous FL's aggregation mechanism that processes individual updates immediately upon reception. SAFL's superiority emerges through its conditional aggregation protocol, which employs dynamic triggering criteria based on predefined system conditions (e.g., resource availability thresholds, update quality metrics, or temporal constraints) to optimize both computational efficiency and model convergence characteristics. The framework's trigger-driven

aggregation mechanism concurrently addresses Synchronous FL's inherent resource under-utilization during prolonged waiting periods and Asynchronous FL's susceptibility to update volatility caused by premature aggregations, thereby achieving enhanced training throughput while maintaining model stability in dynamic network environments.

## C.2 Scalability of FedQS.

FedQS demonstrates strong scalability to large-scale networks comprising thousands of clients, as it introduces only minimal overhead. From the client-side perspective, each client performs only one additional similarity calculation and two numerical comparisons compared to baseline methods, with no inter-client communication required. Therefore, increasing the number of clients does not compromise client-side scalability. From the communication perspective, FedQS adds a 1-bit signal and one floating-point value to the upstream channel (client $\rightarrow$ server), and three floating-point values to the downstream channel (server $\rightarrow$ client). These additions constitute only a small fraction of the total transmitted data, which typically includes gradients or model parameters. Hence, the communication overhead introduced by FedQS remains negligible. The additional metadata increases the total communication volume by less than 1%, consisting of a 1-bit signal and one float uplink (4 bytes), and three floats downlink (12 bytes). For a model such as ResNet-18 ($\sim$43.7 MB), this represents an increase of only 0.000037% over FedAvg/FedSGD. This overhead requires no additional synchronization steps and scales linearly with the number of clients, yet it is substantially outweighed by the significant improvements in accuracy and convergence offered by FedQS. From the server-side perspective, the server maintains only a simple state table containing two integer–float key–value pairs compared to baseline approaches. Operations on this table execute in $O(1)$ time, ensuring that server-side scalability remains unaffected as the number of clients grows. Thus, even with large-scale client participation in federated tasks, FedQS maintains efficient performance on the server side.

## C.3 System Efficiency Analysis

This section details the efficiency analysis of FedQS's each module and discusses the potential optimizations to enhance the computational and communication efficiency.

### C.3.1 Breakdown Analysis

Table 7 provides a comprehensive breakdown of the computational latency, communication overhead, and memory footprint induced by FedQS's core mechanisms. These measurements represent averages over 400 global rounds, with consistent patterns observed across various experimental scenarios and client populations. The computation overhead remains minimal, with similarity scoring and feedback processing introducing only about 4s of additional latency per global round. In terms of communication, each client transmits merely 4B uplink (similarity score) and 1-bit feedback, while the server broadcasts 12B downlink (averaged thresholds). Memory footprint is also negligible: client-side storage for scores and feedback totals under 5B, and server-side state tables require only about 7.8KB for 100 clients.

Table 7: Computational latency, communication overhead, and memory footprint of FedQS components (averaged over 400 global rounds).

| Operation | Computational Cost | Communication Cost | Memory Footprint |
|---|---|---|---|
| Calculating pseudo global gradient (Mod①) | 13.73s | 0 | 42.8MB |
| Calculating similarity scores (Mod②) | 2.23s | 0 | 4B |
| Feedback signal calculation (Mod②) | 1.97s | 0 | 1 bit |
| Feedback signal communications (Mod② $\rightarrow$ Mod③) | 0 | Uplink: 1 bit | 0 |
| Updating aggregation status table (Mod③) | 0.27s | 0 | $\sim$7.8KB |
| Status table communications | 0 | Uplink: $\sim$4B Downlink: $\sim$12B | 0 |

### C.3.2 Computational Overhead of Divide-and-Conquer Strategy

We conducted disaggregated experiments to rigorously quantify the detailed additional runtime introduced by FedQS's divide-and-conquer strategy. The procedure is partitioned into three critical

phases: pseudo-gradient computation in Mod①; client classification in Mod②; and state-table updating in Mod③. Table 8 shows the average time consumed per global round for each step in FedQS when training ResNet-18 on CIFAR-10, where "Average ratio for Mod①" represents the ratio of the time spent calculating the pseudo-gradient in Mod① to the average time required to complete one global round.

Table 8: Computational overhead breakdown of FedQS components when training ResNet-18 on CIFAR-10 under different distributions.

| # Client | Data Dist. | Total Time (s) | Mod① Time (s) | Mod① Ratio | Mod② Time (s) | Mod② Ratio | Mod③ Time (s) | Mod③ Ratio |
|---|---|---|---|---|---|---|---|---|
| | x=0.1 | 44.98 | 6.69 | 14.87% | 2.17 | 4.82% | 0.13 | 0.29% |
| 50 | x=0.5 | 46.59 | 7.43 | 15.94% | 2.89 | 6.20% | 0.14 | 0.30% |
| | x=1 | 43.79 | 7.02 | 16.03% | 2.33 | 5.32% | 0.09 | 0.21% |
| | x=0.1 | 80.97 | 14.14 | 17.46% | 4.51 | 5.57% | 0.23 | 0.28% |
| 100 | x=0.5 | 83.41 | 13.93 | 16.70% | 4.22 | 5.06% | 0.26 | 0.31% |
| | x=1 | 82.77 | 13.11 | 15.84% | 3.96 | 4.78% | 0.33 | 0.39% |
| | x=0.1 | 165.83 | 22.33 | 13.47% | 8.11 | 4.89% | 0.49 | 0.30% |
| 200 | x=0.5 | 152.05 | 27.80 | 18.28% | 9.51 | 6.25% | 0.47 | 0.31% |
| | x=1 | 153.93 | 26.69 | 17.34% | 8.97 | 5.82% | 0.45 | 0.29% |
| **Average** | | - | - | **16.21%** | - | **5.41%** | - | **0.30%** |

### C.3.3 Potential Optimization

FedQS employs GPU-accelerated training with streaming aggregation to handle large models efficiently. This design overlaps communication and computation, boosting throughput for resource-constrained clients. Computationally, optimizations are available for the client-state table (caching scores, low-rank approximations) and the primary overhead of pseudo-gradient computation. Staggered client reclassification (i.e., reducing the frequency of client reclassification every 5 rounds) reduces overhead by 63.6–72.7% (1.4–5.2% accuracy drop), while stratified sampling of clients (i.e., 20% of clients per round re-evaluates the role) reduces it by 53.2–57.5% (2.2–4.1% performance loss). These strategies offer configurable tradeoffs within FedQS's modular design.

### C.4 Limitations of FedQS.

Although FedQS mitigates both the instability of gradient-based methods and the suboptimal convergence of model-based approaches, it introduces a few oscillations in model aggregation mode. Meanwhile, we introduce three hyperparameters $a, m_0, k$ in SAFL to propose FedQS. Despite the discussions in Section 5.4, this raises more difficulties in implementation and reproduction. A potential future work involves the automatic adjustment of these hyperparameters, therefore enhancing the flexibility and accessibility of the FedQS.

**Broader Impact.** This work proposed FedQS, a comprehensive and versatile optimization tool that enhances the performance of gradient aggregation and model aggregation strategies across multiple metrics. Additionally, we conduct a theoretical analysis of FedQS to demonstrate its superiority. We have identified no potential ethical impacts or noticeable negative social impacts associated with this work. On the contrary, FedQS serves as a compatible framework capable of optimizing both aggregation strategies in diverse scenarios, leading to significant performance improvements. This contribution advances the field by offering a practical and theoretically grounded solution for enhancing SAFL systems.

## D   Evaluation details & More evaluation results

### D.1   Details of the dataset

**CIFAR-10 [39].** CIFAR-10 is a widely used benchmark dataset for image classification tasks in machine learning. This dataset consists of ten labeled classes of images. Each class corresponds to 6,000 images, with 5,000 training samples and 1,000 test samples. In our FL scenario, we pursued benchmark [46], assuming that all participant data distributions follow the Hetero-Dirichlet distribution $Dir_k(x)$ based on categories, which can be represented by Equation 13. Each participant will split their local dataset into training and validation sets with an 8:2 ratio.

$$Dir_k(x) = \frac{\Gamma(\sum_{i=1}^{N} x_i)}{\Pi_{i=1}^{N}\Gamma(x_i)}\Pi_{i=1}^{N}\mathbb{P}_{k,i}^{x_i-1},$$ 
(13)

where $k$ represents the $k$-th client, $x$ is a parameter controlling the distribution, $\mathbb{P}_{k,i}$ represents the probability of having data from the $i$-th class and $\Gamma(x)$ is the Gamma-function. We primarily considered three levels of data distribution heterogeneity in our experiments: $x = 0.1$, $0.5$, and $1$.

**Shakespeare [1].** The Shakespeare dataset is a text corpus used for natural language processing tasks, comprising excerpts from *The Complete Works of William Shakespeare*. We embedded 80 unique characters, consisting of 26 uppercase English letters, 26 lowercase English letters, 10 numeric digits, and 18 special characters, into corresponding labels. Notably, we referenced benchmark [47] and modeled the dialogues of distinct roles in various scripts within Shakespeare as a heterogeneous distribution. Each participant was assigned data drawn from dialogue lines of different characters across diverse scripts, and the roles used by different participants do not overlap. Each participant will split their local dataset into training and validation sets with a 9:1 ratio. We primarily considered two levels of data distribution heterogeneity in our experiments: each client has 2 roles (i.e., $R = 200$) and 6 roles (i.e., $R = 600$).

**UCI Adult (Census Income) [41].** We refer to paper [41] to utilize the US Adult Income Dataset, a real-world benchmark dataset, to evaluate the efficacy of each algorithm in a practical task. The dataset comprises 48,842 records and 14 attributes, including age, gender, education level, marital status, occupation, working hours, etc.. We primarily predict whether an individual's annual income exceeds $50,000 based on demographic attributes from the census, assuming that each participant is associated with specific demographic characteristics (gender or ethnicity) corresponding to their income data. The distribution of data quantity among clients with the same characteristics follows a log-normal distribution $Log - \mathcal{N}(0, \sigma^2)$. Each participant will split their local dataset into training and validation sets with an 8:2 ratio. For the heterogeneous distribution based on gender, we primarily considered $\sigma = 1$ in our experiments. For the heterogeneous distribution based on ethnicity, we primarily considered $\sigma = 0.9$ in our experiments.

### D.2 Details of the models

**ResNet-18 [38].** We employ the Residual Network (ResNet) architecture, a class of neural networks introduced by He *et al.* [38], which incorporates residual blocks to alleviate vanishing gradients. Specifically, we utilize ResNet-18, a compact variant within the ResNet family, comprising 18 layers and four residual blocks. Each residual block consists of two convolutional layers with 3x3 kernels and a stride of 1, enabling efficient feature extraction and propagation through the network.

**LSTM [40].** We employ the Long Short-Term Memory (LSTM) network architecture, a subtype of Recurrent Neural Network (RNN) optimized for processing and learning sequential data. In our experimental setup, we employ a straightforward LSTM model comprising an embedding layer, an LSTM recurrent layer, and a fully connected dense layer.

**FCN.** We employ a Full Connection Neural Network (FCN), a canonical deep learning architecture. The FCN utilized in this paper comprises two fully connected dense layers, each enabled by the ReLU activation function to introduce non-linearity and improve feature extraction capabilities. Additionally, we incorporate a dropout layer to mitigate overfitting and enhance the robustness of our model.

### D.3 Details of hyperparameters and resource distributions

To ensure reproducibility, we provide a comprehensive summary of the default hyperparameter configurations for FedQS, which are used across most experiments in this study. The default settings in our experiments are as follows: number of participants $N = 100$, initial local learning rate $\eta_0 = 0.1$, learning rate bounds $\alpha = 0.001$ and $\beta = 0.2$, learning rate adaptation rate $a = 0.002$, initial momentum $m_0 = 0.1$, momentum adaptation parameter $k = 0.2$, global epochs $T = 400$, local epochs $E = 2$, minimum number of updates for aggregation $K = 10$, gradient clipping threshold $G_c = 20$, and momentum clipping threshold $\theta = 0.9$. FedQS introduces three additional hyperparameters—$a$, $m_0$, and $k$—which is comparable to the number used in recent state-of-the-art methods [18, 15]. These parameters control the adaptation rate of the learning rate, the initial momentum value, and the momentum adjustment rate, respectively. As illustrated in Figure 5, the method demonstrates stable performance across wide ranges of these hyperparameters ($a \in [0.001, 0.005]$, $m_0 \in [0.01, 0.1]$, $k \in [0.001, 0.2]$), substantially reducing the need for extensive

tuning. This behavior aligns with common federated learning practices and improves the practical usability of FedQS.

Notice that the simulation of poor communication quality and weak training performance is independent of the dataset and model selection/size. Instead, it is governed by actual client-specific configurations that determine resource demands and execution time. Our base experiments emulate SAFL heterogeneity with 50× speed differentials between the fastest and slowest participants, later extended it through additional tests under 1:20 and 1:100 resource ratios to comprehensively validate FedQS's robustness across diverse environments.

### D.4   Details of baselines

In this section, we will introduce the details of the baseline algorithms.

**FedAvg and FedSGD [1].**  These baseline algorithms are the basic aggregation algorithms implemented into the Synchronous and Semi-Asynchronous FL framework, without any additional optimization or improvement.

**SAFA [31].**  This baseline algorithm employs a caching mechanism to store updated models for each client. At the beginning of each aggregation round, the server updates the cache by incorporating the frequency of local update data uploads from each client. Subsequently, the server performs an aggregation operation on all the cached models and updates the cache once again after completing the aggregation process, taking into account the usage pattern of the latest local updates.

**FedAT [18].**  This baseline algorithm is a hierarchical SAFL framework, which synchronizes local model parameters within each layer and asynchronously updates the global model across layers. FedAT proposes a new weighted aggregation heuristic optimization target, where FL servers update different tiers' aggregation weights based on the statistics of different tiers' model aggregation frequencies, thereby balancing different tiers' model parameters.

**M-step-FedAsync [37].**  This baseline algorithm introduces a novel metric, model deviation degree, which is computed as the inner product between local model parameters and global model parameters. This metric serves as a key component in the aggregation process, where it is used in conjunction with local update frequency to determine the weights assigned to different model parameters during aggregation.

**FedBuff [16].**  This baseline algorithm employs a differential aggregation method, a variant of gradient aggregation, which assigns weights to each updated data. If the staleness of the difference update is high, the corresponding weight becomes smaller.

**WKAFL [15].**  This baseline algorithm extracts effective information from outdated gradients by leveraging recently updated gradients. It calculates the weighted aggregated parameters by calculating the cosine value between the unbiased gradient and the locally updated gradient to accelerate aggregation and stabilize the convergence process.

**FedAC [20].**  This baseline algorithm uses temporal gradient evaluation to assess client weights. It employs proactive weighted momentum for adaptive server updates, incorporating fine-grained gradient correction functionality designed by SCAFFOLD [3] to address the issue of client drift caused by heterogeneous data.

**DeFedAvg [42].**  In non-IID scenarios, DeFedAvg employs uniform sampling with replacement to select clients. Each client continuously receives the latest global model from the server and overwrites its reception buffer with the incoming model. Regardless of participation in global iterations, every client continuously performs local training. The server accepts delayed updates without waiting for participating clients to complete training based on the latest global model.

**FADAS [43].**  This algorithm treats the differences in client model updates as pseudo-gradients and employs an Adam-like update scheme to adjust the global model. Building upon FedBuff's local asynchronous training mechanism, FADAS retains the design philosophy of concurrency and buffer size to enable flexible control over the number of active clients and the frequency of global model updates.

**C$A^2$FL [44].**  This study confirms that asynchronous delay effects are amplified by highly non-IID data distributions and addresses the resulting client drift issue by employing a server-side mechanism

that retains the most recent updates from clients and reuses these cached updates to calibrate the global model.

## D.5 More experimental results

Table 9 shows the discrepancy ($T_s - T_f$) results, representing the convergence stability. Table 10 shows the results of FedQS and two foundational algorithms, FedAvg and FedSGD, under different SAFL system settings. Table 11 presents the average experimental results of FedSGD, FedAvg, and FedQS under various learning rates. Tables 12, 13, and 14 present the average results of FedQS under various settings of hyperparameter $a$, $m_0$, and $k$, respectively.

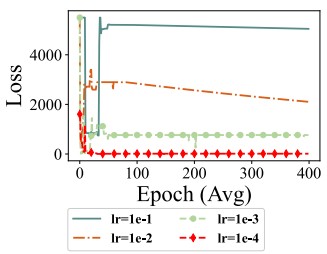

Figure 6: The impact of learning rate on WKAFL's loss in an RWD task (Gender).

In the RWD task (Figure 4f), WKAFL exhibits an extreme overfitting loss function curve, indicating poor adaptability of WKAFL to the RWD task when the local learning rate is set to 0.1. The adaptive learning rate adjustment module of WKAFL also contributes to the overfitting of the learning task. Figure 6 shows the experiment results of WKAFL under various local learning rates. We can see that when the local learning rate is reduced to 0.001, WKAFL no longer exhibits overfitting behavior.

Figure 7 shows the loss of FedQS and baselines under other tasks, which is a supplement of Figure 4. Figures 8 and 9 show the comparison of the accuracy and time latency between FedQS and each baseline within model aggregation and gradient aggregation strategies, respectively.

Oscillation is a metric to measure the occurrences when the accuracy of the global model in round $t$ is below that of the previous round $t - 1$ by a specific threshold [21]. Figure 10 shows the number of total oscillations under different thresholds when using ResNet-18 to train CIFAR-10.

Figure 11 depicts the performance comparison of FedQS between with and without momentum terms. Figure 12 depicts the performance comparison of FedQS between with and without feedback mechanisms.

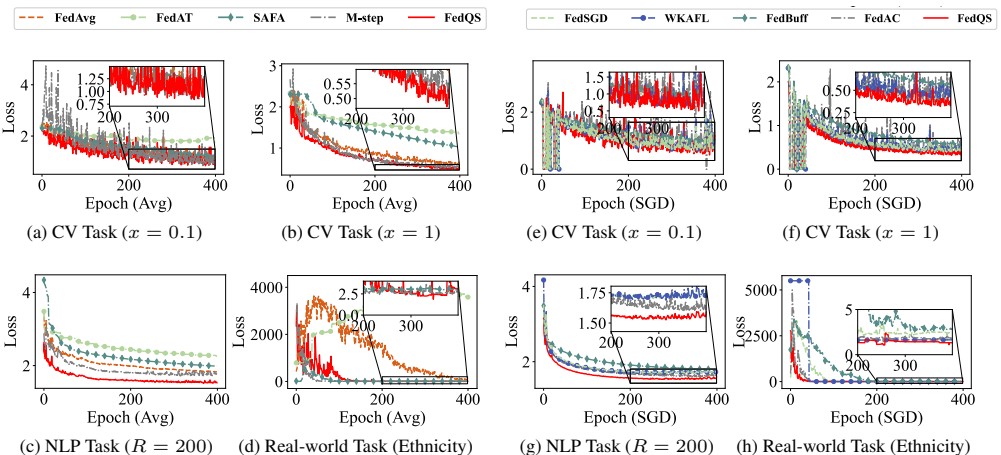

Figure 7: Loss of FedQS and the baselines under other tasks. The left four subfigures ((a)-(d)) are based on model aggregation, while the right four ((e)-(h)) are based on gradient aggregation.

## E Proof details

We will first give some basic notations of the FL tasks. Note that in the following part, all $|| \cdot ||$ symbols represent $\mathcal{L}_2$ norms and $< \cdot, \cdot >$ symbols represent inner products. In order to simplify and save space, we have denoted that $\nabla F_{g,e}(w_{g,e-1}^t) \triangleq \nabla F_g(w_{g,e}^t)$, $\nabla F_{i,e}(w_{i,e-1}^t) \triangleq \nabla F_i(w_{i,e}^t)$.

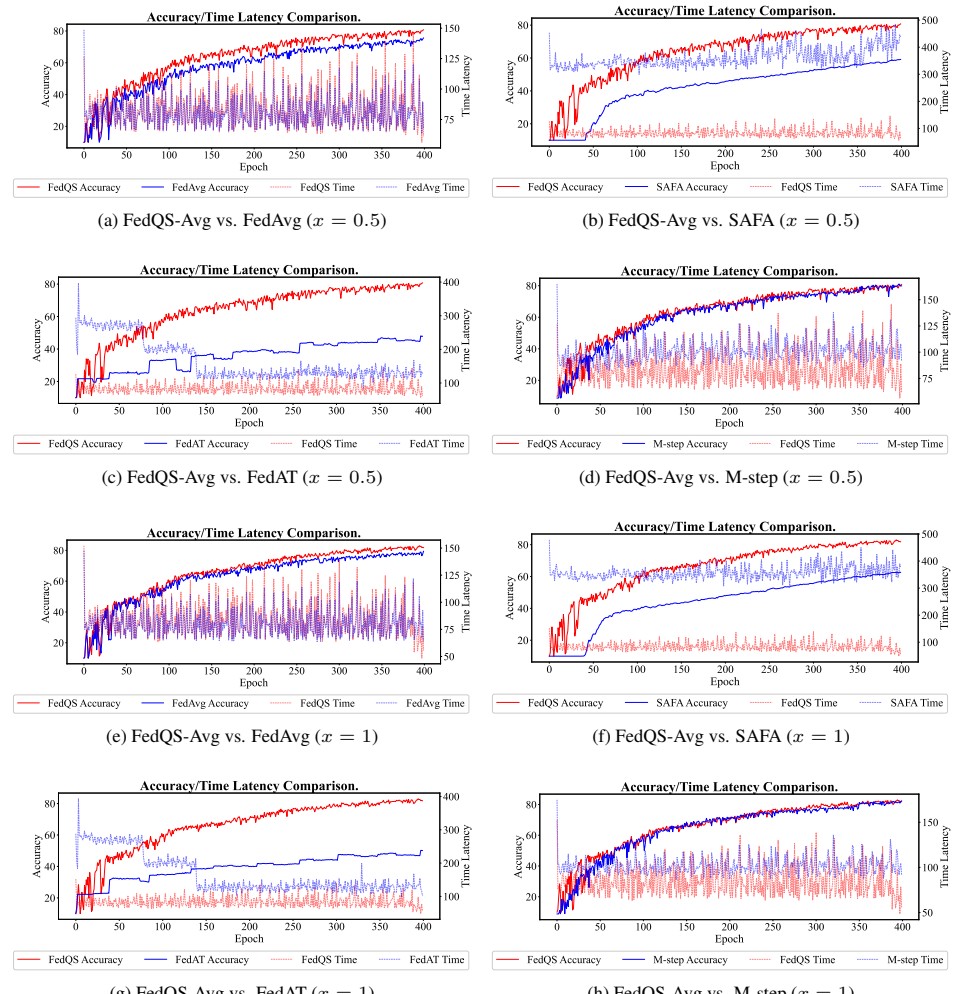

Figure 8: The comparison of accuracy and time latency between FedQS-Avg and the baselines with model aggregation strategy under CV tasks.

Table 9: Convergence stability comparisons between FedQS and baselines.

| Metrics | Tasks | Algorithms | | | | | | | | | | | |
|---------|-------|------------------|-------------------|------|-------|--------|----------|------------------|-------------------|---------|-------|-------|----------|
| | | FedAvg (SFL) | FedAvg (SAFL) | SAFA | FedAT | M-step | FedQS-Avg | FedSGD (SFL) | FedSGD (SAFL) | FedBuff | WKAFL | FedAC | FedQS-SGD |
| $T_s - T_f$ (# epoch) | $x = 0.1$ | 43 | 266 | 19 | 5 | 259 | 227 | 44 | 279 | 66 | 293 | 310 | 226 |
| | $x = 0.5$ | 22 | 33 | 5 | 29 | 40 | 20 | 20 | 45 | 5 | 45 | 322 | 24 |
| | $x = 1$ | 6 | 27 | 0 | 0 | 31 | 17 | 5 | 40 | 11 | 34 | 110 | 14 |
| | $R = 200$ | 0 | 0 | 0 | 0 | 11 | 7 | 0 | 23 | 0 | 4 | 25 | 0 |
| | $R = 600$ | 0 | 2 | 0 | 0 | 4 | 15 | 0 | 21 | 0 | 0 | 39 | 7 |
| | Gender | 12 | 51 | 24 | 0 | 128 | 48 | 16 | 202 | 0 | 0 | 16 | 26 |
| | Ethnicity | 24 | 151 | 114 | 0 | 45 | 77 | 27 | 251 | 178 | 0 | 48 | 31 |

* In column "Tasks", $x$ is the parameter of the Dirichlet distribution within CV tasks; $N$ is the number of roles within NLP tasks; Gender and Ethnicity are the data types within RWD tasks.
* The target accuracy of convergence stability is set to 80% of convergence accuracy in CV and NLP tasks and 95% of convergence accuracy in RWD tasks.

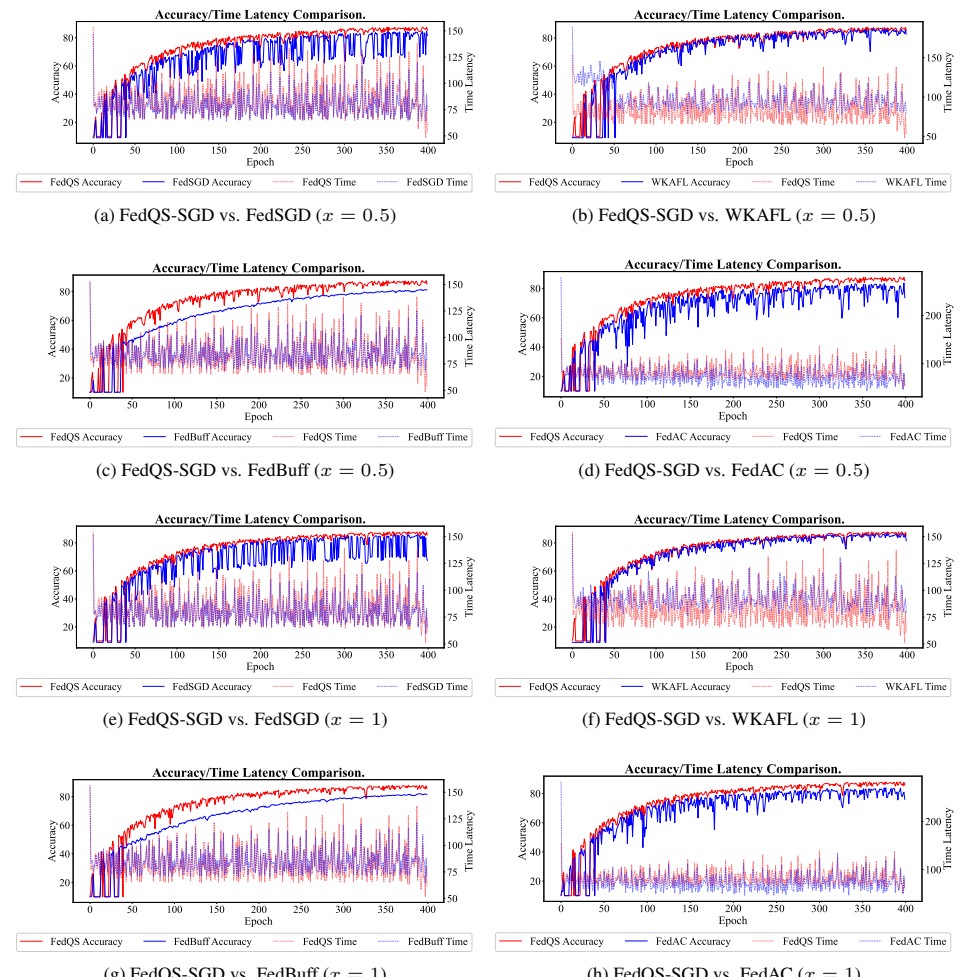

(a) FedQS-SGD vs. FedSGD ($x = 0.5$)

(b) FedQS-SGD vs. WKAFL ($x = 0.5$)

(c) FedQS-SGD vs. FedBuff ($x = 0.5$)

(d) FedQS-SGD vs. FedAC ($x = 0.5$)

(e) FedQS-SGD vs. FedSGD ($x = 1$)

(f) FedQS-SGD vs. WKAFL ($x = 1$)

(g) FedQS-SGD vs. FedBuff ($x = 1$)

(h) FedQS-SGD vs. FedAC ($x = 1$)

Figure 9: The comparison of accuracy and time latency between FedQS-SGD and the baselines with gradient aggregation strategy under CV tasks.

Table 10: Average performance of FedQS, FedAvg, and FedSGD w.r.t. the number of clients and resource distributions.

| | Tasks | | | | | | | | | | | | | | | |
| --- | --- | --- | --- | --- | --- | --- | --- | --- | --- | --- | --- | --- | --- | --- | --- | --- |
| | $N = 50$ | | | | | | | | $N = 200$ | | | | | | | |
| Metrics | 1:20 | | | | 1:50 | | | | 1:100 | | | | 1:50 | | | |
| | Fed | | FedQS | | Fed | | FedQS | | Fed | | FedQS | | Fed | | FedQS | |
| | Avg | SGD | Avg | SGD | Avg | SGD | Avg | SGD | Avg | SGD | Avg | SGD | Avg | SGD | Avg | SGD |
| M1 | 70.1 | 77.4 | **79.2** | **80.7** | 80.6 | 81.1 | **83.7** | **84.8** | 49.4 | 74.4 | **64.7** | **80.1** | 61.3 | 77.7 | **65.9** | **82.9** |
| M2 | 123 | 57 | **118** | **44** | 108 | 57 | **102** | **50** | 190 | 158 | **182** | **123** | 183 | 145 | **138** | **104** |
| M3 | 0.0 | 37.0 | 3.0 | **26.3** | 1.0 | 15.0 | 4.3 | **5.0** | 0.0 | 7.3 | 0.3 | **3.0** | 0.7 | 4.0 | 1.66 | **3.0** |

* In Metrics, M1 means Accuracy (%), M2 means Conv. speed (# epochs), and M3 means # Oscillations.
* In the table, $N = 200$ means the task has 200 clients, and 1:50 means the fastest client exhibits a training speed 50 times that of the slowest one. "Fed + Avg" means FedAvg. The threshold used to calculate the number of oscillations is set to 15. The target accuracy of convergence stability is set to 80% of convergence accuracy.
* Each result is an average value of three experiments corresponding to $x = 0.1, 0.5, 1$ in CV tasks.

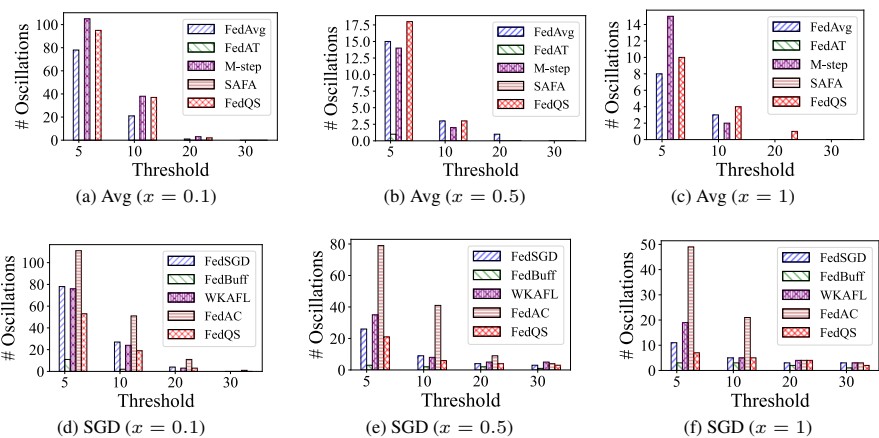

Figure 10: Statistics of oscillations under various thresholds when applying the ResNet-18 model to the CIFAR-10 dataset with $x = 0.1, 0.5, 1$.

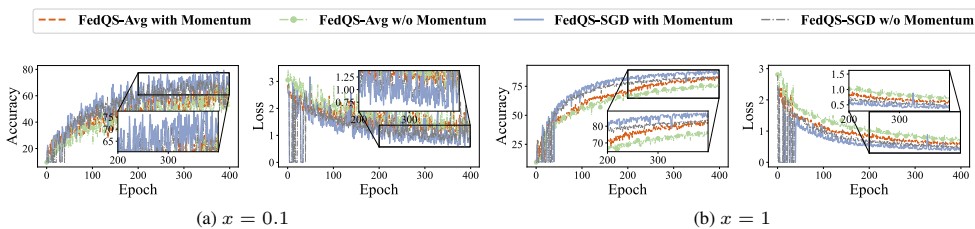

Figure 11: Accuracy and Loss of FedQS with or without Momentum terms under CV tasks.

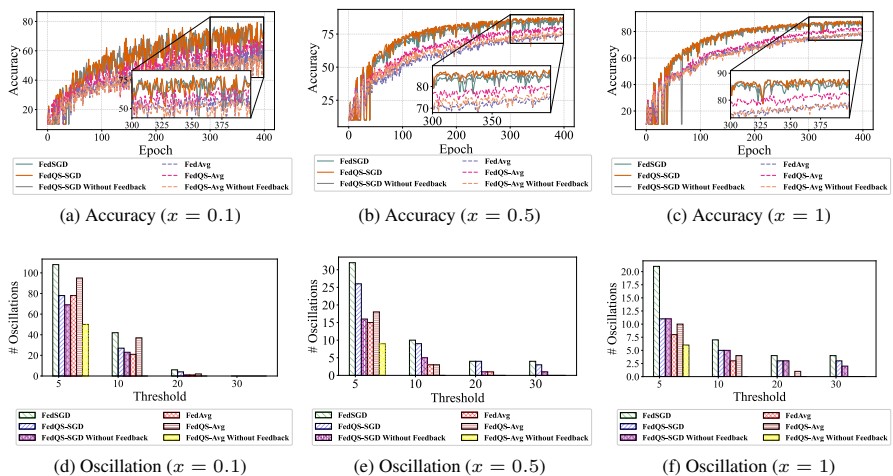

Figure 12: The impact of feedback mechanism on global accuracy and oscillations in CV tasks ($x = 0.1, 0.5, 1$).

Table 11: Impact of different learning rates on FedSGD, FedAvg, and FedQS's performance in a CV task ($x = 0.5$) and an NLP task ($R = 600$).

| Metrics | Task | Learning Rate | Algorithms | | | |
|---|---|---|---|---|---|---|
| | | | FedSGD | FedQS-SGD | FedAvg | FedQS-Avg |
| Accuracy (%) | CV | $\eta_0 = 0.1$ | 83.9 | **86.1** | 73.7 | **80.3** |
| | | $\eta_0 = 0.05$ | 84.9 | **85.6** | 72.8 | **77.1** |
| | | $\eta_0 = 0.01$ | 81.5 | **82.9** | 58.1 | **67.1** |
| | | $\eta_0 = 0.001$ | 61.1 | **80.7** | 40.9 | **61.4** |
| | NLP | $\eta_0 = 0.1$ | 49.6 | **52.5** | 45.5 | **50.1** |
| | | $\eta_0 = 0.05$ | 46.3 | **51.6** | 44.7 | **46.9** |
| | | $\eta_0 = 0.01$ | 35.9 | **48.7** | 30.3 | **41.8** |
| Convergence Speed (# epoch) | CV | $\eta_0 = 0.1$ | 86 | **82** | 144 | **109** |
| | | $\eta_0 = 0.05$ | 88 | **78** | 149 | **114** |
| | | $\eta_0 = 0.01$ | 90 | **84** | 157 | **121** |
| | | $\eta_0 = 0.001$ | 131 | **129** | 161 | **123** |
| | NLP | $\eta_0 = 0.1$ | 67 | **50** | 99 | **78** |
| | | $\eta_0 = 0.05$ | 74 | **68** | 102 | **78** |
| | | $\eta_0 = 0.01$ | 128 | **113** | 145 | 162 |

Table 12: Impact of different changing rates of the learning rate $a$ on FedQS's performance in CV and NLP tasks.

| Algorithms | Tasks | Value | Metrics | | |
|---|---|---|---|---|---|
| | | | Accuracy (%) | Convergence Speed (# epoch) | # Oscillations |
| FedQS-Avg | CV | 0.001 | 74.02 | 110 | **6.9** |
| | | 0.002 | **74.14** | **108** | 7.6 |
| | | 0.005 | 73.88 | 113 | 8.2 |
| | | 0.01 | 71.65 | 124 | 10.3 |
| | NLP | 0.001 | **49.88** | 93 | 3.6 |
| | | 0.002 | 49.75 | **91** | **3.5** |
| | | 0.005 | 49.56 | 95 | 3.9 |
| | | 0.01 | 43.25 | 103 | 3.6 |
| FedQS-SGD | CV | 0.001 | **80.63** | 84 | 4.3 |
| | | 0.002 | 80.59 | **81** | **4.0** |
| | | 0.005 | 80.44 | 83 | 4.7 |
| | | 0.01 | 74.83 | 92 | 6.3 |
| | NLP | 0.001 | **50.88** | **46** | **8.3** |
| | | 0.002 | 50.76 | 48 | 8.5 |
| | | 0.005 | 50.13 | 53 | 8.8 |
| | | 0.01 | 48.36 | 56 | 9.4 |

Table 13: Impact of different initial momentum $m_0$ on FedQS's performance in CV and NLP tasks.

| Algorithms | Tasks | Value | Metrics | | |
|---|---|---|---|---|---|
| | | | Accuracy (%) | Convergence Speed (# epoch) | # Oscillations |
| FedQS-Avg | CV | 0.1 | **74.14** | **108** | **7.6** |
| | | 0.2 | 73.26 | 112 | 7.9 |
| | | 0.3 | 72.48 | 116 | 7.6 |
| | | 0.4 | 70.69 | 123 | 7.8 |
| | NLP | 0.1 | **49.75** | **91** | **3.5** |
| | | 0.2 | 49.44 | 93 | 3.6 |
| | | 0.3 | 49.03 | 96 | 4.1 |
| | | 0.4 | 48.46 | 113 | 5.3 |
| FedQS-SGD | CV | 0.1 | **80.59** | **81** | **4.0** |
| | | 0.2 | 80.43 | 82 | 4.3 |
| | | 0.3 | 80.34 | 82 | 4.6 |
| | | 0.4 | 80.06 | 86 | 3.8 |
| | NLP | 0.1 | **50.76** | 48 | 8.5 |
| | | 0.2 | 50.43 | 46 | 8.6 |
| | | 0.3 | 50.22 | 49 | 8.9 |
| | | 0.4 | 50.13 | 48 | 8.8 |

Table 14: Impact of different changing rates of the momentum $k$ on FedQS's performance in CV and NLP tasks.

| Algorithms | Tasks | Value | Metrics | | |
|---|---|---|---|---|---|
| | | | Accuracy (%) | Convergence Speed (# epoch) | # Oscillations |
| FedQS-Avg | CV | 0.1 | **74.25** | 108 | 4.3 |
| | | 0.2 | 74.14 | 108 | **4.0** |
| | | 0.3 | 74.03 | 116 | 4.6 |
| | | 0.4 | 74.09 | 123 | 4.9 |
| | NLP | 0.1 | 49.65 | 93 | 3.6 |
| | | 0.2 | 49.75 | **91** | 3.5 |
| | | 0.3 | **49.76** | 100 | **3.4** |
| | | 0.4 | 49.66 | 112 | 3.8 |
| FedQS-SGD | CV | 0.1 | **80.77** | 83 | 9.4 |
| | | 0.2 | 80.59 | **81** | 7.6 |
| | | 0.3 | 79.85 | 92 | **7.1** |
| | | 0.4 | 79.33 | 116 | 8.4 |
| | NLP | 0.1 | **50.99** | **46** | **8.3** |
| | | 0.2 | 50.76 | 48 | 8.5 |
| | | 0.3 | 50.41 | 53 | 9.1 |
| | | 0.4 | 49.87 | 66 | 9.6 |

Additionally, when performing theoretical analysis, we also refer to the classic paper [35] and assume that the server uses an ideal dataset (based on balanced, independent, and identically distributed data from all participants) to train an ideal global model (which is just used to theoretical analysis and does not exist in reality), with the training process defined by Equation 14.

$$\widetilde{w}_{g,e}^t = \widetilde{w}_{g,e-1}^t - \eta_g \nabla F_{g,e}(\widetilde{w}_{g,e-1}^t), \tag{14}$$

where we define $\widetilde{w}_g^0 \triangleq w_g^0$, $F_g$ satisfies Assumption A.1, and $\nabla F_g$ satisfies Assumption A.2.

It's evident that we have the following relation between the ideal global model and the real global model:

$$\widetilde{w}_{g,E}^t = \widetilde{w}_g^t \,, \; \widetilde{w}_{g,0}^t = w_g^{t-1}. \tag{15}$$

Then, based on the Assumption A.2 and A.3, we can give the following lemma:

**Lemma E.1.** *Let $a_i := \nabla F_g(\widetilde{w}_g^t) - \nabla F_i(w_i^t)$, $b_i := \nabla F_i(w_i^t)$. Given the Assumption A.1 and A.3, then any one of the following conditions is sufficient to guarantee $\mathbb{E}\|\nabla F_g(\widetilde{w}_g^t)\|^2 \leq \delta^2$. (1) Global Optimum: $\nabla F_g(\widetilde{w}_g^t) = 0$. (2) Local-parameter closeness: there exist constants $\varepsilon_w \geq 0$ and $\widetilde{\delta} \geq 0$ such that $\mathbb{E}\|w_i^t - \widetilde{w}_g^t\|^2 \leq \varepsilon_w^2, \mathbb{E}\|\nabla F_g(\widetilde{w}_g^t) - \nabla F_i(\widetilde{w}_g^t)\|^2 \leq \widetilde{\delta}^2$, and the following numerical relation holds: $\widetilde{\delta}^2 + L^2\varepsilon_w^2 + G_c^2 \leq \frac{\delta^2}{3}$. (3) Negative correlation: The cross-term satisfies $\mathbb{E}\langle a_i, b_i \rangle \leq -\frac{1}{2}\mathbb{E}\|b_i\|^2$, which may occur at local minima/saddle points or under extreme data heterogeneity.*

**Proof.** Items (1) is immediate from algebra. We briefly show (2) and (3).

For (2): First use the triangle inequality

$$\|\nabla F_g(\widetilde{w}_g^t)\| = \|\nabla F_g(\widetilde{w}_g^t) - \nabla F_i(\widetilde{w}_g^t) + \nabla F_i(\widetilde{w}_g^t) - \nabla F_i(w_i^t) + \nabla F_i(w_i^t)\|$$
$$\leq \|\nabla F_g(\widetilde{w}_g^t) - \nabla F_i(\widetilde{w}_g^t)\| + \|\nabla F_i(\widetilde{w}_g^t) - \nabla F_i(w_i^t)\| + \|\nabla F_i(w_i^t)\|.$$

Square both sides and apply the inequality $(u + v + w)^2 \leq 3(u^2 + v^2 + w^2)$:

$$\|\nabla F_g(\widetilde{w}_g^t)\|^2 \leq 3\Big(\|\nabla F_g(\widetilde{w}_g^t) - \nabla F_i(\widetilde{w}_g^t)\|^2 + \|\nabla F_i(\widetilde{w}_g^t) - \nabla F_i(w_i^t)\|^2 + \|\nabla F_i(w_i^t)\|^2\Big).$$

Take expectation over clients/data randomness and use the assumed bounds. By definition of $\widetilde{\delta}^2$ we have $\mathbb{E}\|\nabla F_g(\widetilde{w}_g^t) - \nabla F_i(\widetilde{w}_g^t)\|^2 \leq \widetilde{\delta}^2$. By $L$-smoothness and the bound on parameter deviation,

$$\mathbb{E}\|\nabla F_i(\widetilde{w}_g^t) - \nabla F_i(w_i^t)\|^2 \leq L^2\,\mathbb{E}\|\widetilde{w}_g^t - w_i^t\|^2 \leq L^2\varepsilon_w^2.$$

Finally use $\mathbb{E}\|\nabla F_i(w_i^t)\|^2 \leq G_c^2$. Combining these yields

$$\mathbb{E}\|\nabla F_g(\widetilde{w}_g^t)\|^2 \leq 3\big(\widetilde{\delta}^2 + L^2\varepsilon_w^2 + G_c^2\big).$$

Thus if $\widetilde{\delta}^2 + L^2\varepsilon_w^2 + G_c^2 \leq \delta^2/3$, the claimed inequality $\mathbb{E}\|\nabla F_g(\widetilde{w}_g^t)\|^2 \leq \delta^2$ holds.

For (3): By definition,

$$\|\nabla F_g(\widetilde{w}_g^t)\|^2 = \|a_i + b_i\|^2 = \|a_i\|^2 + \|b_i\|^2 + 2\langle a_i, b_i\rangle.$$

Taking expectation and using $\mathbb{E}\|a_i\|^2 \leq \delta^2$ and the assumed cross-term bound gives

$$\mathbb{E}\|\nabla F_g(\widetilde{w}_g^t)\|^2 \leq \delta^2 + \mathbb{E}\|b_i\|^2 + 2 \cdot (-\tfrac{1}{2}\mathbb{E}\|b_i\|^2) = \delta^2.$$

$\square$

*Remark* E.2. Condition (2) decomposes the sources that contribute to the magnitude of $\nabla F_g(\widetilde{w}_g^t)$: (i) heterogeneity at the ideal point $(\widetilde{\delta}^2)$, (ii) parameter drift of local models $(L^2\varepsilon_w^2)$, and (iii) the intrinsic size of local gradients $(G_c^2)$. This decomposition is useful in practice for designing communication or correction mechanisms.

## E.1 Proof of Theorem A.4

Denoted $Q(t)$ as the maximum number of clients that execute the momentum update at global round $t$, we begin the proof with two lemmas:

**Lemma E.3** (Ideal global model difference). *Given $\mathcal{V} = (3 - \frac{2\beta^2 R}{\beta^2+1})$ and the ideal global model trained in global epoch $t$, the difference between the optimal can be bounded by:*

$$
\begin{aligned}
\mathbb{E}[||\widetilde{w}_g^t - w^*||^2] \leq{}& \mathcal{V}^t\mathbb{E}[||\widetilde{w}_g^0 - w^*||^2] + \frac{6\beta^2(\beta^2+1)}{2\beta^2 R - 2\beta^2 - 2}E^2\delta^2 \\
&+ 3\sum_{j=1}^{t}\mathcal{V}^j\mathbb{E}[||\eta_i^{\tau_i^{t-1}}\sum_{e=1}^{E}\sum_{r=1}^{e}(m_i^{\tau_i^{t-1}})^r\nabla F_i(w_{i,e-r}^{\tau_i^{t-1}})||^2] \\
&+ 2\sum_{j=1}^{t}\mathcal{V}^j\mathbb{E}[||\sum_{e=1}^{E}\nabla F_i(w_{i,e}^{\tau_i^{t-1}})||^2].
\end{aligned}
\tag{16}
$$

**Proof.** By the definition of the ideal global model, we add a zero term and get:

$$
\begin{aligned}
&||\widetilde{w}_g^t - w^*||^2 \\
\leq{}& ||w_g^{t-1} - \eta_g\sum_{e=1}^{E}\nabla F_g(\widetilde{w}_{g,e}^t) - w^*||^2 \\
={}& ||w_g^{t-1} - w^* + \sum_{i=1}^{K}p_i\sum_{e=1}^{E}\eta_i^{\tau_i^{t-1}}\nabla F_i(w_{i,e}^{\tau_i^{t-1}})||^2 \\
&+ ||-\sum_{i=1}^{K}p_i\sum_{e=1}^{E}\eta_i^{\tau_i^{t-1}}\nabla F_i(w_{i,e}^{\tau_i^{t-1}}) - \eta_g\sum_{e=1}^{E}\nabla F_g(\widetilde{w}_{g,e}^t)||^2 \\
&+ 2\langle w_g^{t-1} - w^* - \eta_g\sum_{i=1}^{K}\sum_{e=1}^{E}\nabla F_i(w_{i,e}^{\tau_i^t}), -\sum_{i=1}^{K}p_i\sum_{e=1}^{E}\eta_i^{\tau_i^{t-1}}\nabla F_i(w_{i,e}^{\tau_i^{t-1}}) - \eta_g\sum_{e=1}^{E}\nabla F_g(\widetilde{w}_{g,e}^t)\rangle
\end{aligned}
\tag{17}
$$

Consider the relationship between $w_g^{t-1}$ and $\widetilde{w}_g^{t-1}$, we have:

$$
\begin{aligned}
||\widetilde{w}_g^t - w^*||^2 = ||\widetilde{w}_g^{t-1} + \eta_g \sum_{e=1}^{E} \nabla F_g(\widetilde{w}_{g,e}^{t-1}) - w^* + \sum_{i=1}^{K} p_i \sum_{e=1}^{E} \eta_i^{\tau_i^{t-1}} \nabla F_i(w_{i,e}^{\tau_i^{t-1}}) \\
- \sum_{i=1}^{K} p_i \sum_{e=1}^{E} \eta_i^{\tau_i^{t-1}} [\sum_{r=1}^{e} (m_i^{\tau_i^{t-1}})^r \nabla F_i(w_{i,e-r}^{\tau_i^{t-1}}) + \nabla F_i(w_{i,e}^{\tau_i^{t-1}})]||^2 \\
+ ||\sum_{i=1}^{K} p_i \sum_{e=1}^{E} \eta_i^{\tau_i^{t-1}} \nabla F_i(w_{i,e}^{\tau_i^{t-1}}) + \eta_g \sum_{e=1}^{E} \nabla F_g(\widetilde{w}_{g,e}^t)||^2 \\
+ 2\langle \widetilde{w}_g^{t-1} - \sum_{i=1}^{K} p_i \sum_{e=1}^{E} \eta_i^{\tau_i^{t-1}} [\sum_{r=1}^{e} (m_i^{\tau_i^{t-1}})^r \nabla F_i(w_{i,e-r}^{\tau_i^{t-1}}) + \nabla F_i(w_{i,e}^{\tau_i^{t-1}})], \\
\sum_{i=1}^{K} p_i \sum_{e=1}^{E} \eta_i^{\tau_i^{t-1}} \nabla F_i(w_{i,e}^{\tau_i^{t-1}}) + \eta_g \sum_{e=1}^{E} \nabla F_g(\widetilde{w}_{g,e}^t)\rangle \\
+ 2\langle \eta_g \sum_{e=1}^{E} \nabla F_g(\widetilde{w}_{g,e}^{t-1}) - w^* + \sum_{i=1}^{K} p_i \sum_{e=1}^{E} \eta_i^{\tau_i^{t-1}} \nabla F_i(w_{i,e}^{\tau_i^{t-1}}), \\
\sum_{i=1}^{K} p_i \sum_{e=1}^{E} \eta_i^{\tau_i^{t-1}} \nabla F_i(w_{i,e}^{\tau_i^{t-1}}) + \eta_g \sum_{e=1}^{E} \nabla F_g(\widetilde{w}_{g,e}^t)\rangle.
\end{aligned}
$$

Therefore,

$$
\begin{aligned}
||\widetilde{w}_g^t - w^*||^2 \\
\leq ||\widetilde{w}_g^{t-1} - w^*||^2 + ||\eta_g \sum_{e=1}^{E} \nabla F_g(\widetilde{w}_{g,e}^{t-1}) - \sum_{i=1}^{K} p_i \sum_{e=1}^{E} \eta_i^{\tau_i^{t-1}} \sum_{r=1}^{e} (m_i^{\tau_i^{t-1}})^r \nabla F_i(w_{i,e-r}^{\tau_i^{t-1}})||^2 \\
+ 2\langle \widetilde{w}_g^{t-1} - w^*, \eta_g \sum_{e=1}^{E} \nabla F_g(\widetilde{w}_{g,e}^{t-1}) - \sum_{i=1}^{K} p_i \sum_{e=1}^{E} \eta_i^{\tau_i^{t-1}} \sum_{r=1}^{e} (m_i^{\tau_i^{t-1}})^r \nabla F_i(w_{i,e-r}^{\tau_i^{t-1}})\rangle \\
+ ||\sum_{i=1}^{K} p_i \sum_{e=1}^{E} \eta_i^{\tau_i^{t-1}} \nabla F_i(w_{i,e}^{\tau_i^{t-1}}) + \eta_g \sum_{e=1}^{E} \nabla F_g(\widetilde{w}_{g,e}^t)||^2 \\
+ 2\langle \widetilde{w}_g^{t-1} - w^*, \sum_{i=1}^{K} p_i \sum_{e=1}^{E} \eta_i^{\tau_i^{t-1}} \nabla F_i(w_{i,e}^{\tau_i^{t-1}}) + \eta_g \sum_{e=1}^{E} \nabla F_g(\widetilde{w}_{g,e}^t)\rangle \\
+ 2\langle \eta_g \sum_{e=1}^{E} \nabla F_g(\widetilde{w}_{g,e}^{t-1}) - \sum_{i=1}^{K} p_i \sum_{e=1}^{E} \eta_i^{\tau_i^{t-1}} \sum_{r=1}^{e} (m_i^{\tau_i^{t-1}})^r \nabla F_i(w_{i,e-r}^{\tau_i^{t-1}}), \\
\sum_{i=1}^{K} p_i \sum_{e=1}^{E} \eta_i^{\tau_i^{t-1}} \nabla F_i(w_{i,e}^{\tau_i^{t-1}}) + \eta_g \sum_{e=1}^{E} \nabla F_g(\widetilde{w}_{g,e}^t)\rangle.
\end{aligned} \tag{18}
$$

For the 2nd term:

$$
\begin{aligned}
||\eta_g \sum_{e=1}^{E} \nabla F_g(\widetilde{w}_{g,e}^{t-1}) - \sum_{i=1}^{K} p_i \sum_{e=1}^{E} \eta_i^{\tau_i^{t-1}} \sum_{r=1}^{e} (m_i^{\tau_i^{t-1}})^r \nabla F_i(w_{i,e-r}^{\tau_i^{t-1}})||^2 \\
\leq \beta^2 ||\sum_{e=1}^{E} \nabla F_g(\widetilde{w}_{g,e}^{t-1})||^2 + ||\sum_{i=1}^{K} p_i \sum_{e=1}^{E} \eta_i^{\tau_i^{t-1}} \sum_{r=1}^{e} (m_i^{\tau_i^{t-1}})^r \nabla F_i(w_{i,e-r}^{\tau_i^{t-1}})||^2 \\
- 2\beta^2 \langle \sum_{e=1}^{E} \nabla F_g(\widetilde{w}_{g,e}^{t-1}), \sum_{i=1}^{K} p_i \sum_{e=1}^{E} \sum_{r=1}^{e} (m_i^{\tau_i^{t-1}})^r \nabla F_i(w_{i,e-r}^{\tau_i^{t-1}})\rangle.
\end{aligned} \tag{19}
$$

Based on [35], we have:

$$\mathbb{E}[||\eta_g \sum_{e=1}^{E} \nabla F_g(\widetilde{w}_{g,e}^{t-1}) - \sum_{i=1}^{K} p_i \sum_{e=1}^{E} \eta_i^{\tau_i^{t-1}} \sum_{r=1}^{e} (m_i^{\tau_i^{t-1}})^r \nabla F_i(w_{i,e-r}^{\tau_i^{t-1}})||^2]$$

$$\leq \beta^2 E^2 \delta^2 + \mathbb{E}[||\eta_i^{\tau_i^{t-1}} \sum_{e=1}^{E} \sum_{r=1}^{e} (m_i^{\tau_i^{t-1}})^r \nabla F_i(w_{i,e-r}^{\tau_i^{t-1}})||^2]$$

$$- 2\beta^2 \sum_{e=1}^{E} \mathbb{E}[\langle \nabla F_g(\widetilde{w}_{g,e}^{t-1}), \sum_{i=1}^{K} p_i \sum_{r=1}^{e} (m_i^{\tau_i^{t-1}})^r \nabla F_i(w_{i,e-r}^{\tau_i^{t-1}})\rangle]$$

$$\leq \beta^2 E^2 \delta^2 + \mathbb{E}[||\eta_i^{\tau_i^{t-1}} \sum_{e=1}^{E} \sum_{r=1}^{e} (m_i^{\tau_i^{t-1}})^r \nabla F_i(w_{i,e-r}^{\tau_i^{t-1}})||^2]$$

$$- 2\beta^2 \sum_{e=1}^{E} \sum_{i=1}^{K} p_i \sum_{r=1}^{e} (m_i^{\tau_i^{t-1}})^r \mathbb{E}[\langle \nabla F_g(\widetilde{w}_{g,e}^{t-1}), \nabla F_i(w_{i,e-r}^{\tau_i^{t-1}})\rangle]$$

$$= \beta^2 E^2 \delta^2 + \mathbb{E}[||\eta_i^{\tau_i^{t-1}} \sum_{e=1}^{E} \sum_{r=1}^{e} (m_i^{\tau_i^{t-1}})^r \nabla F_i(w_{i,e-r}^{\tau_i^{t-1}})||^2]. \tag{20}$$

For the 3rd term, using a variant of reversed Cauchy-Schwarz inequality [48, 49, 50] with $\gamma = \beta$ and $\Gamma = \frac{1}{\beta}$ and $m_i^{\tau_i^{t-1}} \leq 1$, we have:

$$2\langle \widetilde{w}_g^{t-1} - w^*, \eta_g \sum_{e=1}^{E} \nabla F_g(\widetilde{w}_{g,e}^{t-1}) - \sum_{i=1}^{K} p_i \sum_{e=1}^{E} \eta_i^{\tau_i^{t-1}} \sum_{r=1}^{e} (m_i^{\tau_i^{t-1}})^r \nabla F_i(w_{i,e-r}^{\tau_i^{t-1}})\rangle$$

$$= 2\langle \widetilde{w}_g^{t-1} - w^*, \eta_g \sum_{e=1}^{E} \nabla F_g(\widetilde{w}_{g,e}^{t-1})\rangle - 2\langle \widetilde{w}_g^{t-1} - w^*, \sum_{i=1}^{K} p_i \sum_{e=1}^{E} \eta_i^{\tau_i^{t-1}} \sum_{r=1}^{e} (m_i^{\tau_i^{t-1}})^r \nabla F_i(w_{i,e-r}^{\tau_i^{t-1}})\rangle$$

$$\leq ||\widetilde{w}_g^{t-1} - w^*||^2 + ||\eta_g \sum_{e=1}^{E} \nabla F_g(\widetilde{w}_{g,e}^{t-1})||^2 - \frac{2\beta^2 R}{\beta^2 + 1}||\widetilde{w}_g^{t-1} - w^*||^2$$

$$\leq (1 - \frac{2\beta^2 R}{\beta^2 + 1})||\widetilde{w}_g^{t-1} - w^*||^2 + ||\eta_g \sum_{e=1}^{E} \nabla F_g(\widetilde{w}_{g,e}^{t-1})||^2. \tag{21}$$

Then,

$$\mathbb{E}[2\langle \widetilde{w}_g^{t-1} - w^*, \eta_g \sum_{e=1}^{E} \nabla F_g(\widetilde{w}_{g,e}^{t-1}) - \sum_{i=1}^{K} p_i \sum_{e=1}^{E} \eta_i^{\tau_i^{t-1}} \sum_{r=1}^{e} (m_i^{\tau_i^{t-1}})^r \nabla F_i(w_{i,e-r}^{\tau_i^{t-1}})\rangle]$$

$$\leq (1 - \frac{2\beta^2 R}{\beta^2 + 1})\mathbb{E}[||\widetilde{w}_g^{t-1} - w^*||^2] + \beta^2 E^2 \delta^2. \tag{22}$$

For the 4th term:

$$\|\sum_{i=1}^{K} p_i \sum_{e=1}^{E} \eta_i^{\tau_i^{t-1}} \nabla F_i(w_{i,e}^{\tau_i^{t-1}}) + \eta_g \sum_{e=1}^{E} \nabla F_g(\widetilde{w}_{g,e}^t)\|^2$$

$$= \|\sum_{i=1}^{K} p_i \sum_{e=1}^{E} \eta_i^{\tau_i^{t-1}} \nabla F_i(w_{i,e}^{\tau_i^{t-1}})\|^2 + \|\eta_g \sum_{e=1}^{E} \nabla F_g(\widetilde{w}_{g,e}^t)\|^2$$

$$+ \langle \sum_{i=1}^{K} p_i \sum_{e=1}^{E} \eta_i^{\tau_i^{t-1}} \nabla F_i(w_{i,e}^{\tau_i^{t-1}}), \eta_g \sum_{e=1}^{E} \nabla F_g(\widetilde{w}_{g,e}^t) \rangle$$

$$\leq \eta_g^2 E^2 \|\nabla F_g(\widetilde{w}_{g,e}^t)\|^2 + \|\sum_{e=1}^{E} \eta_i^{\tau_i^{t-1}} \nabla F_i(w_{i,e}^{\tau_i^{t-1}})\|^2 + \sum_{e=1}^{E}\sum_{r=1}^{E} \beta^2 \langle p_i \nabla F_i(w_{i,e}^{\tau_i^{t-1}}), \nabla F_g(\widetilde{w}_{g,r}^t) \rangle.$$

$$(23)$$

Then based on [35],

$$\mathbb{E}[\|\sum_{i=1}^{K} p_i \sum_{e=1}^{E} \eta_i^{\tau_i^{t-1}} \nabla F_i(w_{i,e}^{\tau_i^{t-1}}) + \eta_g \sum_{e=1}^{E} \nabla F_g(\widetilde{w}_{g,e}^t)\|^2]$$

$$\leq \eta_g^2 E^2 \delta^2 + \mathbb{E}[\|\eta_i^{\tau_i^{t-1}} \sum_{e=1}^{E} \nabla F_i(w_{i,e}^{\tau_i^{t-1}})\|^2] + \sum_{i=1}^{K}\sum_{e=1}^{E}\sum_{r=1}^{E} \beta^2 \mathbb{E}[\langle p_i \nabla F_i(w_{i,e}^{\tau_i^{t-1}}), \nabla F_g(\widetilde{w}_{g,r}^t) \rangle]$$

$$= \beta^2 E^2 \delta^2 + \mathbb{E}[\|\eta_i^{\tau_i^{t-1}} \sum_{e=1}^{E} \nabla F_i(w_{i,e}^{\tau_i^{t-1}})\|^2].$$

$$(24)$$

For the 5th term, using AM-GM inequality and Cauchy-Schwarz inequality, we have:

$$2\langle \widetilde{w}_g^{t-1} - w^*, \sum_{i=1}^{K} p_i \sum_{e=1}^{E} \eta_i^{\tau_i^{t-1}} \nabla F_i(w_{i,e}^{\tau_i^{t-1}}) + \eta_g \sum_{e=1}^{E} \nabla F_g(\widetilde{w}_{g,e}^t) \rangle$$

$$\leq \|\widetilde{w}_g^{t-1} - w^*\|^2 + \|\sum_{e=1}^{E} \eta_i^{\tau_i^{t-1}} \nabla F_i(w_{i,e}^{\tau_i^{t-1}})\|^2 + \|\eta_g \sum_{e=1}^{E} \nabla F_g(\widetilde{w}_{g,e}^t)\|^2$$

$$+ \langle \sum_{i=1}^{K} p_i \sum_{e=1}^{E} \eta_i^{\tau_i^{t-1}} \nabla F_i(w_{i,e}^{\tau_i^{t-1}}), \eta_g \sum_{e=1}^{E} \nabla F_g(\widetilde{w}_{g,e}^t) \rangle \qquad (25)$$

$$\leq \|\widetilde{w}_g^{t-1} - w^*\|^2 + \eta_g^2 \|\sum_{e=1}^{E} \nabla F_g(\widetilde{w}_{g,e}^t)\|^2 + \|\eta_i^{\tau_i^{t-1}} \sum_{e=1}^{E} \nabla F_i(w_{i,e}^{\tau_i^{t-1}})\|^2$$

$$+ \langle \sum_{i=1}^{K} p_i \sum_{e=1}^{E} \eta_i^{\tau_i^{t-1}} \nabla F_i(w_{i,e}^{\tau_i^{t-1}}), \eta_g \sum_{e=1}^{E} \nabla F_g(\widetilde{w}_{g,e}^t) \rangle.$$

Then, taking expectation, we have,

$$\mathbb{E}[2\langle \widetilde{w}_g^{t-1} - w^*, \sum_{i=1}^{K} p_i \sum_{e=1}^{E} \eta_i^{\tau_i^{t-1}} \nabla F_i(w_{i,e}^{\tau_i^{t-1}}) + \eta_g \sum_{e=1}^{E} \nabla F_g(\widetilde{w}_{g,e}^t)\rangle]$$

$$\leq \mathbb{E}[||\widetilde{w}_g^{t-1} - w^*||^2] + \beta^2 E^2 \delta^2 + \mathbb{E}[||\eta_i^{\tau_i^{t-1}} \sum_{e=1}^{E} \nabla F_i(w_{i,e}^{\tau_i^{t-1}})||^2]$$

$$+ \mathbb{E}[\langle \sum_{i=1}^{K} p_i \sum_{e=1}^{E} \eta_i^{\tau_i^{t-1}} \nabla F_i(w_{i,e}^{\tau_i^{t-1}}), \eta_g \sum_{e=1}^{E} \nabla F_g(\widetilde{w}_{g,e}^t)\rangle]$$

$$= \mathbb{E}[||\widetilde{w}_g^{t-1} - w^*||^2] + \beta^2 E^2 \delta^2 + \mathbb{E}[||\sum_{e=1}^{E} \nabla F_i(w_{i,e}^{\tau_i^{t-1}})||^2].$$

(26)

For the 6th term:

$$2\langle \eta_g \sum_{e=1}^{E} \nabla F_g(\widetilde{w}_{g,e}^{t-1}), \sum_{i=1}^{K} p_i \sum_{e=1}^{E} \eta_i^{\tau_i^{t-1}} \nabla F_i(w_{i,e}^{\tau_i^{t-1}}) + \eta_g \sum_{e=1}^{E} \nabla F_g(\widetilde{w}_{g,e}^t)\rangle$$

$$= 2\langle \eta_g \sum_{e=1}^{E} \nabla F_g(\widetilde{w}_{g,e}^{t-1}), \sum_{i=1}^{K} p_i \sum_{e=1}^{E} \eta_i^{\tau_i^{t-1}} \nabla F_i(w_{i,e}^{\tau_i^{t-1}})\rangle + 2\langle \eta_g \sum_{e=1}^{E} \nabla F_g(\widetilde{w}_{g,e}^{t-1}), \eta_g \sum_{e=1}^{E} \nabla F_g(\widetilde{w}_{g,e}^t)\rangle$$

$$\leq 2\eta_g \sum_{e=1}^{E} \sum_{i=1}^{K} p_i \sum_{r=0}^{E} \beta \langle \nabla F_g(\widetilde{w}_{g,e}^{t-1}), \nabla F_i(w_{i,r}^{\tau_i^{t-1}})\rangle + 2||\eta_g \sum_{e=1}^{E} \nabla F_g(\widetilde{w}_{g,e}^{t-1})||^2.$$

(27)

Then, based on [35] we have:

$$\mathbb{E}[2\langle \eta_g \sum_{e=1}^{E} \nabla F_g(\widetilde{w}_{g,e}^{t-1}), \sum_{i=1}^{K} p_i \sum_{e=1}^{E} \eta_i^{\tau_i^{t-1}} \nabla F_i(w_{i,e}^{\tau_i^{t-1}}) + \eta_g \sum_{e=1}^{E} \nabla F_g(\widetilde{w}_{g,e}^t)\rangle]$$

$$\leq \mathbb{E}[2\eta_g \sum_{e=1}^{E} \sum_{i=1}^{K} p_i \sum_{r=0}^{E} \beta \langle \nabla F_g(\widetilde{w}_{g,e}^{t-1}), \nabla F_i(w_{i,r}^{\tau_i^{t-1}})\rangle] + \mathbb{E}[2||\eta_g \sum_{e=1}^{E} \nabla F_g(\widetilde{w}_{g,e}^{t-1})||^2]$$

$$\leq 2\beta^2 E^2 \delta^2.$$

(28)

For the 7th term:

$$2\langle -\sum_{i=1}^{K} p_i \sum_{e=1}^{E} \eta_i^{\tau_i^{t-1}} \sum_{r=1}^{e} (m_i^{\tau_i^{t-1}})^r \nabla F_i(w_{i,e-r}^{\tau_i^{t-1}}), \sum_{i=1}^{K} p_i \sum_{e=1}^{E} \eta_i^{\tau_i^{t-1}} \nabla F_i(w_{i,e}^{\tau_i^{t-1}}) + \eta_g \sum_{e=1}^{E} \nabla F_g(\widetilde{w}_{g,e}^t)\rangle$$

$$= -2\langle \sum_{i=1}^{K} p_i \sum_{e=1}^{E} \eta_i^{\tau_i^{t-1}} \sum_{r=1}^{e} (m_i^{\tau_i^{t-1}})^r \nabla F_i(w_{i,e-r}^{\tau_i^{t-1}}), \sum_{i=1}^{K} p_i \sum_{e=1}^{E} \eta_i^{\tau_i^{t-1}} \nabla F_i(w_{i,e}^{\tau_i^{t-1}})\rangle$$

$$- 2\langle \sum_{i=1}^{K} p_i \sum_{e=1}^{E} \eta_i^{\tau_i^{t-1}} \sum_{r=1}^{e} (m_i^{\tau_i^{t-1}})^r \nabla F_i(w_{i,e-r}^{\tau_i^{t-1}}), \eta_g \sum_{e=1}^{E} \nabla F_g(\widetilde{w}_{g,e}^t)\rangle$$

$$\leq -2 \sum_{i=1}^{K} p_i \sum_{l=0}^{E} \beta \sum_{r=1}^{e} \eta_g \sum_{e=1}^{E} (m_i^{\tau_i^{t-1}})^r \langle \nabla F_i(w_{i,e-r}^{\tau_i^{t-1}}), \nabla F_g(\widetilde{w}_{g,l}^t)\rangle$$

$$+ ||\eta_i^{\tau_i^{t-1}} \sum_{e=1}^{E} \sum_{r=1}^{e} (m_i^{\tau_i^{t-1}})^r \nabla F_i(w_{i,e-r}^{\tau_i^{t-1}})||^2 + ||\eta_i^{\tau_i^{t-1}} \sum_{e=1}^{E} \nabla F_i(w_{i,e}^{\tau_i^{t-1}})||^2.$$

(29)

Then we have:

$$
\mathbb{E}[2\langle -\sum_{i=1}^{K} p_i \sum_{e=1}^{E} \eta_i^{\tau_i^{t-1}} \sum_{r=1}^{e} (m_i^{\tau_i^{t-1}})^r \nabla F_i(w_{i,e-r}^{\tau_i^{t-1}}),
$$

$$
\sum_{i=1}^{K} p_i \sum_{e=1}^{E} \eta_i^{\tau_i^{t-1}} \nabla F_i(w_{i,e}^{\tau_i^{t-1}}) + \eta_g \sum_{e=1}^{E} \nabla F_g(\widetilde{w}_{g,e}^{t})\rangle]
$$

$$
\leq -2\sum_{i=1}^{K} p_i \sum_{l=0}^{E} \beta \sum_{r=1}^{e} \eta_g \sum_{e=1}^{E} (m_i^{\tau_i^{t-1}})^r \langle \nabla F_i(w_{i,e-r}^{\tau_i^{t-1}}), \nabla F_g(\widetilde{w}_{g,l}^{t})\rangle \tag{30}
$$

$$
+ \mathbb{E}[||\eta_i^{\tau_i^{t-1}} \sum_{e=1}^{E} \sum_{r=1}^{e} (m_i^{\tau_i^{t-1}})^r \nabla F_i(w_{i,e-r}^{\tau_i^{t-1}})||^2] + \mathbb{E}[||\eta_i^{\tau_i^{t-1}} \sum_{e=1}^{E} \nabla F_i(w_{i,e}^{\tau_i^{t-1}})||^2]
$$

$$
\leq \mathbb{E}[||\eta_i^{\tau_i^{t-1}} \sum_{e=1}^{E} \sum_{r=1}^{e} (m_i^{\tau_i^{t-1}})^r \nabla F_i(w_{i,e-r}^{\tau_i^{t-1}})||^2] + \mathbb{E}[||\eta_i^{\tau_i^{t-1}} \sum_{e=1}^{E} \nabla F_i(w_{i,e}^{\tau_i^{t-1}})||^2].
$$

To sum up, we have:

$$
\mathbb{E}[||\widetilde{w}_g^t - w^*||^2]
$$

$$
\leq \mathbb{E}[||\widetilde{w}_g^{t-1} - w^*||^2] + \mathbb{E}[||\eta_i^{\tau_i^{t-1}} \sum_{e=1}^{E} \sum_{r=1}^{e} (m_i^{\tau_i^{t-1}})^r \nabla F_i(w_{i,e-r}^{\tau_i^{t-1}})||^2] + \beta^2 E^2 \delta^2
$$

$$
+ (1 - \frac{2\beta^2 R}{\beta^2 + 1})\mathbb{E}[||\widetilde{w}_g^{t-1} - w^*||^2] + \beta^2 E^2 \delta^2 + \beta^2 E^2 \delta^2
$$

$$
+ \mathbb{E}[||\eta_i^{\tau_i^{t-1}} \sum_{e=1}^{E} \nabla F_i(w_{i,e}^{\tau_i^{t-1}})||^2] + \mathbb{E}[||\widetilde{w}_g^{t-1} - w^*||^2] + \beta^2 E^2 \delta^2 + \mathbb{E}[||\sum_{e=1}^{E} \nabla F_i(w_{i,e}^{\tau_i^{t-1}})||^2]
$$

$$
+ 2\beta^2 E^2 \delta^2 + \mathbb{E}[||\eta_i^{\tau_i^{t-1}} \sum_{e=1}^{E} \nabla F_i(w_{i,e}^{\tau_i^{t-1}})||^2] + \mathbb{E}[||\eta_i^{\tau_i^{t-1}} \sum_{e=1}^{E} \sum_{r=1}^{e} (m_i^{\tau_i^{t-1}})^r \nabla F_i(w_{i,e-r}^{\tau_i^{t-1}})||^2].
$$

$$(31)$$

Combining like terms:

$$
\mathbb{E}[||\widetilde{w}_g^t - w^*||^2] \leq (3 - \frac{2\beta^2 R}{\beta^2 + 1})\mathbb{E}[||\widetilde{w}_g^{t-1} - w^*||^2] + 6\beta^2 E^2 \delta^2
$$

$$
+ 3\mathbb{E}[||\eta_i^{\tau_i^{t-1}} \sum_{e=1}^{E} \sum_{r=1}^{e} (m_i^{\tau_i^{t-1}})^r \nabla F_i(w_{i,e-r}^{\tau_i^{t-1}})||^2] + 2\mathbb{E}[||\sum_{e=1}^{E} \nabla F_i(w_{i,e}^{\tau_i^{t-1}})||^2].
$$

$$(32)$$

Take summarizing with $\sqrt{\frac{1}{R-1}} < \beta < \sqrt{\frac{3}{2R-3}}$, we have:

$$
\mathbb{E}[||\widetilde{w}_g^t - w^*||^2] \leq \mathcal{V}^t \mathbb{E}[||\widetilde{w}_g^0 - w^*||^2] + \frac{6\beta^2(\beta^2 + 1)}{2\beta^2 R - 2\beta^2 - 2} E^2 \delta^2
$$

$$
+ 3\sum_{j=1}^{t} \mathcal{V}^j \mathbb{E}[||\eta_i^{\tau_i^{t-1}} \sum_{e=1}^{E} \sum_{r=1}^{e} (m_i^{\tau_i^{t-1}})^r \nabla F_i(w_{i,e-r}^{\tau_i^{t-1}})||^2] \tag{33}
$$

$$
+ 2\sum_{j=1}^{t} \mathcal{V}^j \mathbb{E}[||\sum_{e=1}^{E} \nabla F_i(w_{i,e}^{\tau_i^{t-1}})||^2],
$$

where $\mathcal{V} = (3 - \frac{2\beta^2 R}{\beta^2+1})$.

$\square$

**Lemma E.4** (global model one-step difference in gradient aggregation). *Given the real global model $w_g^t$ and its one-step nearby global model $w_g^{t-1}$, the expected square norm of the difference can be described below:*

$$||w_g^t - w_g^{t-1}||^2 \le 2||\eta_i^{\tau_i^t} \sum_{e=1}^{E} \nabla F_i(w_{i,e}^{\tau_i^t})||^2 + 2||\sum_{e=1}^{E} \eta_i^{\tau_i^t} \sum_{r=1}^{e} (m_i^t)^r \nabla F_i(w_{i,e-r}^{\tau_i^t})||^2. \quad (34)$$

**Proof.** From the Equation 3, we know:

$$w_g^t = w_g^{t-1} - \sum_{i=1}^{K} p_i \sum_{e=1}^{E} \eta_i^{\tau_i^t} [\sum_{r=1}^{e} (m_i^{\tau_i^t})^r \nabla F_i(w_{i,e-r}^{\tau_i^t}) + \nabla F_i(w_{i,e}^{\tau_i^t})]. \quad (35)$$

Therefore, we have:

$$w_g^t - w_g^{t-1} = - \sum_{i=1}^{K} p_i \sum_{e=1}^{E} \eta_i^{\tau_i^t} [\sum_{r=1}^{e} (m_i^{\tau_i^t})^r \nabla F_i(w_{i,e-r}^{\tau_i^t}) + \nabla F_i(w_{i,e}^{\tau_i^t})]. \quad (36)$$

Therefore, it's easy to get the following equation since we normalize $p_i$ such that $\sum_{i=1}^{K} p_i = 1$:

$$
\begin{aligned}
&||w_g^t - w_g^{t-1}||^2 \\
&= || - \sum_{i=1}^{K} p_i \sum_{e=1}^{E} \eta_i^{\tau_i^t} [\sum_{r=1}^{e} (m_i^{\tau_i^t})^r \nabla F_i(w_{i,e-r}^{\tau_i^t}) + \nabla F_i(w_{i,e}^{\tau_i^t})]||^2 \\
&= || \sum_{i=1}^{K} p_i \sum_{e=1}^{E} \eta_i^{\tau_i^t} \sum_{r=1}^{e} (m_i^{\tau_i^t})^r \nabla F_i(w_{i,e-r}^{\tau_i^t}) + \sum_{i=1}^{K} p_i \sum_{e=1}^{E} \eta_i^{\tau_i^t} \nabla F_i(w_{i,e}^{\tau_i^t})||^2 \\
&= || \sum_{i=1}^{K} p_i \sum_{e=1}^{E} \eta_i^{\tau_i^t} \nabla F_i(w_{i,e}^{\tau_i^t})||^2 + || \sum_{i=1}^{K} p_i \sum_{e=1}^{E} \eta_i^{\tau_i^t} \sum_{r=1}^{e} (m_i^t)^r \nabla F_i(w_{i,e-r}^{\tau_i^t})||^2 \\
&\quad + 2\langle \sum_{i=1}^{K} p_i \sum_{e=1}^{E} \eta_i^{\tau_i^t} \sum_{r=1}^{e} \frac{1}{e} \nabla F_i(w_{i,e}^{\tau_i^t}), \sum_{i=1}^{K} p_i \sum_{e=1}^{E} \eta_i^{\tau_i^t} \sum_{r=1}^{e} (m_i^t)^r \nabla F_i(w_{i,e-r}^{\tau_i^t}) \rangle \\
&\le ||\eta_i^{\tau_i^t} \sum_{e=1}^{E} \nabla F_i(w_{i,e}^{\tau_i^t})||^2 + || \sum_{e=1}^{E} \eta_i^{\tau_i^t} \sum_{r=1}^{e} (m_i^t)^r \nabla F_i(w_{i,e-r}^{\tau_i^t})||^2 \\
&\quad + 2\beta^2 \sum_{i=1}^{K} p_i \sum_{j=1}^{K} p_j \sum_{e=1}^{E} \sum_{r=1}^{e} (m_j^t)^r \langle \nabla F_i(w_{i,e}^{\tau_i^t}), \nabla F_j(w_{j,e-r}^{\tau_j^t}) \rangle \\
&\le 2||\eta_i^{\tau_i^t} \sum_{e=1}^{E} \nabla F_i(w_{i,e}^{\tau_i^t})||^2 + 2|| \sum_{e=1}^{E} \eta_i^{\tau_i^t} \sum_{r=1}^{e} (m_i^t)^r \nabla F_i(w_{i,e-r}^{\tau_i^t})||^2.
\end{aligned}
\quad (37)
$$

$\square$

Then, based on the Lemma E.3 and E.4, we can easily proof the Theorem A.4 as following:

**Proof.** Based on Assumption 1, we have:

$$F(w_g^t) - F^* \le \langle \nabla F^*, F(w_g^t) \rangle + \frac{L}{2}||w_g^t - w^*||^2. \tag{38}$$

Since $F^*$ is the global optima, we have $\nabla F^* = 0$, therefore:

$$
\begin{aligned}
&F(w_g^t) - F^* \\
&\le \frac{L}{2}||w_g^t - w^*||^2 \\
&\le \frac{L}{2}||w_g^t - \widetilde{w}_g^t - (\widetilde{w}_g^t - w^*)||^2 \\
&\le \frac{L}{2}||w_g^t - w_g^{t-1} + \eta_g \sum_{e=1}^{E} \nabla F_g(\widetilde{w}_{g,e}^t) - (\widetilde{w}_g^t - w^*)||^2 \\
&\le 2L[||w_g^t - w_g^{t-1}||^2 + 2L||\eta_g \sum_{e=1}^{E} \nabla F_g(\widetilde{w}_{g,e}^t)||^2 + L||\widetilde{w}_g^t - w^*||^2.
\end{aligned}
\tag{39}
$$

Then, based on Lemma E.1, E.3 and E.4, we have:

$$
\begin{aligned}
\mathbb{E}[F(w_g^t)] - F^* &\le 2L[\mathbb{E}[||w_g^t - w_g^{t-1}||^2] + 2L\mathbb{E}[||\eta_g \sum_{e=1}^{E} \nabla F_g(\widetilde{w}_{g,e}^t)||^2] + L\mathbb{E}[||\widetilde{w}_g^t - w^*||^2] \\
&\le L\mathcal{V}^t \mathbb{E}[||w_g^0 - w^*||^2] + \mathcal{U} + \mathcal{W},
\end{aligned}
\tag{40}
$$

where $\mathcal{U} = [2L\beta^2 + \frac{6\beta^2(\beta^2 L + L)}{2\beta^2 R - 2\beta^2 - 2}]E^2\delta^2$, $\mathcal{V} = (3 - \frac{2\beta^2 KR}{\beta^2 + 1})$, and

$$
\begin{aligned}
\mathcal{W} &= 4L\mathbb{E}[||\eta_i^{\tau_i^t} \sum_{e=1}^{E} \nabla F_{i,e}(w_{i,e-1}^{\tau_i^t})||^2] + 4L\mathbb{E}[||\eta_i^{\tau_i^t} \sum_{e=1}^{E}\sum_{r=1}^{e}(m_i^t)^r \nabla F_{i,e-r}(w_{i,e-r-1}^{\tau_i^t})||^2] \\
&+ \sum_{j=0}^{t} \mathcal{V}^j 3L\mathbb{E}[||\eta_i^{\tau_i^{t-1}} \sum_{e=1}^{E} \nabla F_{i,e}(w_{i,e-1}^{\tau_i^{t-1}})||^2] \\
&+ \sum_{j=0}^{t} \mathcal{V}^j 2L\mathbb{E}[||\eta_i^{\tau_i^{t-1}} \sum_{e=1}^{E}\sum_{r=1}^{e}(m_i^{t-1})^r \nabla F_{i,e-r}(w_{i,e-r-1}^{\tau_i^{t-1}})||^2].
\end{aligned}
\tag{41}
$$

Based on Assumption A.2, we can further bound the gradient expectation term $\mathcal{W}$ by:

$$
\begin{aligned}
\mathcal{W} &\le 4L\beta^2 E^2 G_c^2 + 4L\beta^2 RQ(t)G_c^2 + \sum_{j=0}^{t} \mathcal{V}^j 3L\beta^2 E^2 G_c^2 + \sum_{j=0}^{t} \mathcal{V}^j 2L\beta^2 RQ(t)G_c^2 \\
&\le [4LE^2 + 4LRQ(t) + \frac{(\beta^2 L + L)(2RQ(t) + 3E^2)}{2\beta^2 R - 2\beta^2 - 2}]\beta^2 G_c^2.
\end{aligned}
\tag{42}
$$

$\square$

## E.2 Proof of Theorem A.5

Denoted $Q(t)$ as the maximum number of clients that execute the momentum update at global round $t$ and $0 \le q \le p_i \le p \le 1$, we begin the proof by two lemmas:

**Lemma E.5** (Ideal global model difference). *Given* $\mathcal{V} = 3 - \frac{2\beta^2(R+E^2)}{\beta^2+1}$ *and the ideal global model trained in global epoch $t$, the difference between the optimal can be bounded by:*

$$
\begin{aligned}
\mathbb{E}[||\widetilde{w}_g^t - w^*||^2] \leq{} & \mathcal{V}^t \mathbb{E}[||w_g^0 - w^*||^2] \\
& + \frac{1-\mathcal{V}^t}{1-\mathcal{V}} 8\beta^2 E^2 \delta^2 \\
& + \sum_{j=1}^{t} \mathcal{V}^j \mathbb{E}[||\eta_i^{t-1} \sum_{e=1}^{E}[\nabla F_i(w_{i,e}^{t-1})]||^2] \\
& + \sum_{j=1}^{t} \mathcal{V}^j \mathbb{E}[||\eta_i^{t-1} \sum_{e=1}^{E}\sum_{r=1}^{e}(m_i^{t-1})^r \nabla F_i(w_{i,e-r}^{t-1})||^2].
\end{aligned}
\tag{43}
$$

**Proof.** Based on the momentum update equation 3, we have:

$$
\begin{aligned}
w_i^{\tau_i^t} &= w_{i,0}^{\tau_i^t} - \eta_i^{\tau_i^t} \sum_{e=1}^{E}\sum_{r=1}^{e}[(m_i^{\tau_i^t})^r \nabla F_i(w_{i,e-r}^{\tau_i^t}) + \nabla F_i(w_{i,e}^{\tau_i^t})] \\
&= \widetilde{w}_g^{\tau_i^t} - \eta_i^{\tau_i^t} \sum_{e=1}^{E}\sum_{r=1}^{e}[(m_i^{\tau_i^t})^r \nabla F_i(w_{i,e-r}^{\tau_i^t}) + \nabla F_i(w_{i,e}^{\tau_i^t})] + \eta_g \sum_{e=1}^{E} \nabla F_g(\widetilde{w}_{g,e}^{\tau_i^t}).
\end{aligned}
\tag{44}
$$

We add a zero term:

$$
\begin{aligned}
& ||\widetilde{w}_g^t - w^*||^2 \\
={} & ||w_g^{t-1} - \eta_g \sum_{e=1}^{E} \nabla F_g(\widetilde{w}_{g,e}^t) - w^*||^2 \\
={} & ||\sum_{i=1}^{K} p_i w_i^{\tau_i^{t-1}} - \eta_g \sum_{e=1}^{E} \nabla F_g(\widetilde{w}_{g,e}^t) - w^*||^2 \\
={} & ||\sum_{i=1}^{K} p_i(\widetilde{w}_g^{\tau_i^{t-1}} - \eta_i^{t-1} \sum_{e=1}^{E}\sum_{r=1}^{e}[(m_i^{t-1})^r \nabla F_i(w_{i,e-r}^{t-1}) + \nabla F_i(w_{i,e}^{t-1})] \\
& + \eta_g \sum_{e=1}^{E} \nabla F_g(\widetilde{w}_{g,e}^{\tau_i^{t-1}})) - \eta_g \sum_{e=1}^{E} \nabla F_g(\widetilde{w}_{g,e}^t) - w^*||^2 \\
={} & ||\sum_{i=1}^{K} p_i(\widetilde{w}_g^{\tau_i^{t-1}} - \eta_i^{t-1} \sum_{e=1}^{E}\sum_{r=1}^{e}[(m_i^{t-1})^r \nabla F_i(w_{i,e-r}^{t-1}) + \nabla F_i(w_{i,e}^{t-1})]) \\
& + \sum_{i=1}^{K} p_i(\eta_g \sum_{e=1}^{E} \nabla F_g(\widetilde{w}_{g,e}^{\tau_i^{t-1}}) - \eta_g \sum_{e=1}^{E} \nabla F_g(\widetilde{w}_{g,e}^t)) - \sum_{i=1}^{K} p_i w^*||^2.
\end{aligned}
\tag{45}
$$

Therefore,

$$||\widetilde{w}_g^t - w^*||^2$$

$$= ||\sum_{i=1}^{K} p_i(\widetilde{w}_g^{\tau_i^{t-1}} - w^*)||^2 + ||\sum_{i=1}^{K} p_i\eta_i^{t-1}\sum_{e=1}^{E}[\sum_{r=1}^{e}(m_i^{t-1})^r\nabla F_i(w_{i,e-r}^{t-1}) + \nabla F_i(w_{i,e}^{t-1})]||^2$$

$$- 2\langle\sum_{i=1}^{K} p_i(\widetilde{w}_g^{\tau_i^{t-1}} - w^*), \sum_{i=1}^{K} p_i\eta_i^{t-1}\sum_{e=1}^{E}[\sum_{r=1}^{e}(m_i^{t-1})^r\nabla F_i(w_{i,e-r}^{t-1}) + \nabla F_i(w_{i,e}^{t-1})]\rangle$$

$$+ ||\sum_{i=1}^{K} p_i(\eta_g\sum_{e=1}^{E}\nabla F_g(\widetilde{w}_{g,e}^{\tau_i^{t-1}}) - \eta_g\sum_{e=1}^{E}\nabla F_g(\widetilde{w}_{g,e}^t))||^2$$

$$+ 2\langle\sum_{i=1}^{K} p_i(\widetilde{w}_g^{\tau_i^{t-1}} - w^*), \sum_{i=1}^{K} p_i\eta_g\sum_{e=1}^{E}\nabla F_g(\widetilde{w}_{g,e}^{\tau_i^{t-1}})\rangle \qquad (46)$$

$$- 2\langle\sum_{i=1}^{K} p_i(\widetilde{w}_g^{\tau_i^{t-1}} - w^*), \sum_{i=1}^{K} p_i\eta_g\sum_{e=1}^{E}\nabla F_g(\widetilde{w}_{g,e}^t)\rangle$$

$$- \langle\sum_{i=1}^{K} p_i\eta_i^{t-1}\sum_{e=1}^{E}[\sum_{r=1}^{e}(m_i^{t-1})^r\nabla F_i(w_{i,e-r}^{t-1}) + \nabla F_i(w_{i,e}^{t-1})],$$

$$\sum_{i=1}^{K} p_i(\eta_g\sum_{e=1}^{E}\nabla F_g(\widetilde{w}_{g,e}^{\tau_i^{t-1}}) - \eta_g\sum_{e=1}^{E}\nabla F_g(\widetilde{w}_{g,e}^t))\rangle.$$

For the 2nd term, we bound it by the AM-GM inequality:

$$||\sum_{i=1}^{K} p_i\eta_i^{t-1}\sum_{e=1}^{E}[\sum_{r=1}^{e}(m_i^{t-1})^r\nabla F_i(w_{i,e-r}^{t-1}) + \nabla F_i(w_{i,e}^{t-1})]||^2$$

$$\leq ||\eta_i^{t-1}\sum_{e=1}^{E}\sum_{r=1}^{e}(m_i^{t-1})^r\nabla F_i(w_{i,e-r}^{t-1})||^2 + ||\eta_i^{t-1}\sum_{e=1}^{E}[\nabla F_i(w_{i,e}^{t-1})]||^2. \qquad (47)$$

Therefore,

$$\mathbb{E}[||\sum_{i=1}^{K} p_i\eta_i^{t-1}\sum_{e=1}^{E}[\sum_{r=1}^{e}(m_i^{t-1})^r\nabla F_i(w_{i,e-r}^{t-1}) + \nabla F_i(w_{i,e}^{t-1})]||^2]$$

$$\leq \mathbb{E}[||\eta_i^{t-1}\sum_{e=1}^{E}\sum_{r=1}^{e}(m_i^{t-1})^r\nabla F_i(w_{i,e-r}^{t-1})||^2] + \mathbb{E}[||\eta_i^{t-1}\sum_{e=1}^{E}[\nabla F_i(w_{i,e}^{t-1})]||^2]. \qquad (48)$$

For the 3rd term, using a variant of reversed Cauchy-Schwarz inequality [48, 49, 50] with $\gamma = \beta$ and $\Gamma = \frac{1}{\beta}$ and $m_i^{\tau_i^{t-1}} \leq 1$, we have:

$$-2\langle\sum_{i=1}^{K}p_i(\widetilde{w}_g^{\tau_i^{t-1}}-w^*),\sum_{i=1}^{K}p_i\eta_i^{t-1}\sum_{e=1}^{E}[\sum_{r=1}^{e}(m_i^{t-1})^r\nabla F_i(w_{i,e-r}^{t-1})+\nabla F_i(w_{i,e}^{t-1})]\rangle$$

$$=-2\langle\sum_{i=1}^{K}p_i(\widetilde{w}_g^{\tau_i^{t-1}}-w^*),\sum_{i=1}^{K}p_i\eta_i^{t-1}\sum_{e=1}^{E}[\sum_{r=1}^{e}(m_i^{t-1})^r\nabla F_i(w_{i,e-r}^{t-1})]\rangle$$

$$-2\langle\sum_{i=1}^{K}p_i(\widetilde{w}_g^{\tau_i^{t-1}}-w^*),\sum_{i=1}^{K}p_i\eta_i^{t-1}\sum_{e=1}^{E}\nabla F_i(w_{i,e}^{t-1})\rangle \tag{49}$$

$$\leq-\frac{2\beta^2 R}{\beta^2+1}||\sum_{i=1}^{K}p_i(\widetilde{w}_g^{\tau_i^{t-1}}-w^*)||^2-p^2\frac{2\beta^2 E^2}{\beta^2+1}||\sum_{i=1}^{K}(\widetilde{w}_g^{\tau_i^{t-1}}-w^*)||^2$$

$$=-\frac{2\beta^2(R+E^2)}{\beta^2+1}||\sum_{i=1}^{K}p_i(\widetilde{w}_g^{\tau_i^{t-1}}-w^*)||^2.$$

For the 4th term, using AM-GM inequality:

$$||\sum_{i=1}^{K}p_i(\eta_g\sum_{e=1}^{E}\nabla F_g(\widetilde{w}_{g,e}^{\tau_i^{t-1}})-\eta_g\sum_{e=1}^{E}\nabla F_g(\widetilde{w}_{g,e}^{t}))||^2$$

$$\leq 2||\sum_{i=1}^{K}p_i(\eta_g\sum_{e=1}^{E}\nabla F_g(\widetilde{w}_{g,e}^{\tau_i^{t-1}})||^2+2||\eta_g\sum_{e=1}^{E}\nabla F_g(\widetilde{w}_{g,e}^{t}))||^2 \tag{50}$$

$$\leq 2\eta_g^2 E^2||\nabla F_g(\widetilde{w}_{g,e}^{\tau_i^{t-1}})||^2+2\eta_g^2 E^2||\nabla F_g(\widetilde{w}_{g,e}^{t}))||^2.$$

Therefore:

$$\mathbb{E}[||\sum_{i=1}^{K}p_i(\eta_g\sum_{e=1}^{E}\nabla F_g(\widetilde{w}_{g,e}^{\tau_i^{t-1}})-\eta_g\sum_{e=1}^{E}\nabla F_g(\widetilde{w}_{g,e}^{t}))||^2]$$

$$\leq 2\eta_g^2 E^2\mathbb{E}[||\nabla F_g(\widetilde{w}_{g,e}^{\tau_i^{t-1}})||^2]+2\eta_g^2 E^2\mathbb{E}[||\nabla F_g(\widetilde{w}_{g,e}^{t}))||^2] \tag{51}$$

$$\leq 4\beta^2 E^2\delta^2.$$

For the 5th term:

$$2\langle\sum_{i=1}^{K}p_i(\widetilde{w}_g^{\tau_i^{t-1}}-w^*),\sum_{i=1}^{K}p_i(\eta_g\sum_{e=1}^{E}\nabla F_g(\widetilde{w}_{g,e}^{\tau_i^{t-1}}))\rangle$$

$$\leq||\sum_{i=1}^{K}p_i(\widetilde{w}_g^{\tau_i^{t-1}}-w^*)||^2+||\sum_{i=1}^{K}p_i(\eta_g\sum_{e=1}^{E}\nabla F_g(\widetilde{w}_{g,e}^{\tau_i^{t-1}}))||^2 \tag{52}$$

$$\leq||\sum_{i=1}^{K}p_i(\widetilde{w}_g^{\tau_i^{t-1}}-w^*)||^2+2\eta_g^2 E^2||\nabla F_g(\widetilde{w}_{g,e}^{\tau_i^{t-1}})||^2.$$

Therefore,

$$\mathbb{E}[2\langle \sum_{i=1}^{K} p_i(\widetilde{w}_g^{\tau_i^{t-1}} - w^*), \sum_{i=1}^{K} p_i(\eta_g \sum_{e=1}^{E} \nabla F_g(\widetilde{w}_{g,e}^{\tau_i^{t-1}}))\rangle]$$

$$\leq \mathbb{E}[|| \sum_{i=1}^{K} p_i(\widetilde{w}_g^{\tau_i^{t-1}} - w^*)||^2] + 2\eta_g^2 E^2 \mathbb{E}[||\nabla F_g(\widetilde{w}_{g,e}^{\tau_i^{t-1}})||^2] \quad (53)$$

$$\leq \mathbb{E}[|| \sum_{i=1}^{K} p_i(\widetilde{w}_g^{\tau_i^{t-1}} - w^*)||^2] + 2\beta^2 E^2 \delta^2.$$

The 6th term is as same as the 5th term:

$$2\mathbb{E}[\langle \sum_{i=1}^{K} p_i(\widetilde{w}_g^{\tau_i^{t-1}} - w^*), - \sum_{i=1}^{K} p_i(\eta_g \sum_{e=1}^{E} \nabla F_g(\widetilde{w}_{g,e}^t))\rangle]$$

$$\leq \mathbb{E}[|| \sum_{i=1}^{K} p_i(\widetilde{w}_g^{\tau_i^{t-1}} - w^*)||^2] + 2\eta_g^2 E^2 \mathbb{E}[||\nabla F_g(\widetilde{w}_{g,e}^t)||^2] \quad (54)$$

$$\leq \mathbb{E}[|| \sum_{i=1}^{K} p_i(\widetilde{w}_g^{\tau_i^{t-1}} - w^*)||^2] + 2\beta^2 E^2 \delta^2.$$

For the 7th term:

$$2\langle - \sum_{i=1}^{K} p_i(\eta_i^{t-1} \sum_{e=1}^{E} [\sum_{r=1}^{e} (m_i^{t-1})^r \nabla F_i(w_{i,e-r}^{t-1}) + \nabla F_i(w_{i,e}^{t-1})],$$

$$\sum_{i=1}^{K} p_i(\eta_g \sum_{e=1}^{E} \nabla F_g(\widetilde{w}_{g,e}^{\tau_i^{t-1}}) - \eta_g \sum_{e=1}^{E} \nabla F_g(\widetilde{w}_{g,e}^t))\rangle$$

$$= 2\langle - \sum_{i=1}^{K} p_i(\eta_i^{t-1} \sum_{e=1}^{E} [\sum_{r=1}^{e} (m_i^{t-1})^r \nabla F_i(w_{i,e-r}^{t-1}) + \nabla F_i(w_{i,e}^{t-1})], \sum_{i=1}^{K} p_i(\eta_g \sum_{e=1}^{E} \nabla F_g(\widetilde{w}_{g,e}^{\tau_i^{t-1}}))\rangle$$

$$- 2\langle - \sum_{i=1}^{K} p_i(\eta_i^{t-1} \sum_{e=1}^{E} [\sum_{r=1}^{e} (m_i^{t-1})^r \nabla F_i(w_{i,e-r}^{t-1}) + \nabla F_i(w_{i,e}^{t-1})], \eta_g \sum_{e=1}^{E} \nabla F_g(\widetilde{w}_{g,e}^t))\rangle.$$
$$\quad (55)$$

Therefore, we can easily get:

$$\mathbb{E}[2\langle - \sum_{i=1}^{K} p_i(\eta_i^{t-1} \sum_{e=1}^{E} [\sum_{r=1}^{e} (m_i^{t-1})^r \nabla F_i(w_{i,e-r}^{t-1}) + \nabla F_i(w_{i,e}^{t-1})],$$

$$\sum_{i=1}^{K} p_i(\eta_g \sum_{e=1}^{E} \nabla F_g(\widetilde{w}_{g,e}^{\tau_i^{t-1}}) - \eta_g \sum_{e=1}^{E} \nabla F_g(\widetilde{w}_{g,e}^t))\rangle] = 0. \quad (56)$$

To sum up, we have:

$$\mathbb{E}[||\widetilde{w}_g^t - w^*||^2]$$

$$\leq \mathbb{E}[||\sum_{i=1}^{K} p_i(\widetilde{w}_g^{\tau_i^{t-1}} - w^*)||^2] + \mathbb{E}[||\eta_i^{t-1} \sum_{e=1}^{E} [\nabla F_i(w_{i,e}^{t-1})]||^2]$$

$$+ \mathbb{E}[||\eta_i^{t-1} \sum_{e=1}^{E} \sum_{r=1}^{e} (m_i^{t-1})^r \nabla F_i(w_{i,e-r}^{t-1})||^2] + 4\beta^2 E^2 \delta^2 + \mathbb{E}[||\sum_{i=1}^{K} p_i(\widetilde{w}_g^{\tau_i^{t-1}} - w^*)||^2] \quad (57)$$

$$+ 2\beta^2 E^2 \delta^2 + \mathbb{E}[||\sum_{i=1}^{K} p_i(\widetilde{w}_g^{\tau_i^{t-1}} - w^*)||^2] + 2\beta^2 E^2 \delta^2$$

$$- \frac{2\beta^2(R+E^2)}{\beta^2+1} \mathbb{E}[||\sum_{i=1}^{K} p_i(\widetilde{w}_g^{\tau_i^{t-1}} - w^*)||^2].$$

Combining like terms:

$$\mathbb{E}[||\widetilde{w}_g^t - w^*||^2] \leq (3 - \frac{2\beta^2(R+E^2)}{\beta^2+1}) \mathbb{E}[||\sum_{i=1}^{K} p_i(\widetilde{w}_g^{\tau_i^{t-1}} - w^*)||^2] + 8\beta^2 E^2 \delta^2$$

$$+ \mathbb{E}[||\eta_i^{t-1} \sum_{e=1}^{E} [\nabla F_i(w_{i,e}^{t-1})]||^2] + \mathbb{E}[||\eta_i^{t-1} \sum_{e=1}^{E} \sum_{r=1}^{e} (m_i^{t-1})^r \nabla F_i(w_{i,e-r}^{t-1})||^2]. \quad (58)$$

Although the sequence $\{\mathbb{E}[||\widetilde{w}_g^t - w^*||^2]\}_t$ is not monotonically increasing, the tracing process from $t$ to 0 is less than $t$ steps. Therefore, we can accumulate it for $t$ steps and still maintain the inequality. Then, take summarizing with $\sqrt{\frac{1}{KR+E^2-1}} < \beta < \sqrt{\frac{3}{2RK+2E^2-3}}$ step by step, we have:

$$\mathbb{E}[||\widetilde{w}_g^t - w^*||^2]$$

$$\leq \mathcal{V}^t \mathbb{E}[||w_g^0 - w^*||^2] + \sum_{j=1}^{t} (\sum_{i=1}^{K} p_i \mathcal{V}^j) 8\beta^2 E^2 \delta^2 + \sum_{j=1}^{t} (\sum_{i=1}^{K} p_i \mathcal{V}^j) \mathbb{E}[||\eta_i^{t-1} \sum_{e=1}^{E} [\nabla F_i(w_{i,e}^{t-1})]||^2]$$

$$+ \sum_{j=1}^{t} (\sum_{i=1}^{K} p_i \mathcal{V}^j) \mathbb{E}[||\eta_i^{t-1} \sum_{e=1}^{E} \sum_{r=1}^{e} (m_i^{t-1})^r \nabla F_i(w_{i,e-r}^{t-1})||^2]$$

$$\leq \mathcal{V}^t \mathbb{E}[||w_g^0 - w^*||^2] + \frac{1-\mathcal{V}^t}{1-\mathcal{V}} 8\beta^2 E^2 \delta^2 + \sum_{j=1}^{t} \mathcal{V}^j \mathbb{E}[||\eta_i^{t-1} \sum_{e=1}^{E} [\nabla F_i(w_{i,e}^{t-1})]||^2]$$

$$+ \sum_{j=1}^{t} \mathcal{V}^j \mathbb{E}[||\eta_i^{t-1} \sum_{e=1}^{E} \sum_{r=1}^{e} (m_i^{t-1})^r \nabla F_i(w_{i,e-r}^{t-1})||^2], \quad (59)$$

where $\mathcal{V} = 3 - \frac{2\beta^2(R+E^2)}{\beta^2+1}$. $\qquad\qquad\square$

**Lemma E.6** (global model and ideal model difference in model aggregation). *Given the real global model $w_g^t$ and its ideal global model $\widetilde{w}_g^t$ at the same global epoch $t$, the expected square norm of the difference can be described below:*

$$\mathbb{E}[||\widetilde{w}_g^t - w_g^t||^2] \leq p^2 K (3\mathbb{E}[||\widetilde{w}_g^t - w^*||^2] + 3\beta^2 E^2 \delta^2$$

$$+ \mathbb{E}[||\eta_i^t \sum_{e=1}^{E} \nabla F_i(w_{i,e}^t)||^2] + \mathbb{E}[||\eta_i \sum_{e=1}^{E} \nabla F_i(w_{i,e}^{\tau_i^t})||^2]) \tag{60}$$

$$+ 3\mathbb{E}[||\eta_i^t \sum_{e=1}^{E} \sum_{r=1}^{e} (m_i^t)^r \nabla F_i(w_{i,e-r}^t)||^2].$$

**Proof.** We have that:

$$||\widetilde{w}_g^t - w_g^t||^2 = ||\widetilde{w}_g^t - \sum_{i=1}^{K} p_i w_i^{\tau_i^t}||^2 = ||\sum_{i=1}^{K} p_i (\widetilde{w}_g^t - w_i^{\tau_i^t})||^2 p^2 K ||(\widetilde{w}_g^t - w_i^{\tau_i^t})||^2. \tag{61}$$

Based on Equation 44, we have:

$$||\widetilde{w}_g^t - w_i^{\tau_i^t}||^2$$

$$= ||\widetilde{w}_g^t - \widetilde{w}_g^{\tau_i^t} - \eta_g \sum_{e=1}^{E} \nabla F_g(\widetilde{w}_{g,e}^{\tau_i^t}) + \eta_i^t \sum_{e=1}^{E} [\sum_{r=1}^{e} (m_i^t)^r \nabla F_i(w_{i,e-r}^t) + \nabla F_i(w_{i,e}^t)]||^2$$

$$= ||\widetilde{w}_g^t - \widetilde{w}_g^{\tau_i^t} + \eta_i^t \sum_{e=1}^{E} \sum_{r=1}^{e} (m_i^t)^r \nabla F_i(w_{i,e-r}^t)||^2 + ||\eta_i \sum_{e=1}^{E} \nabla F_i(w_{i,e}^{\tau_i^t}) - \eta_g \sum_{e=1}^{E} \nabla F_g(\widetilde{w}_{g,e}^{\tau_i^t})||^2$$

$$+ 2\langle \widetilde{w}_g^t - \widetilde{w}_g^{\tau_i^t} + \eta_i^t \sum_{e=1}^{E} \sum_{r=1}^{e} (m_i^t)^r \nabla F_i(w_{i,e-r}^t), \eta_i \sum_{e=1}^{E} \nabla F_i(w_{i,e}^{\tau_i^t}) - \eta_g \sum_{e=1}^{E} \nabla F_g(\widetilde{w}_{g,e}^{\tau_i^t}) \rangle$$

$$\leq ||\widetilde{w}_g^t - w^*||^2 + ||\widetilde{w}_g^{\tau_i^t} - w^*||^2 + || - \eta_g \sum_{e=1}^{E} \nabla F_g(\widetilde{w}_{g,e}^{\tau_i^t})||^2 + ||\eta_i^t \sum_{e=1}^{E} \sum_{r=1}^{e} (m_i^t)^r \nabla F_i(w_{i,e-r}^t)||^2$$

$$+ ||\eta_i^t \sum_{e=1}^{E} \nabla F_i(w_{i,e}^t)||^2 + 2\langle \widetilde{w}_g^t - w^*, \eta_i^t \sum_{e=1}^{E} \sum_{r=1}^{e} (m_i^t)^r \nabla F_i(w_{i,e-r}^t) \rangle$$

$$- 2\langle \widetilde{w}_g^{\tau_i^t} - w^*, \eta_i^t \sum_{e=1}^{E} \sum_{r=1}^{e} (m_i^t)^r \nabla F_i(w_{i,e-r}^t) \rangle$$

$$+ 2\langle \widetilde{w}_g^t - w^*, \eta_i \sum_{e=1}^{E} \nabla F_i(w_{i,e}^{\tau_i^t}) - \eta_g \sum_{e=1}^{E} \nabla F_g(\widetilde{w}_{g,e}^{\tau_i^t}) \rangle$$

$$- 2\langle \widetilde{w}_g^{\tau_i^t} - w^*, \eta_i \sum_{e=1}^{E} \nabla F_i(w_{i,e}^{\tau_i^t}) - \eta_g \sum_{e=1}^{E} \nabla F_g(\widetilde{w}_{g,e}^{\tau_i^t}) \rangle$$

$$+ 2\langle \eta_i^t \sum_{e=1}^{E} \sum_{r=1}^{e} (m_i^t)^r \nabla F_i(w_{i,e-r}^t), \eta_i \sum_{e=1}^{E} \nabla F_i(w_{i,e}^{\tau_i^t}) \rangle$$

$$- 2\langle \eta_i^t \sum_{e=1}^{E} \sum_{r=1}^{e} (m_i^t)^r \nabla F_i(w_{i,e-r}^t), \eta_g \sum_{e=1}^{E} \nabla F_g(\widetilde{w}_{g,e}^{\tau_i^t}) \rangle. \tag{62}$$

For the 3th term:

$$|| - \eta_g \sum_{e=1}^{E} \nabla F_g(\widetilde{w}_{g,e}^{\tau_i^t})||^2 \leq \eta_g^2 E^2 ||\nabla F_g(\widetilde{w}_{g,e}^{\tau_i^t})||^2. \tag{63}$$

Therefore, we have:

$$\mathbb{E}[||-\eta_g \sum_{e=1}^{E} \nabla F_g(\widetilde{w}_{g,e}^{\tau_i^t})||^2] \leq \eta_g^2 E^2 \mathbb{E}[||\nabla F_g(\widetilde{w}_{g,e}^{\tau_i^t})||^2] \leq \beta^2 E^2 \delta^2. \tag{64}$$

For the 6th term, using AM-GM inequality and Cauchy-Schwarz inequality, we have:

$$\mathbb{E}[2\langle \widetilde{w}_g^t - w^*, \eta_i^t \sum_{e=1}^{E} \sum_{r=1}^{e} (m_i^t)^r \nabla F_i(w_{i,e-r}^t)\rangle]$$

$$\leq \mathbb{E}[||\widetilde{w}_g^t - w^*||^2] + \mathbb{E}[||\eta_i^t \sum_{e=1}^{E} \sum_{r=1}^{e} (m_i^t)^r \nabla F_i(w_{i,e-r}^t)||^2]. \tag{65}$$

Similarly, the 8th term satisfies:

$$\mathbb{E}[2\langle \widetilde{w}_g^t - w^*, \eta_i \sum_{e=1}^{E} \nabla F_i(w_{i,e}^{\tau_i^t}) - \eta_g \sum_{e=1}^{E} \nabla F_g(\widetilde{w}_{g,e}^{\tau_i^t})\rangle]$$

$$\leq \mathbb{E}[||\widetilde{w}_g^t - w^*||^2] + \mathbb{E}[||\eta_i \sum_{e=1}^{E} \nabla F_i(w_{i,e}^{\tau_i^t}) - \eta_g \sum_{e=1}^{E} \nabla F_g(\widetilde{w}_{g,e}^{\tau_i^t})||^2] \tag{66}$$

$$\leq \mathbb{E}[||\widetilde{w}_g^t - w^*||^2] + \beta^2 E^2 \mathbb{E}[||\nabla F_i(w_{i,e}^{\tau_i^t}) - \nabla F_g(\widetilde{w}_{g,e}^{\tau_i^t})||^2]$$

$$\leq \mathbb{E}[||\widetilde{w}_g^t - w^*||^2] + \beta^2 E^2 \delta^2.$$

For the 7th term, based on the reversed Cauchy-Schwarz inequality [48, 49, 50] with $\gamma = \beta$ and $\Gamma = \frac{1}{\beta}$ and $m_i^{\tau_i^{t-1}} \leq 1$:

$$\mathbb{E}[-2\langle \widetilde{w}_g^{\tau_i^t} - w^*, \eta_i^t \sum_{e=1}^{E} \sum_{r=1}^{e} (m_i^t)^r \nabla F_i(w_{i,e-r}^t)\rangle] \leq -\frac{2\beta^2 R}{\beta^2 + 1} \mathbb{E}[||\widetilde{w}_g^{\tau_i^t} - w^*||^2]. \tag{67}$$

Similarly, the 9th term satisfies:

$$\mathbb{E}[-2\langle \widetilde{w}_g^{\tau_i^t} - w^*, \eta_i \sum_{e=1}^{E} \nabla F_i(w_{i,e}^{\tau_i^t}) - \eta_g \sum_{e=1}^{E} \nabla F_g(\widetilde{w}_{g,e}^{\tau_i^t})\rangle]$$

$$= -2\mathbb{E}[\langle \widetilde{w}_g^{\tau_i^t} - w^*, \eta_i \sum_{e=1}^{E} \nabla F_i(w_{i,e}^{\tau_i^t})] + 2\mathbb{E}[\langle \widetilde{w}_g^{\tau_i^t} - w^*, \eta_g \sum_{e=1}^{E} \nabla F_g(\widetilde{w}_{g,e}^{\tau_i^t})\rangle] \tag{68}$$

$$\leq -\frac{2\beta^2 E^2}{\beta^2 + 1} \mathbb{E}[||\widetilde{w}_g^{\tau_i^t} - w^*||^2] + \mathbb{E}[||\widetilde{w}_g^{\tau_i^t} - w^*||^2] + \beta^2 E^2 \mathbb{E}[||\nabla F_g(\widetilde{w}_{g,e}^{\tau_i^t})||^2]$$

$$\leq -(\frac{2\beta^2 E^2}{\beta^2 + 1} - 1)\mathbb{E}[||\widetilde{w}_g^{\tau_i^t} - w^*||^2] + \beta^2 E^2 \delta^2.$$

For the 10th term, we use AM-GM inequality and Cauchy-Schwarz inequality to bound it:

$$\mathbb{E}[2\langle \eta_i^t \sum_{e=1}^{E} \sum_{r=1}^{e} (m_i^t)^r \nabla F_i(w_{i,e-r}^t), \eta_i \sum_{e=1}^{E} \nabla F_i(w_{i,e}^{\tau_i^t})\rangle]$$

$$\leq \mathbb{E}[||\eta_i^t \sum_{e=1}^{E} \sum_{r=1}^{e} (m_i^t)^r \nabla F_i(w_{i,e}^{\tau_i^t})||^2] + \mathbb{E}[||\eta_i \sum_{e=1}^{E} \nabla F_i(w_{i,e}^{\tau_i^t})||^2]. \tag{69}$$

For the 11th term, based on [35], we have:

$$\mathbb{E}[-2\langle \eta_i^t \sum_{e=1}^{E} \sum_{r=1}^{e} (m_i^t)^r \nabla F_i(w_{i,e-r}^t), \eta_g \sum_{e=1}^{E} \nabla F_g(\widetilde{w}_{g,e}^{\tau_i^t}) \rangle] = 0. \tag{70}$$

To sum up, we have:

$$
\begin{aligned}
&\mathbb{E}[||\widetilde{w}_g^t - w_g^t||^2] \\
&\leq p^2 K \mathbb{E}[(||\widetilde{w}_g^t - w_i^{\tau_i^t})||^2] \\
&\leq p^2 K (\mathbb{E}[||\widetilde{w}_g^t - w^*||^2] + \mathbb{E}[||\widetilde{w}_g^{\tau_i^t} - w^*||^2] + \beta^2 E^2 \delta^2 \\
&\quad + \mathbb{E}[||\eta_i^t \sum_{e=1}^{E} \sum_{r=1}^{e} (m_i^t)^r \nabla F_i(w_{i,e-r}^t)||^2] + \mathbb{E}[||\eta_i^t \sum_{e=1}^{E} \nabla F_i(w_{i,e}^t)||^2] \\
&\quad + \mathbb{E}[||\widetilde{w}_g^t - w^*||^2] + \mathbb{E}[||\eta_i^t \sum_{e=1}^{E} \sum_{r=1}^{e} (m_i^t)^r \nabla F_i(w_{i,e-r}^t)||^2] + \mathbb{E}[||\widetilde{w}_g^t - w^*||^2] + \beta^2 E^2 \delta^2 \\
&\quad - \frac{2\beta^2 R}{\beta^2 + 1} \mathbb{E}[||\widetilde{w}_g^{\tau_i^t} - w^*||^2] - (\frac{2\beta^2 E^2}{\beta^2 + 1} - 1)\mathbb{E}[||\widetilde{w}_g^{\tau_i^t} - w^*||^2] + \beta^2 E^2 \delta^2 \\
&\quad + \mathbb{E}[||\eta_i^t \sum_{e=1}^{E} \sum_{r=1}^{e} (m_i^t)^r \nabla F_i(w_{i,e-r}^t)||^2] + \mathbb{E}[||\eta_i \sum_{e=1}^{E} \nabla F_i(w_{i,e}^{\tau_i^t}||^2]).
\end{aligned}
\tag{71}
$$

Combining like terms, we have:

$$
\begin{aligned}
&\mathbb{E}[||\widetilde{w}_g^t - w_g^t||^2] \\
&\leq p^2 K (3\mathbb{E}[||\widetilde{w}_g^t - w^*||^2] - (\frac{2\beta^2(R + E^2)}{\beta^2 + 1} - 2)\mathbb{E}[||\widetilde{w}_g^{\tau_i^t} - w^*||^2] + \mathbb{E}[||\eta_i^t \sum_{e=1}^{E} \nabla F_i(w_{i,e}^t)||^2] \\
&\quad + 3\beta^2 E^2 \delta^2 + 3\mathbb{E}[||\eta_i^t \sum_{e=1}^{E} \sum_{r=1}^{e} (m_i^t)^r \nabla F_i(w_{i,e}^t)||^2] + \mathbb{E}[||\eta_i \sum_{e=1}^{E} \nabla F_i(w_{i,e}^{\tau_i^t})||^2]).
\end{aligned}
\tag{72}
$$

Since when $\sqrt{\frac{1}{KR+E^2-1}} < \beta < \sqrt{\frac{3}{2RK+2E^2-3}}$, we have $2 < \frac{2\beta^2 R}{\beta^2+1} < 3$, therefore $0 < \frac{2\beta^2(R+E^2)}{\beta^2+1} - 2 < 1$. Since $||\widetilde{w}_g^{\tau_i^t} - w^*||^2 \geq 0$, we know $(\frac{2\beta^2(R+E^2)}{\beta^2+1} - 2)\mathbb{E}[||\widetilde{w}_g^{\tau_i^t} - w^*||^2] \geq 0$. Then, we have:

$$
\begin{aligned}
\mathbb{E}[||\widetilde{w}_g^t - w_g^t||^2] &\leq p^2 K (3\mathbb{E}[||\widetilde{w}_g^t - w^*||^2] + 3\beta^2 E^2 \delta^2 + \mathbb{E}[||\eta_i^t \sum_{e=1}^{E} \nabla F_i(w_{i,e}^t)||^2] \\
&\quad + \mathbb{E}[||\eta_i \sum_{e=1}^{E} \nabla F_i(w_{i,e}^{\tau_i^t})||^2]) + 3\mathbb{E}[||\eta_i^t \sum_{e=1}^{E} \sum_{r=1}^{e} (m_i^t)^r \nabla F_i(w_{i,e-r}^t)||^2].
\end{aligned}
\tag{73}
$$

$\square$

Then, based on Lemma E.5 and E.6, we can easily proof the Theorem A.5 as following:

**Proof.** Based on Assumption 1, we have:

$$F(w_g^t) - F^* \leq \langle \nabla F^*, F(w_g^t) \rangle + \frac{L}{2}||w_g^t - w^*||^2. \tag{74}$$

Since $F^*$ is the global optima, we have $\nabla F^* = 0$, therefore:

$$\begin{aligned}
&F(w_g^t) - F^* \\
&\leq \frac{L}{2}||w_g^t - w^*||^2 \leq \frac{L}{2}||w_g^t - \widetilde{w}_g^t - (\widetilde{w}_g^t - w^*)||^2 \\
&\leq (3Lp^2K + L)\mathbb{E}[||\widetilde{w}_g^t - w^*||^2] + p^2KL(3\beta^2E^2\delta^2 + \mathbb{E}[||\eta_i^t \sum_{e=1}^{E} \nabla F_i(w_{i,e}^t)||^2] \\
&\quad + 3\mathbb{E}[||\eta_i^t \sum_{e=1}^{E}\sum_{r=1}^{e}(m_i^t)^r \nabla F_i(w_{i,e-r}^t)||^2] + \mathbb{E}[||\eta_i \sum_{e=1}^{E}\nabla F_i(w_{i,e}^{\tau_i^t})||^2])).
\end{aligned} \tag{75}$$

Then, based on Lemma E.1, E.5 and E.6, we have:

$$\begin{aligned}
&\mathbb{E}[F(w_g^t)] - F^* \\
&\leq (3Lp^2K+L)(\mathcal{V}^t\mathbb{E}[||w_g^0 - w^*||^2] + \frac{1-\mathcal{V}^t}{1-\mathcal{V}}8\beta^2E^2\delta^2 + \sum_{j=1}^{t}\mathcal{V}^j\mathbb{E}[||\eta_i^{t-1}\sum_{e=1}^{E}[\nabla F_i(w_{i,e}^{t-1})]||^2] \\
&\quad + \sum_{j=1}^{t}\mathcal{V}^j\mathbb{E}[||\eta_i^{t-1}\sum_{e=1}^{E}\sum_{r=1}^{e}(m_i^{t-1})^r\nabla F_i(w_{i,e-r}^{t-1})||^2]) + p^2KL(3\beta^2E^2\delta^2 \\
&\quad + \mathbb{E}[||\eta_i^t\sum_{e=1}^{E}\nabla F_i(w_{i,e}^t)||^2] + 3\mathbb{E}[||\eta_i^t\sum_{e=1}^{E}\sum_{r=1}^{e}(m_i^t)^r\nabla F_i(w_{i,e-r}^t)||^2] + \mathbb{E}[||\eta_i\sum_{e=1}^{E}\nabla F_i(w_{i,e}^{\tau_i^t})||^2]) \\
&\leq (3LpK^2+L)\mathcal{V}^t\mathbb{E}[||w_g^0 - w^*||^2] + \mathcal{U} + \mathcal{W},
\end{aligned} \tag{76}$$

where $\mathcal{U} = [3p^2KL + \frac{8(3pK^2+1)(\beta^2L+L)}{2\beta^2(R+E^2)-2\beta^2-2}]\beta^2E^2\delta^2$, $\mathcal{V} = (3 - \frac{2\beta^2(R+E^2)}{\beta^2+1})$, and

$$\begin{aligned}
\mathcal{W} &= (3Lp^2K+L)\sum_{j=1}^{t}\mathcal{V}^j\mathbb{E}[||\eta_i^{t-1}\sum_{e=1}^{E}[\nabla F_i(w_{i,e}^{t-1})]||^2] + p^2KL(\mathbb{E}[||\eta_i\sum_{e=1}^{E}\nabla F_i(w_{i,e}^{\tau_i^t})||^2] \\
&\quad + \mathbb{E}[||\eta_i^t\sum_{e=1}^{E}\nabla F_i(w_{i,e}^t)||^2]) + (3Lp^2K+L)\sum_{j=1}^{t}\mathcal{V}^j\mathbb{E}[||\eta_i^{t-1}\sum_{e=1}^{E}\sum_{r=1}^{e}(m_i^{t-1})^r\nabla F_i(w_{i,e-r}^{t-1})||^2]) \\
&\quad + 3p^2KL\mathbb{E}[||\eta_i^t\sum_{e=1}^{E}\sum_{r=1}^{e}(m_i^t)^r\nabla F_i(w_{i,e-r}^t)||^2].
\end{aligned} \tag{77}$$

Based on Assumption A.2, we can further bound the gradient expectation term $\mathcal{W}$ by:

$$\begin{aligned}
\mathcal{W} &\leq \frac{(3p^2K+1)(\beta^2L+L)}{2\beta^2(R+E^2)-2\beta^2-2}\beta^2E^2G_c^2 + 2p^2KL\beta^2E^2G_c^2 \\
&\quad + \frac{(3p^2K+1)(\beta^2L+L)}{2\beta^2(R+E^2)-2\beta^2-2}\beta^2RQ(t)G_c^2 + 3p^2KL\beta^2RQ(t)G_c^2 \\
&\leq [p^2KL(2E^2+3RQ(t)) + \frac{(3p^2K+1)(\beta^2L+L)(E^2+RQ(t))}{2\beta^2(R+E^2)-2\beta^2-2}]\beta^2G_c^2.
\end{aligned} \tag{78}$$

$\square$

