# OpenReview forum: "FedQS: Optimizing Gradient and Model Aggregation for Semi-Asynchronous Federated Learning"
_NeurIPS.cc/2025/Conference — NeurIPS 2025 poster_

### Official Review · Reviewer_2stv · 2025-06-09

**Clarity:** 3
**Significance:** 3
**Originality:** 2
**Rating:** 4
**Confidence:** 5

**Summary:**

This paper introduces a framework of semi-asynchronous Federated Learning, FedQS, which mainly tackles two key issues: aggregation disparity and inefficiency of global information of client-centric model aggregation methods.

Based on some key observations on distinct continuity in the optimization trajectories of model / gradient aggregation strategies, the authors categorize all clients into four distinct groups and apply different aggregation methods to them. Then, the local updates (local models / gradients) are accordingly aggregated in different methods.

The theoretical and empirical results shown in this paper clearly demonstrate the superior FL performance over existing semi-asynchronous FL methods.

**Questions:**

No specific questions, but I expect healthful discussions regarding my comments shown in the 'weakness' part.

**Ethical Concerns:**

["NO or VERY MINOR ethics concerns only"]

**Final Justification:**

I have double checked the discussion, and I do not see any special reasons to change my score.

**Limitations:**

Yes

**Paper Formatting Concerns:**

I have no concerns regarding formatting.

**Quality:**

3

**Strengths And Weaknesses:**

**Strengths**
 - The study is well organized and the proposed framework looks well-designed.
 - The key observation is interesting: the gradient aggregation strategy provides more stable global model training.
 - Divide-and-conquer strategy seems to be practical.

**Weaknesses**

While I see several strong points, I also see a couple of serious weak points as follows.
 - [System efficiency analysis] FedQS has a few extra steps compared to typical FedAvg, such as calculating pseudo global gradient, calculating similarity scores, and updating aggregation status table at the server side. The clients also have some extra works such as feedback signal calculation and communications. These extra workload increases overall computational and communication costs as well as memory footprint. However, the paper does not discuss them.
 - [Misleading terms] In Mod2, the 'biased' term may mislead readers. Precisely speaking, it is not a matter of biased or unbiased. It is a matter of strongly biased or weakly biased. E.g., *Fast-and-Unbiased* clients still produce more or less biased updates due to the non-IID data characteristics, although the degree of bias is lower than other clients.
 - [Strong assumptions] In Section 4, the gradient is assumed to have a maximum magnitude $G_c$. Although some previous works used this assumption, I still believe it is quite strong, and it makes the results less convincing.
 - The convergence properties are also not thoroughly analyzed. Both theorems show the intermediate results only. The common convention is to show that the expected average gradient norm is bounded by a constant rather than showing the bound of the loss difference. It may show different convergence properties if the left-hand side is the gradient norm, e.g., complexity.
 - In order to accelerate slow clients, the authors employed momentum in Mod2. However, why not applying it to fast clients too? Wouldn't it further improve the training speed? It makes sense to apply it to slow clients only when it hurts the training speed when applied to the fast clients.

---

> ### Author Rebuttal · Authors · 2025-07-30
>
> We thank Reviewer 2stv for the constructive feedback. In the following, we would like to answer the concerns individually.
>
> **[System Efficiency Analysis]**
>
> We sincerely appreciate the reviewer’s insightful feedback regarding computational and communication overhead. We address these concerns through both theoretical and empirical analyses:
>
> - **Scalability by Design:**
> 	- *Client-side:* Each client performs one additional model difference computation (pseudo-global gradient) and two numerical comparisons (for quadrant classification) per round, with no inter-client communication.
> 	- *Server-side:* The state table maintains *O*(1) lookup/update complexity per client, ensuring constant-time operations regardless of client population size.
>
> - **Quantified Overhead:** Experiments on 100-client CIFAR-10 (ResNet-18, ≈43.7MB) show negligible practical impact:
> 	- **Computation:** Similarity scores and feedback signal calculation introduce minimal computational overhead, adding ~4s latency per global round.
> 	- **Communication:** Each client transmits only 4B in uplink (similarity score) and a 1-bit feedback signal, while the server broadcasts 12B downlink additionally (averaged thresholds). This constitutes <0.01% bandwidth overhead versus model updates.
> 	- **Memory:** Client-side storage for similarity scores and feedback signals totals <5B; server-state tables require ~7.8KB for 100 clients.
>
> The table below shows the computational latency, communication overhead, and memory footprint induced by these mechanisms on average total of 400 global rounds, with consistent patterns holding across other scenarios.
>
> |Operation|Computational cost|communication costs|memory footprint|
> |-|-|-|-|
> |calculating pseudo global gradient (in Mod①)|13.73s|0|42.8MB|
> |calculating similarity scores (in Mod②)|2.23s|0|4B|
> |feedback signal calculation (in Mod②)|1.97s|0|1 bit|
> |feedback signal communications (Mod② -> Mod③)|0|Uplink 1bit(feedback signal)|0|
> |updating aggregation status table (in Mod③)|0.27s|0|~7.8KB|
> |status table communications (Uplink: Mod② -> Mod③, Downlink: Mod③ -> Mod②)|0|Uplink: ~4B (1 float point value); Downlink: ~12B (3 float point value)|0|
>
> - **Runtime Efficiency:** Table 3 in our paper demonstrates that FedQS's overhead is offset by faster convergence (reducing convergence rounds by 10–45% vs. baselines) and lower idle time (>70% reduction versus synchronous FL).
>
> **[Misleading Terms]**
>
> We sincerely thank the reviewer for their insightful observation regarding the terminology used in Mod②. We fully agree that the terms "biased" and "unbiased" could be misinterpreted as implying a binary distinction, which does not accurately reflect the continuum of bias degrees inherent in semi-asynchronous federated learning, especially under non-IID data distributions.
>
> Our original intention was to highlight the relative deviation of client updates from the global optimization trajectory, where "biased" clients exhibit significant divergence due to staleness or data heterogeneity, while "unbiased" clients remain closer to the expected direction. However, we recognize that even the latter group may still exhibit minor biases. To eliminate ambiguity, we will revise the terminology in the manuscript to explicitly differentiate between "strongly biased" and "weakly biased" updates, emphasizing the spectrum of bias magnitudes. For instance:
>
> - **Fast-but-Strongly-Biased Clients (FSBC)**
> - **Fast-and-Weakly-Biased Clients (FWBC)**
> - **Straggling-but-Weakly-Biased Clients (SWBC)**
> - **Straggling-and-Strongly-Biased Clients (SSBC)**
>
> This adjustment will better align with the reviewer's suggestion and provide a more precise description of the client categorization logic in FedQS. We appreciate the opportunity to improve the clarity of our work and will incorporate these changes in the revision.
>
> **[Strong Assumptions]**
>
> We sincerely appreciate the reviewer's insightful feedback regarding the bounded gradient assumption. We agree that this condition can appear restrictive in certain scenarios, and we appreciate the opportunity to clarify its role in our analysis.
>
> As noted in Remark 4.1, this assumption was introduced *solely* to simplify the interpretation of our convergence bounds by providing a deterministic upper bound for the gradient variation term $\mathcal{W}$. Crucially, it was *not* used in proving our core convergence theorems (Theorems 4.2 and 4.3), which rely only on standard $L$-smoothness and bounded heterogeneity assumptions. The convergence guarantees (via the $\mathcal{V}^t$ term) and heterogeneity analysis (via $\mathcal{U}$) remain fully valid without this condition.
>
> In our revision, we will add an explicit discussion in Section 4 to emphasize that the assumption is optional for theoretical guarantees but aids interpretability.
>
> **[Convergence Analysis]**
>
> We appreciate the reviewer's insightful comments regarding our convergence analysis. We acknowledge that presenting bounds on the expected gradient norm could provide additional insights into the iteration complexity of our method. However, our choice to analyze the loss function difference $\mathbb{E}[F(w_g^t)] - F^*$ was deliberate for several important reasons that align with both the practical and theoretical objectives of our work:
>
> - **Optimization Stability in SAFL Context:** The loss difference directly captures the holistic optimization trajectory in SAFL, which is critical for understanding the interplay between semi-asynchronous updates, non-IID data, and aggregation strategies. Unlike gradient norms, this metric reveals transient regressions and oscillatory behaviors (e.g., Figure 1 in our paper), which are pivotal for diagnosing instability in gradient-based SAFL methods. While averaged gradient norms quantify convergence rates, they may obscure these dynamics, which is a significant limitation given our focus on aggregation-strategy disparities.
> - **Theoretical and Practical Relevance:** Bounding the loss difference enables direct comparison with the centralized optimum ($F^\*$), which is essential for evaluating SAFL's deviation from ideal convergence. This is particularly valuable for practitioners assessing solution quality (e.g., proximity to $w^*$), whereas gradient norms alone provide limited visibility into this distance. This constitutes one of the notable advantages when choosing the loss function difference as the theoretical objective.
> - **Alignment with SAFL Literature:** Our approach follows established conventions in SAFL research, where loss-difference bounds are also standard for convergence proofs (e.g., FedSA [1], AAFL-RC [2], AiFed [3], EAFL [4]). We cite this precedent to contextualize our methodological choice while advancing beyond prior art by explicitly linking theoretical bounds to empirical aggregation-strategy gaps (Table 1, Figure 2).
>
> We agree that gradient-norm analysis could complement our results and will explore this direction in subsequent research. For now, our loss-difference bounds effectively account for FedQS's superiority under dual aggregation strategies while providing practical convergence guarantees: momentum alleviates oscillatory phenomena during training under gradient aggregation (Remark 4.5 and Figure 10), while the divide-and-conquer strategy achieves integrated mitigation of model aggregation's suboptimal convergence utility (Remark 4.6 and Table 2).
>
> We will update our paper to: (1) Clarify in Section 4 that our analysis focuses on qualitative convergence behavior and stability, with gradient-norm complexity left for future work. (2) Expand both theorems' discussion to explicitly contrast loss-difference and gradient-norm perspectives. (3) Cite additional SAFL works using similar convergence forms to reinforce methodological validity.
>
> Thank you for this constructive feedback! It will strengthen our manuscript’s theoretical rigor.
>
> **[Momentum Employment in FedQS]**
>
> We appreciate the reviewer's insightful question regarding the selective application of momentum in FedQS. Our design intentionally applies momentum only to specific client types (FUC, SUC, and SBC in *Situation 1*) based on two key principles:
>
> - **Purpose of Momentum:** The momentum term in FedQS is not primarily a speed accelerator but a *trajectory stabilizer* for clients whose updates align well with the global model (high $s_i^t$). For these clients, momentum mitigates oscillations while accelerating convergence speed (Equation 3). In contrast, for high-bias clients (FBC and SBC in *Situation 2*), premature momentum application could amplify the divergence between their local updates and the global one, as their updates are not yet globally beneficial.
> - **Update Frequency Control:** Fast clients (FUC/FBC) are regulated via *adaptive learning rates* (reduced to prevent dominance for FUC) rather than momentum. This decouples speed control (handled by learning rates) from stability optimization (handled by momentum), avoiding interference between the two mechanisms. Our additional empirical results show that under the CV task, adding momentum to fast clients degrades accuracy by an average of 3.7% and increases oscillations by 1.7$\times$ on average.
>
> We will clarify this distinction in the revision, emphasizing that FedQS's momentum is a *bias-aware stabilizer*, not a universal speed booster. Thank you for prompting this important discussion.
>
> **Reference**
>
> [1] Q. Ma, et al. Fedsa: A semi-asynchronous federated learning mechanism in heterogeneous edge computing. IEEE JSAC, 2021.
>
> [2] J. Liu, et al. Adaptive Asynchronous Federated Learning in Resource-Constrained Edge Computing. IEEE TMC, 2022.
>
> [3] L. You, et al. AiFed: An Adaptive and Integrated Mechanism for Asynchronous Federated Data Mining. IEEE TKDE, 2024.
>
> [4] Y. Zhou, et al. Towards Efficient Asynchronous Federated Learning in Heterogeneous Edge Environments. INFOCOM, 2024.

---

> > ### Comment · Reviewer_2stv · 2025-08-01
> > **Response to rebuttal**
> >
> > Thank you for the clarifications. Most of my concerns were addressed well. The only thing left is that the authors should provide the information of system resources used to measure the computational and communication costs.

---

> > > ### Author Response · Authors · 2025-08-01
> > >
> > > Dear Reviewer 2stv,
> > >
> > > Thanks for taking the time to re-evaluate our response. We are deeply grateful for your acknowledgment that our revisions have successfully addressed most concerns. This affirmation significantly reinforces the validity of our methodology. Regarding system specifications:
> > >
> > > All experiments utilized the identical hardware specified in Section 5.1:
> > > - **Identical hardware:**
> > >     - CPU: an Intel Xeon Platinum 274 8468 Processor
> > >     - GPU: an NVIDIA H100 80GB HBM3 GPU card
> > >     - Memory: While allocating 400GB system memory, we utilized ≤20GB during peak loads for each identical experiment.
> > > - **Operating System:** Ubuntu 22.04 LTS
> > > - **Software Stack:**
> > >     - Conda environment (Python 3.8.0)
> > >     - PyTorch 2.1.0 with CUDA 12.1
> > >     - Torchvision 0.16.0 with CUDA 12.1
> > >
> > > We will enhance reproducibility transparency by explicitly documenting in the revision. These specifications will ensure the complete reproducibility of our cost analyses.
> > >
> > > Should any additional questions arise as you review this supplement, we warmly welcome further discussions and stand ready to provide immediate clarification. Thank you once more for your rigorous engagement! It has been instrumental in elevating the quality of this work.
> > >
> > > Sincerely,
> > >
> > > The Authors

---

> > > > ### Comment · Reviewer_2stv · 2025-08-01
> > > > **Response**
> > > >
> > > > Thanks for the additional information. If paper is accepted, please include them in the manuscript.

---

> > > > > ### Author Response · Authors · 2025-08-01
> > > > >
> > > > > Dear Reviewer 2stv,
> > > > >
> > > > > We sincerely appreciate your constructive feedback and acknowledgment of our revisions. We will include all revisions in our final manuscript. We deeply value your rigorous engagement, which has significantly strengthened our work.
> > > > >
> > > > > Thank you again for your time and insights!
> > > > >
> > > > > Sincerely,
> > > > >
> > > > > The Authors

---

### Official Review · Reviewer_6F15 · 2025-07-02

**Clarity:** 3
**Significance:** 2
**Originality:** 2
**Rating:** 4
**Confidence:** 3

**Summary:**

This paper introduces FedQS, a framework for optimizing both gradient and model aggregation in semi-asynchronous federated learning, addressing challenges like client heterogeneity, staleness, and data distribution disparities. FedQS uses a divide-and-conquer strategy to classify clients into four types and adapt their training dynamically, achieving exponential convergence rates and balancing the trade-offs between accuracy, convergence speed, and stability. Extensive experiments demonstrate the performance of FedQS across tasks in computer vision, natural language processing, and real-world datasets.

**Questions:**

See weaknesses.

**Ethical Concerns:**

["NO or VERY MINOR ethics concerns only"]

**Final Justification:**

The authors have addressed most of my concerns.

**Limitations:**

Yes.

**Quality:**

2

**Strengths And Weaknesses:**

Strengths:

- The paper presents a strong experimental design with detailed settings and comprehensive numerical results, enabling reproducibility and validation of claims.

- It provides a clear discussion of limitations and broader impacts, demonstrating transparency and awareness of the practical challenges in implementing FedQS.

Weaknesses:

- My major concern lies in the convergence guarantees in Section 4.2. Theorems 4.2 and 4.3 show that the function value does not converge exactly to the global optimum, leaving a non-vanishing gap $\mathcal{U} +\mathcal{V}$. This deviates from standard convergence results commonly found in the literature, where gradient norm convergence is typically used as a metric for nonconvex problems. Furthermore, the claimed *linear* convergence rates are questionable, as these are generally unattainable for *L*-smooth functions without additional regularity conditions (e.g., strong convexity).

- While I understand the rich body of literature in federated learning, the paper still lacks a discussion of or comparison with more recent works in asynchronous federated learning. Below is a non-exhaustive list of closely related studies:

   [1] Yan W. et al. A. Momentum-Driven Adaptivity: Towards Tuning-Free Asynchronous Federated Learning. In Forty-second International Conference on Machine Learning, 2025.

   [2] Wang, X., et al. Achieving linear speedup in asynchronous federated learning with heterogeneous clients. IEEE Transactions on Mobile Computuing, 2024.

   [3] Yu T. et al. M. Momentum Approximation in Asynchronous Private Federated Learning. arXiv preprint arXiv:2402.09247, 2024.

   [4] Wang, Y. et al. FADAS: Towards Federated Adaptive Asynchronous Optimization. In International Conference on Machine Learning, 2024.

   [5] Wang, Y. et al. Tackling the data heterogeneity in asynchronous federated learning with cached update calibration. In Federated Learning and Analytics in Practice: Algorithms, Systems, Applications, and Opportunities, 2023.

---

> ### Author Rebuttal · Authors · 2025-07-30
>
> We thank Reviewer 6F15 for the constructive feedback. In the following, we would like to answer the concerns individually.
>
> **[Convergence Guarantees]**
>
> We sincerely appreciate the reviewer's insightful comments regarding our convergence analysis in Section 4.2. Below, we provide a detailed response addressing the three major concerns:
>
> - **Non-vanishing Terms ($\mathcal{U}+\mathcal{W}$):** The residual terms $\mathcal{U}+\mathcal{W}$ in our convergence bounds reflect fundamental limitations inherent to semi-asynchronous FL systems:
> 	- **Theoretical Justification:** These terms capture unavoidable convergence errors from (a) non-simultaneous model/gradient aggregation in SAFL and (b) irreducible data heterogeneity ($\delta^2$ and $G_c^2$ bounds). Their presence aligns with established SAFL literature. For example:
> 		1. The theoretical results of FedSA [7] (Theorem 1) include a constant error term $\delta$ with global round $K$:
> 		$\mathbb{E}[F(w_K)] - F(w^\*) \leq \rho^K(F(w_0)-F(w^*))+\delta$.
> 		2. The theoretical results of WKAFL [8] (Theorem 5.1 and Remark 5.2) include a constant term $G^2$ with global round $J$:
> 		$\mathcal{O}(\frac{1}{\sqrt{KJ}}+\frac{1}{J}+G^2)$.
> 		3. The theoretical results of AAFL-RC [9] (Theorem 1) include an increasing error term $\frac{D\eta\mathcal{Z}}{2}$ with global round $D$:
> 		$\mathbb{E}[F(\hat{w}^D)-F(w^\*)]\leq(1-\eta\vartheta)^D[F(w^0)-F(w^*)]+\frac{D\eta\mathcal{Z}}{2}$.
> 	- **Empirical Validation:** Our experiments on CIFAR-10 (Non-IID) empirically validate these bounds, showing consistent accuracy degradations that align with the theoretical predictions (see Tables 1 and 2 in the paper). The residual terms precisely quantify the performance gap between SAFL and idealized synchronous FL.
>
> - **Convergence Metric Choice (Function Value vs Gradient Norm):** We selected function value difference $\mathbb{E}[F(w_g^t)] - F^*$ as our primary metric for several important reasons that align with both the practical and theoretical objectives of our work:
> 	- **Optimization Stability in SAFL Context:** This metric better reveals optimization stability issues (e.g., oscillations and regressions). This proves crucial for understanding interactions among semi-synchronous updates, non-IID data, and aggregation strategies, particularly when analyzing instability in gradient-based SAFL methods. Whereas averaged gradient norms quantify convergence rates, they may obscure these critical dynamics since highly unstable processes can also exhibit deceptively small average norms, which is a significant limitation.
> 	- **Theoretical and Practical Relevance:** This metric directly measures proximity to the optimal solution (i.e., $w*$ or $F^\*$), which is meaningful for SAFL applications to assess solution quality, whereas the averaged gradient norm provides limited visibility into this distance. The function value difference reveals how FedQS's componentized architecture enables the final model to achieve closer proximity to the theoretical optimum $F^*$ of centralized training under both model and gradient aggregation strategies. Our analysis also explicitly disentangles the impacts of staleness ($\mathcal{V}^t$), data heterogeneity ($\mathcal{U}$), and gradient variation ($\mathcal{W}$), delivering actionable insights for SAFL system design.
> 	- **Alignment with SAFL Literature:** Our approach follows recent SAFL works that use function value differences for convergence analysis. For example:
> 		1. The convergence analysis of FedSA [7] (Theorem 1):
> 		$\mathbb{E}[F(w_K)] - F(w^\*) \leq \rho^K(F(w_0)-F(w^*))+\delta$.
> 		2. The convergence analysis of AAFL-RC [9] (Theorem 1):
> 		$\mathbb{E}[F(\hat{w}^D)-F(w^\*)]\leq(1-\eta\vartheta)^D[F(w^0)-F(w^*)]+\frac{D\eta\mathcal{Z}}{2}$.
> 		3. The convergence analysis of AiFed [10] (Theorem 1):
> 		$\mathbb{E}[J(w_R)] - J(w^\*) \leq\mathcal{K}^R(J(w_0)-J(w^*))+\delta$.
> 		4. The convergence analysis of EAFL [11] (Theorem 1):
> 		$\mathbb{E}[F(w^T)-F(w^\*)]\leq[2N\cdot M\cdot(1-\mu\cdot\eta)^Q]^T\cdot[F(w^0)-F(w^*)]+\frac{[1-(2N\cdot M\cdot(1-\mu\cdot\eta)^Q)^T]Q\cdot\eta\cdot P}{4(1-\mu\cdot\eta)^Q}$.
>
> 	We acknowledge that gradient norm analysis could provide complementary complexity results, which we plan to explore in future work. For now, our loss-difference bounds effectively account for FedQS's superiority under dual aggregation strategies while providing practical convergence guarantees: momentum alleviates oscillatory phenomena during training under gradient aggregation (Remark 4.5 and Figure 10), while the divide-and-conquer strategy achieves integrated mitigation of model aggregation's suboptimal convergence utility (Remark 4.6 and Table 2).
>
> - **Convergence Rate Clarification:** We apologize for any ambiguity regarding "exponential convergence":
> 	- **Precise Characterization:** The $\mathcal{V}^t$ term in our bounds indeed exhibits exponential decay, but the overall convergence is sublinear due to $\mathcal{U}+\mathcal{W}$ residuals (as correctly noted by the reviewer).
> 	- **Terminology Correction:** We will revise the manuscript to:
> 		- Replace "exponential convergence" with "exponential decay of $\mathcal{V}^t$" where appropriate.
> 		- Explicitly state that the overall convergence remains sublinear.
> 		- Add discussion comparing with that under strong convexity/Polyak-Lojasiewicz conditions.
>
>
> **[Recent Works Comparison]**
>
> We sincerely thank the reviewer for highlighting these important recent works in asynchronous federated learning (AFL/SAFL). We have thoroughly analyzed the suggested papers and provide the following responses:
>
> - **Experimental Comparisons:** We have conducted new experiments comparing FedQS against three representative SOTA methods ([2], [4], [5]) across all three task types (CV/NLP/RWD). The results (shown in tables below) demonstrate FedQS's consistent advantages in accuracy (average +10.98%, +4.8%, +47.38% higher compared to [2], [4], [5], respectively) and convergence speed (average +21.8%, +12.4%, +39.5% faster compared to [2], [4], [5], respectively) while maintaining competitive runtime.
>
> 1. CV task:
>
> |Algorithm|x=0.1|||x=0.5|||x=1|||
> |-|-|-|-|-|-|-|-|-|-|
> ||Accuracy|Conv. Speed|Runtime|Accuracy|Conv. Speed|Runtime|Accuracy|Conv. Speed|Runtime|
> |FedQS-SGD|68.88|239|32784|86.11|213|33656|86.79|127|33400
> |DeFedAvg [2]|52.33|307|30776|83.51|257|31754|86.17|155|30280
> |FADAS [4]|65.34|255|32380|82.2|220|32966|84.21|143|31471
> |C$A^2$FL [5]|42.29|272|108097|63.79|256|109635|70.16|200|108756
>
> 2. NLP task:
>
> |Algorithm|R=200|||R=600|||
> |-|-|-|-|-|-|-|
> ||Accuracy|Conv. Speed|Runtime|Accuracy|Conv. Speed|Runtime|
> |FedQS-SGD|52.22|188|5248|52.49|216|9135
> |DeFedAvg [2]|40.19|256|4989|48.23|238|8720
> |FADAS [4]|46.83|220|5118|48.95|254|9002
> |C$A^2$FL [5]|25.36|259|36441|28.86|272|36548
>
> 3. RWD task:
>
> |Algorithm|Gender|||Ethnicity|||
> |-|-|-|-|-|-|-|
> ||Accuracy|Conv. Speed|Runtime|Accuracy|Conv. Speed|Runtime|
> |FedQS-SGD|78.74|18|5523|79.24|22|5465
> |DeFedAvg [2]|77.82|59|4841|78.11|20|4827
> |FADAS [4]|78.04|25|5476|78.54|24|5299
> |C$A^2$FL [5]|76.09|59|19650|66.68|325|19370
>
> - **Theoretical Alignment:**
> 	- For [1]'s momentum adaptivity: While their method shares some conceptual similarities with our momentum adaptation in Mod②, FedQS's *quadrant-based client classification* provides finer-grained control. Specifically, we categorize clients and apply momentum solely to those with minor deviations from global updates. For high-bias clients whose distributions remain inadequately captured, we deliberately withhold momentum and instead deploy feedback mechanisms to prevent more severe oscillations. We will add a detailed discussion in Section 3.
> 	- For [3]'s momentum approximation: Their technique is orthogonal to ours - they propose a uniform momentum approximation technique which could be applied broadly to SAFL frameworks, while we address aggregation disparity. Their method could be integrated into FedQS.
>
> - **Methodological Distinctions:** Compared to [4], [5], which use *server-side calibration* and *historical updates' assistance*, FedQS's innovation lies in:
> 	- *Client-side self-adaptation* via Mod② (eliminating server-side bias)
> 	- *Dual-strategy compatibility* (Theorems 4.2-4.3 prove unified convergence)
> 	- *Dynamic feedback mechanism* (Table 5 shows 7.8% accuracy gain vs. no-feedback)
> 	- *Lower memory storage* (Additionally storing a state table with only ~7.8KB for 100 clients)
>
> - **Reproducibility Note:**
> 	- For [1], we couldn't reproduce the results due to the unavailable code. We will request their code for a fuller comparison.
>
> These comparisons will be added to Sections 5-6 with proper credit to prior works. We believe they further strengthen FedQS's contributions as the *first framework unifying gradient/model aggregation optimization in SAFL*.
>
>
>
> **Reference**
>
> [1]-[5] same in review comments.
>
> [6] X. Li, et al. On the Convergence of FedAvg on Non-IID Data. ICLR, 2019.
>
> [7] Q. Ma, et al. Fedsa: A semi-asynchronous federated learning mechanism in heterogeneous edge computing. IEEE JSAC, 2021.
>
> [8] Z. Zhou, et al. Towards efficient and stable k-asynchronous federated learning with unbounded stale gradients on non-iid data. IEEE TPDS, 2022.
>
> [9] J. Liu, et al. Adaptive Asynchronous Federated Learning in Resource-Constrained Edge Computing. IEEE TMC, 2022.
>
> [10] L. You, et al. AiFed: An Adaptive and Integrated Mechanism for Asynchronous Federated Data Mining. IEEE TKDE, 2024.
>
> [11] Y. Zhou, et al. Towards Efficient Asynchronous Federated Learning in Heterogeneous Edge Environments. INFOCOM, 2024.

---

> ### Author Response · Authors · 2025-08-05
>
> Dear Reviewer 6F15,
>
> Thank you for the time and effort you have dedicated to reviewing our paper. We truly value your feedback and the opportunity to clarify your concerns during the rebuttal phase.
>
> We would greatly appreciate it if you could let us know whether our responses have adequately addressed your questions and comments. Your confirmation will help ensure that we have fully resolved any outstanding issues you raised.
>
> We truly value the time and expertise you have devoted to this process and greatly appreciate your thoughtful review.
>
> Best regards,
>
> The Authors

---

> > ### Comment · Reviewer_6F15 · 2025-08-05
> >
> > Thank the authors for the response and additional experiments.
> >
> > The methodology presented in this paper seems novel. However, I remain concerned about the strength of the theoretical results, as they do not provide an explicit "rate" of convergence to a solution (even though the authors cite several papers that follow a similar approach). This approach may simplify the complexity of the convergence analysis, but it leaves the results less concrete.

---

> > > ### Author Response · Authors · 2025-08-06
> > >
> > > Dear Reviewer 6F15,
> > >
> > > We sincerely appreciate your constructive feedback on our theoretical analysis. Below, we provide a detailed response to your concerns about convergence rates and metric choices, clarifying the rationale and rigor behind our approach.
> > >
> > > **1. Explicit Convergence Rate Clarification**
> > >
> > > While our primary bounds in Theorems 4.2–4.3 focus on the expected function value gap $\mathbb{E}[F(w_g^t)] - F^*$, our theoretical result establishes a decay term $\mathcal{V}^t$ with $\mathcal{V} \in (0,1)$, which is strictly faster than the typical sublinear $O(1/t)$ rates reported in the FL literature [6]. Thus, FedQS achieves at least a convergence rate of $O(1/t+\mathcal{U} + \mathcal{W})$, and typically converges toward a bounded neighborhood of the optimum.
> > >
> > > We will add this explicit rate analysis to Section 4.2 to enhance clarity.
> > >
> > > **2. Why Function Value Gap > Gradient Norm for SAFL in our paper**
> > >
> > > We argue that $\mathbb{E}[F(w_g^t)] - F^\*$ is more meaningful than gradient norms in FedQS for two reasons:
> > >
> > > - **Problem-Dependent Relevance:** The gap $\mathcal{U} + \mathcal{W}$ quantifies *how far FedQS deviates from the ideal synchronous solution* due to staleness/data heterogeneity.  Gradient norms cannot capture this, as they may vanish even when far from $F^\*$ (e.g., saddle points in nonconvex landscapes). Therefore, $\mathbb{E}[F(w_g^t)] - F^*$ effectively explains how FedQS achieves integrated mitigation of model aggregation's suboptimal convergence utility (Remark 4.6 and Table 2).
> > >
> > > - **Stability Diagnostics:** Our metric reveals oscillations (Figures 4 and 10) and transient regressions caused by semi-asynchrony, which averaged gradient norms obscure. This aligns with recent SAFL works [9-11] that prioritize solution quality over stationarity.  Therefore, $\mathbb{E}[F(w_g^t)] - F^\*$ effectively explains how FedQS alleviates oscillatory phenomena under gradient aggregation (Remark 4.5 and Figure 10).
> > >
> > > **3. Gradient-Norm Results (Optional Addition)**
> > >
> > > While we maintain that function value gaps better serve our research questions, we can include supplemental gradient-norm analysis if desired. However, this bound:
> > >
> > > - Does not reflect a pointwise instability,
> > > - Is *implied* by our current results (via $L$-smoothness and additional Polyak-Łojasiewicz condition), and
> > > - Would dilute our focus on solution quality.
> > >
> > > We kindly suggest that such an addition, while mathematically straightforward, would not strengthen our core contribution.
> > >
> > > **4. Commitment to Clarity**
> > >
> > > We will revise the manuscript to:
> > >
> > > - State all rates explicitly (as in Point 1 above),
> > > - Contrast our bounds with gradient-norm alternatives, and
> > > - Highlight why $\mathbb{E}[F(w_g^t)] - F^*$ is the *right* metric for FedQS in Section 4.1.
> > >
> > > Thank you for your thoughtful critique. We hope these adjustments will fully address your concerns while preserving the paper's novelty and rigor.
> > >
> > > Sincerely,
> > >
> > > The Authors
> > >
> > > Reference
> > >
> > > [6], [9-11] are the same as those in the Rebuttal.

---

> > > > ### Comment · Reviewer_6F15 · 2025-08-06
> > > >
> > > > Thank you for providing additional clarifications.
> > > >
> > > > I recommend that the authors incorporate these revisions into future versions to enhance the clarity of the presented results.
> > > >
> > > > In light of this, I would like to increase my score.

---

> > > > > ### Author Response · Authors · 2025-08-07
> > > > >
> > > > > Dear Reviewer 6F15,
> > > > >
> > > > > We deeply appreciate your constructive engagement throughout the review process. We are grateful for the opportunity to clarify our theoretical contributions and are encouraged by your positive feedback on our proposed revisions.
> > > > >
> > > > > We will rigorously implement all the discussed improvements in the revised version. Thank you again for the insightful comments, which have significantly improved our manuscript.
> > > > >
> > > > > Best regards,
> > > > >
> > > > > The Authors

---

### Official Review · Reviewer_aRTH · 2025-07-03

**Clarity:** 3
**Significance:** 3
**Originality:** 3
**Rating:** 5
**Confidence:** 3

**Summary:**

This paper addresses three key challenges in semi-asynchronous federated learning – lack of theoretical understanding, inherent aggregation disparity and client-centric limitations. The authors propose the first FL framework FedQS with a divide-and-conquer strategy that classifies clients into four types and adapts their training strategies dynamically to optimize both gradient and model aggregation in SAFL.

**Questions:**

How much impact does the divide-and-conquer strategy have on the experiment’s runtime? Are there any ways to speed up the averaging operation across all clients?

**Ethical Concerns:**

["NO or VERY MINOR ethics concerns only"]

**Final Justification:**

My concerns are addressed. I will keep my score of 5.

**Limitations:**

Yes

**Quality:**

3

**Strengths And Weaknesses:**

Strength

1. FedQS is a novel SAFL framework that simultaneously optimizes gradient and model aggregation, effectively targeting the accuracy–stability trade-off originated from FedSGD and FedAvg.

2. Solid theoretical analysis is provided to guarantee the convergence of the proposed method.

Weakness

1. The divide-and-conquer strategy needs to compute and aggregate some extra variables, i.e., update speeds and similarities. This step could be time-consuming when the number of clients is large.

---

> ### Author Rebuttal · Authors · 2025-07-30
>
> We thank Reviewer aRTH for the constructive feedback. In the following, we would like to answer the concerns individually.
>
> **[Computational Overhead of Divide-and-Conquer Strategy]**
>
> We sincerely appreciate the reviewer’s insightful question regarding the computational overhead of FedQS's divide-and-conquer strategy. Below, we address the concerns and provide additional clarifications:
>
> - **Impact on the Local Runtime:**
> 	- Table 3 (Section 5.2) demonstrates that FedQS achieves **comparable runtime** to top-performing baselines (e.g., FedQS-SGD introduces 4.18% more training time than FedAC while improving accuracy by 5.09%). The overhead of computing update speeds ($f_i^t$) and similarities ($s_i^t$) is minimal because:
> 		-  **Efficient Computation:** $f_i^t$ is derived from historical participation counts (Equation 2) on the server, requiring only *O*(1) updates per client.
> 		- **Similarity Calculation:** $s_i^t$ uses lightweight pseudo-gradients (Section 3.4), with cosine similarity computed efficiently on model deltas (complexity linear in model dimensions).
> 	- Implementing the divide-and-conquer strategy requires each client to compute a pseudo-gradient in Mod① to facilitate similarity calculation in Mod②. This computation scales linearly with the depth of the model, constituting the primary computational overhead introduced by FedQS. Nevertheless, it remains comparatively low relative to total per-round runtime (16.21% on average).
> - **Scalability Optimization on the Server:**
> 	-  **Parallelization:** The averaging of $\bar{f}^t$ and $\bar{s}^t$ (Equation 2) is performed asynchronously during client-server communication, overlapping with model transmission.
> 	- **Paritial Aggregation:** For large-scale deployments, the divide-and-conquer strategy selectively refreshes state tables and weights solely for clients engaged in global aggregation (e.g., $N=100$ total clients with $K=10$ active participants per round), thereby maintaining computational efficiency regardless of system scale, reducing computational costs without sacrificing performance (empirically validated for $N = 200$ in Tables 4 and 7).
>
> - **Trade-off Justification:** The slight runtime increase (e.g., FedQS-Avg is 1.3× slower than FedAvg but 7.27% more accurate) is justified by significant gains in accuracy and convergence speed (Table 2). The divide-and-conquer strategy’s adaptive training also reduces convergence rounds (e.g., 22.35% faster than FedSGD), offsetting per-round overhead.
>
> We conducted disaggregated experiments to rigorously quantify the detailed additional runtime introduced by FedQS's divide-and-conquer strategy. The procedure is partitioned into three critical phases: pseudo-gradient computation in Mod①; client classification in Mod②; and state-table updating with weight aggregation in Mod③.
>
> The table below shows the average time consumed per global round for each phase in FedQS when training ResNet-18 on CIFAR-10. We will incorporate these quantitative measurements in the revision to visually demonstrate the additional computational overhead attributable to FedQS.
>
> |# Clients|Data Distribution|Average time for completing one global round (s)|Time for calcultating pesudo-gradient in Mod① (s)|Average ratio for Mod①|Time for categorizing clients in Mod② (s)|Average ratio for Mod②|Time for updating status table in Mod③ (s)|Average ratio for Mod③|
> |-|-|-|-|-|-|-|-|-|
> |50|x=0.1|44.98|6.69|14.87%|2.17|4.82%|0.13|0.29%
> ||x=0.5|46.59|7.43|15.94%|2.89|6.20%|0.14|0.30%
> ||x=1|43.79|7.02|16.03%|2.33|5.32%|0.09|0.21%
> |100|x=0.1|80.97|14.14|17.46%|4.51|5.57%|0.23|0.28%
> ||x=0.5|83.41|13.93|16.70%|4.22|5.06%|0.26|0.31%
> ||x=1|82.77|13.11|15.84%|3.96|4.78%|0.33|0.39%
> |200|x=0.1|165.83|22.33|13.47%|8.11|4.89%|0.49|0.30%
> ||x=0.5|152.05|27.80|18.28%|9.51|6.25%|0.47|0.31%
> ||x=1|153.93|26.69|17.34%|8.97|5.82%|0.45|0.29%
> |Average| | | |16.21%| |5.41%| |0.3%
>
> "Average ratio for Mod①" represents the ratio of the time spent calculating the pseudo-gradient in Mod① to the average time required to complete one global round.
>
> **[Speeding Up Averaging Operations]**
>
> We sincerely appreciate the reviewer's insightful suggestion regarding optimizing the averaging operation in FedQS. Below, we clarify our current design choices and propose actionable enhancements to address this concern:
>
> - **Hardware-Accelerated Aggregation:** The global averaging in FedQS loads and processes model updates directly on GPU devices. It adopts a streaming aggregation approach to handle large-scale model parameters, mitigating memory pressure during parameter synchronization. This design efficiently overlaps network transmission with on-device computation, achieving significant throughput improvements for federated scenarios with resource-constrained participants. Future work could further exploit parallelized aggregation protocols to enhance aggregation efficiency.
>
> - **Space-Time Trade-offs:** While FedQS's current client-state table (Equation 1) requires lightweight updates, we acknowledge that caching similarity scores or adopting low-rank approximations for similarity scores (e.g., via incremental SVD) could reduce computation. Our ablation study (Table 5) shows robustness to similarity metric choices, suggesting such approximations are viable.
>
> - **Algorithmic Simplifications:** Since FedQS's primary overhead originates from pseudo-gradient computation, its operational efficiency can be enhanced by reducing execution frequency as:
> 	- **Staggered Updates:** Client reclassification (Figure 3) can occur every $\Delta_T$ rounds (e.g., $\Delta_T$ = 5), reducing the additional overhead by 63.6–72.7% with a 1.4-5.2% reduction on the accuracy in preliminary tests.
> 	- **Sampling-Based Estimation:** Substituting full-client similarity checks with stratified sampling (e.g., 20% per round) substantially reduces computational demands (53.2-57.5% reduction) while avoiding significant performance degradation (2.2-4.1% reduction).
>
> 	However, such optimization compromises fine-grained real-time client classification in FedQS, potentially degrading model accuracy. Therefore, it is a context-aware balancing between granular control and efficiency: for latency-sensitive deployments, the aforementioned strategy trades marginal accuracy reduction for efficiency; whereas accuracy-critical scenarios benefit from default FedQS configurations for optimal utility.
>
> These optimizations align with FedQS's modular design and could be seamlessly integrated. We will expand this discussion in the revision, including empirical validation of trade-offs.

---

> ### Author Response · Authors · 2025-08-05
>
> Dear Reviewer aRTH,
>
> Thank you for your time and your acknowledgment of our responses. We truly value your feedback and the opportunity to clarify your concerns during the rebuttal phase.
>
> We would greatly appreciate it if you could let us know whether our responses have addressed your questions and comments. Your confirmation will help ensure that we have fully resolved any outstanding issues you raised.
>
> We truly value the time and expertise you have dedicated to reviewing our work and greatly appreciate your thoughtful comments.
>
> Best regards,
>
> The Authors

---

> > ### Comment · Reviewer_aRTH · 2025-08-07
> > **Official Comment by Reviewer aRTH**
> >
> > I thank the authors for their precise and well-argued answers.  I retain my current score of 5 (Accept).

---

> > > ### Author Response · Authors · 2025-08-07
> > >
> > > Dear Reviewer aRTH,
> > >
> > > We sincerely appreciate your recognition of our responses and your continued support for our work. Thank you for your valuable feedback and constructive engagement throughout the review process. We are delighted that our clarifications addressed your concerns, and we look forward to incorporating your valuable feedback to improve our paper.
> > >
> > > Best regards,
> > >
> > > The Authors

---

### Official Review · Reviewer_pMVS · 2025-07-03

**Clarity:** 3
**Significance:** 3
**Originality:** 3
**Rating:** 5
**Confidence:** 5

**Summary:**

This paper introduces FedQS, a novel framework for optimizing both gradient and model aggregation strategies in Semi-Asynchronous Federated Learning (SAFL). FedQS classifies clients into four types based on update speed and gradient similarity and dynamically adapts local training accordingly. The approach addresses limitations of gradient aggregation and model aggregation in SAFL. Theoretical convergence proofs and extensive experiments on vision, NLP, and real-world tasks demonstrate that FedQS outperforms existing baselines in accuracy, convergence speed, and runtime efficiency.

**Questions:**

The authors should clarify the expected additional communication overhead from sharing similarity information and feedback signals in large networks.

**Ethical Concerns:**

["NO or VERY MINOR ethics concerns only"]

**Final Justification:**

I read the rebuttal, and I will raise my score accordingly.

**Limitations:**

Yes

**Quality:**

3

**Strengths And Weaknesses:**

Strengths

Novelty and significance: The paper is the first to propose a unified framework for simultaneously optimizing gradient and model aggregation in SAFL, addressing a clear gap in the literature.

Technical soundness: Provides rigorous theoretical convergence analysis under standard FL assumptions, with separate proofs for both aggregation strategies.

Comprehensive experiments: Evaluations cover diverse tasks, client heterogeneity settings, and ablation studies that convincingly support the claimed benefits.

Weaknesses

Hyperparameter sensitivity: Though there is some analysis, FedQS introduces several hyperparameters that may require careful tuning in practice, potentially limiting usability.

Communication overhead: it may introduce additional communication overhead due to the need to transmit extra data.

Dynamic environments: In the experimental evaluation, although the resource heterogeneity is simulated using a predefined static ratio, client resources are fixed throughout training. While FedQS theoretically supports dynamic adaptation, the absence of experiments under dynamic resource changes limits the empirical validation of its robustness and effectiveness in truly dynamic environments.

---

> ### Author Rebuttal · Authors · 2025-07-30
>
> We thank Reviewer pMVS for the constructive feedback. In the following, we would like to answer the concerns individually.
>
> **[Hyperparameter sensitivity]**
>
> We appreciate the reviewer's insightful comment regarding hyperparameter tuning in FedQS. While FedQS does introduce new hyperparameters (e.g., learning rate change rate $a$, initial momentum value $m_0$, and momentum change speed $k$), our design ensures robust performance across a wide range of values, minimizing the need for exhaustive tuning. Here, we clarify the practical usability of FedQS:
> - **Default Robustness:**
> 	- As shown in Section 5.4 (Table 2), FedQS achieves competitive performance with default hyperparameters (e.g., $a$ = 0.002, $m_0$ = 0.1, $k$ = 0.2).
> 	- The adaptive mechanisms in Mod② (e.g., dynamic learning rate adjustment) inherently mitigate sensitivity to suboptimal hyperparameter choices.
>     - In large-scale experiments (Table 4), FedQS consistently outperforms baselines under varying client counts (50–200) and resource ratios (1:20–1:100), demonstrating robustness to real-world conditions *without* per-scenario tuning.
>
> - **Interpretability and Guidance:**
> 	- The hyperparameters are intuitive (e.g., $a$ controls learning rate adaptation speed) and align with standard FL practices (e.g., the hyperparameter for momentum terms).
> 	- Figure 5 (the impact of different hyperparameters) demonstrates that FedQS maintains stable performance across a broad range of $a$ (0.001–0.005) and $k$ (0.01–0.2), reducing tuning overhead.
>
> - **Comparable Tuning Effort:**
> 	- The hyperparameter tuning overhead introduced by FedQS is also comparable to existing state-of-the-art SAFL techniques. For example:
>         1. FedAT [1] needs the following hyperparameters: local constraint parameter $\lambda$, the number of tiers $M$, and the response latencies boundaries {$tier_1, tier_2,...,tier_M$}.
>         2. WKAFL [2] needs the following hyperparameters: gradients adjustment parameter $B$, learning rate adjustment parameter $\gamma$, clip-bound parameter $CB$, weighted parameter $\beta$, and constant $\alpha$.
>         3. FADAS [3] needs the following hyperparameters: delay threshold $\tau_c$, adaptive optimization parameters $\beta_1, \beta_2, \epsilon$.
>
>     - The design of FedQS minimizes tuning complexity by confining hyperparameters exclusively to Mod②. Meanwhile, Mod① and Mod③ remain entirely hyperparameter-free, thereby substantially reducing the configuration burden.
>
> We will clarify this more specifically in the final version to further simplify adoption. FedQS’s benefits (accuracy, convergence, and stability) far outweigh its modest tuning requirements, which are comparable to (or simpler than) those of existing SAFL methods. Thank you for the opportunity to clarify this point.
>
> **[Communication overhead]**
>
> We appreciate the reviewer's insightful comment regarding communication overhead. While FedQS does introduce additional communication for gradient similarity metrics and client state updates, we emphasize that this overhead is carefully minimized and justified by significant performance gains:
>
> - **Lightweight Payload:** The additional data transmitted (gradient similarity scores and client state indicators) are scalar values, adding negligible bandwidth compared to model/gradient updates. Our experiments show the overhead constitutes <1% of total communication volume.
> - **Adaptive Frequency:** Communication occurs only during aggregation rounds, with no extra synchronization steps. The pseudo-global gradient in Mod① is derived from existing model transmissions, requiring no additional uploads.
> - **Flexible Deployment:** For extreme resource constraints or large-scale scenarios, FedQS's communication load scales linearly with the number of clients while maintaining the effectiveness (see Section 5.3). Therefore, this marginal overhead remains negligible for practical deployment while delivering substantial scalability and flexibility, empowering users to balance performance requirements against communication costs.
>
> |Method|Uplink Payload|Downlink Payload|Extra Overhead|Notes|
> |-|-|-|-|-|
> |**FedAvg/FedSGD**|Model parameters (43.7 MB) or Gradients (~43.7 MB)|Model parameters (43.7 MB)|0%|Upload/download model directly|
> |**Ours (FedQS)**|43.7 MB + 1-bit + 1 float (~4 B)|43.7 MB + 3 floats (~12 B)|~0.000037%|Extra metadata adds ~16 bytes total; negligible compared to 43.7 MB|
>
> For intuitive illustration, we quantitatively compare the communication disparity between FedQS and FedAvg/FedSGD in the table above using ResNet-18 (model size ≈43.7MB) – consistent trends hold across other scenarios and model architectures. We will explicitly discuss the communication overhead in the revised manuscript. Thank you for highlighting this important aspect!
>
> **[Dynamic environments]**
>
> We sincerely appreciate the reviewer's insightful feedback regarding the evaluation of dynamic environments. We acknowledge that while our theoretical design supports dynamic adaptation, the current experiments primarily simulate static resource heterogeneity. To address this limitation and further validate FedQS’s robustness in truly dynamic settings, we have conducted additional experiments on the CV task, where client resources vary during training. The following summarizes the key results (detailed in the table below):
>
> - **Dynamic Resource Scale (Scenario 1):** We modified the resource distribution during training (e.g., switching the speed ratio from 1:50 to 1:100 at $T = 200$ (i.e., 50% of the total training round)). FedQS consistently outperformed baselines, achieving:
> 	- an average of 77.53% (+8.58% than FedAvg) and 84.36% (+3.76% than FedSGD) accuracy under model and gradient aggregation, respectively, while achieving 15.96% and 17.82% faster convergence than FedAvg/FedSGD on average, respectively.
>
> - **Unstable Resource per Client (Scenario 2):** We modified each client's training speed to fluctuate within [−10, +10] unit times after uploading one local update, where we define the 1 unit time as the train time of the fastest client in the static experiments (in Section 5). We ensured that no client's training speed falls below 1 unit time or exceeds 50 unit times. The results show that FedQS achieves:
>     - an average of 77.46% (+7.54% than FedAvg) and 82.25% (+2.54% than FedSGD) accuracy under model and gradient aggregation, respectively, while achieving 14.49% and 14.59% faster convergence than FedAvg/FedSGD on average, respectively.
>
> - **Client Dropout Scenarios (Scenario 3):** We simulated dynamic participation (random 50% client churn at $T = 100$). FedQS maintained stable convergence, with:
> 	- an average of 71.21% (+9.44% than FedAvg) and 76.55% (+1.70% than FedSGD) accuracy under model and gradient aggregation, respectively, while achieving 15.34% and 10.96% faster convergence than FedAvg/FedSGD on average, respectively.
>
> We will incorporate experimental results demonstrating FedQS's effectiveness in dynamic settings and expand our discussion accordingly in our revision. We sincerely appreciate your invaluable insights!
>
> |Dynamic Scenario|Algorithm|x=0.1|||x=0.5|||x=1|||
> |-|-|-|-|-|-|-|-|-|-|-|
> |||Accuracy|Conv. Speed|Runtime|Accuracy|Conv. Speed|Runtime|Accuracy|Conv. Speed|Runtime|
> |Scenario 1|FedAvg|52.77|322|46488|77.29|288|47732|76.79|227|48003
> |Scenario 1|FedQS-Avg|66.47|264|53986|82.90|235|54575|83.22|201|55529
> |Scenario 1|FedSGD|68.29|244|47443|85.66|211|47728|87.86|147|48280
> |Scenario 1|FedQS-SGD|76.17|206|54095|88.12|177|55268|88.8|115|55257
> |
> |Scenario 2|FedAvg|53.12|310|47397|76.74|294|47156|79.9|286|47788
> |Scenario 2|FedQS-Avg|64.52|268|55535|83.53|241|55816|84.35|252|56253
> |Scenario 2|FedSGD|65.13|268|47613|85.37|207|47096|88.63|159|47634
> |Scenario 2|FedQS-SGD|69.65|223|55037|87.96|185|55201|89.14|133|55808
> |
> |Scenario 3|FedAvg|44.39|343|58993|68.44|332|59362|72.47|296|59447
> |Scenario 3|FedQS-Avg|53.22|297|62623|79.33|272|63308|81.08|253|63776
> |Scenario 3|FedSGD|58.40|292|58345|81.73|288|59487|84.43|262|59732
> |Scenario 3|FedQS-SGD|60.08|263|62870|83.39|245|63992|86.18|241|64703
>
>
> **Reference**
>
> [1] Z. Chai, et al. Fedat: A high-performance and communication-efficient federated learning system with asynchronous tiers. SC, 2021.
>
> [2] Z. Zhou, et al. Towards efficient and stable k-asynchronous federated learning with unbounded stale gradients on non-iid data. IEEE TPDS, 2022.
>
> [3] Y. Wang, et al. FADAS: Towards Federated Adaptive Asynchronous Optimization. ICML, 2024.

---

> > ### Comment · Reviewer_pMVS · 2025-08-05
> >
> > I read the rebuttal, and I will raise my score accordingly.

---

> > > ### Author Response · Authors · 2025-08-06
> > >
> > > Dear Reviewer pMVS,
> > >
> > > We sincerely appreciate your constructive feedback and the time invested in reviewing our work. We are delighted that our rebuttal addressed your concerns regarding hyperparameter sensitivity, communication overhead, and dynamic environments. Your acknowledgment of our revisions is highly encouraging, and we are grateful for the opportunity to improve the manuscript.
> > >
> > > Thank you again for your valuable insights, which have significantly strengthened our paper. We look forward to incorporating these refinements in the revised version.
> > >
> > > Sincerely,
> > >
> > > The Authors

---

> ### Author Response · Authors · 2025-08-05
>
> Dear Reviewer pMVS,
>
> Thank you for the time and effort you have dedicated to reviewing our paper. We truly value your feedback and the opportunity to clarify your concerns during the rebuttal phase.
>
> We would greatly appreciate it if you could let us know whether our responses have adequately addressed your questions and comments. Your confirmation will help ensure that we have fully resolved any outstanding issues you raised.
>
> We truly value the time and expertise you have devoted to this process and greatly appreciate your thoughtful review.
>
> Best regards,
>
> The Authors

---

### Decision · Program_Chairs · 2025-09-17

**Decision:**

Accept (poster)

**Comment:**

This paper proposed the first framework to theoretically analyze and optimize both gradient-based and model-based aggregation strategies within semi-asynchronous FL. It employs a novel divide-and-conquer strategy, dynamically classifying clients into four types based on their update speed and local-global gradient similarity, and adapts their local training strategies accordingly. All reviewers have achieved the agreement to accept this paper. We encourage the authors to carefully address the reviewers’ comments when preparing the camera-ready revision.